# Saturn: Sample-efficient Generative Molecular Design using Memory Manipulation

## Abstract

Generative molecular design for drug discovery has very recently achieved a wave of experimental validation, with language-based backbones being the most common architectures employed. The most important factor for downstream success is whether an *in silico* oracle is well correlated with the desired end-point. To this end, current methods use cheaper proxy oracles with higher throughput before evaluating the most promising subset with high-fidelity oracles. The ability to *directly* optimize high-fidelity oracles would greatly enhance generative design and be expected to improve hit rates. However, current models are not efficient enough to consider such a prospect, exemplifying the sample efficiency problem. In this work, we introduce **Saturn**, which demonstrates the first application of the Mamba architecture for generative molecular design. We elucidate *how* experience replay with data augmentation improves sample efficiency and *how* Mamba synergistically exploits this mechanism. Saturn outperforms 22 models on multi-parameter optimization tasks relevant to drug discovery and may possess sufficient sample efficiency to consider the prospect of directly optimizing high-fidelity oracles. The code is available at `https://figshare.com/s/21059896530e222b9cd5`.

## 1 Introduction

Within the last year, there has been a surge of works reporting experimental validation of generative molecular design for drug discovery (Du et al., 2024). The fundamental task of generative molecular design is learn a distribution of molecules with *tailored* property profiles. All generative models achieve this in one of two ways: distribution learning, where a base model is subjected to transfer learning on a set of known positives, and goal-directed generation, which encompasses both conditional generation and using an optimization algorithm to shift the distribution. Experimental validation has been demonstrated for all methods, but with a notable over-representation from optimization algorithms as of the last 9 months, particularly reinforcement learning (RL) (Du et al., 2024). Algorithmic molecular optimization always proceeds via the following workflow: generate molecules, assess *desirability* using an *in silico* oracle, update the model, and repeat. When assessing the suitability of molecules absent experimental validation, the crucial indicator to success is *correlation* of an *in silico* oracle to the actual end-point. All protocols that *directly* optimize for an oracle without the use of a surrogate predictor follow a funnel workflow where less resource-intensive oracles are initially used to prioritize the most promising subset for evaluation with computationally expensive high-fidelity oracles. A concrete and ubiquitous example is designing molecules with high binding affinity to a protein target. By far the most common oracle used to estimate binding affinity is molecular docking, and many works (Guo et al., 2021; Thomas et al., 2022; Shen et al., 2023; Yang et al., 2021; Lee et al., 2023; 2024; Fu et al., 2022) have demonstrated the ability to generate molecules with improved docking scores. However, docking scores are often poorly correlated with binding affinity, especially when applied out-of-the-box (Guo et al., 2021; Crivelli-Decker et al., 2024). Correspondingly, the most promising candidates from docking are subjected to higher-fidelity oracles, particularly molecular dynamics (MD) simulations, which offer a much more accurate estimation of binding affinity (Wang et al., 2019; Moore et al., 2022; 2023; Crivelli-Decker et al., 2024), but with industry-standard methods typically being closed-source (Moore et al., 2023). *Directly* optimizing high-fidelity oracles offers the prospect of learning the distribution and can greatly improve the quality of the generated set (Eckmann et al., 2024). However, doing so is infeasible due to computational cost, exemplifying the sample efficiency problem. Either simulation protocols become much faster without sacrificing

accuracy, or generative models become *sufficiently efficient* to optimize under an acceptable oracle budget. We note that QSAR models are often used, which can have great predictive accuracy, but may suffer from a narrow domain of applicability (within their training data) (Neves et al., 2018).

Recently, more works have explicitly considered sample efficiency by constraining the oracle budget on various drug discovery optimization tasks (Yang et al., 2021; Fu et al., 2022; Guo & Schwaller, 2024a;b; Lee et al., 2023; 2024; Shen et al., 2023). More recently, Guo et al. (Guo & Schwaller, 2024a) proposed Augmented Memory which is built on REINVENT (Olivecrona et al., 2017; Blaschke et al., 2020a). It combines experience replay with SMILES augmentation (Weininger, 1988; Bjerrum, 2017) and empirically shows that this data augmentation can improve sample efficiency. In this work, we push towards the prospect of direct optimization of high-fidelity oracles and release **Saturn**. First, we elucidate the mechanism of Augmented Memory, which uses an LSTM (Hochreiter & Schmidhuber, 1997) recurrent neural network (RNN) as the language model backbone, and characterize exactly *how* data augmentation and experience replay improve sample efficiency. Next, we systematically assess more advanced generative architectures from just RNNs (Hochreiter & Schmidhuber, 1997) to decoder transformers (Vaswani et al., 2017; Radford et al., 2019), and the recent Mamba (Gu & Dao, 2023) state space model (SSM). Our results show that the Mamba architecture, in conjunction with data augmentation and experience replay, displays synergistic behavior to improve sample efficiency by *strategic* overfitting. Our contribution is as follows:

1. We show the first application of Mamba for molecular generative design and specifically for goal-directed generation with reinforcement learning.

2. We elucidate the mechanism into *how* Augmented Memory improves sample efficiency, as the original work only showed its empirical benefits.

3. We comprehensively evaluate language model backbones (> 500 experiments, all across 10 seeds) including RNN, decoder transformer, and Mamba, which enables us to characterize *model-intrinsic* and *scaling* properties that lead to improved sample efficiency.

4. Through ablation studies, we demonstrate that *local sampling* in chemical space is a key component for sample efficiency. Our results provide discourse on the nature of optimization landscapes *commonly* encountered in drug discovery.

5. We propose **Saturn**, which leverages Mamba and outperforms 22 models on multi-parameter optimization (MPO) drug discovery tasks under heavily-constrained oracle budgets.

## 2 RELATED WORK

**Sample Efficiency in Goal-directed Molecular Design.** The goal of inverse design is to achieve *tailored* molecular generation. Existing works have tackled this problem using a variety of architectures, including SMILES (Weininger, 1988)-based RNNs (Olivecrona et al., 2017; Segler et al., 2018; Popova et al., 2018; Neeser et al., 2023), transformers (Vaswani et al., 2017; Radford et al., 2019; Bagal et al., 2021; Wang et al., 2023; Feng et al., 2023; Mazuz et al., 2023; Hu et al., 2024; He et al., 2024), variational autoencoders (VAEs) (Kingma & Welling, 2013; Gómez-Bombarelli et al., 2018; Jin et al., 2018; Zhavoronkov et al., 2019), adversarial approaches (Goodfellow et al., 2014; Kadurin et al., 2017; De Cao & Kipf, 2018; Ivanenkov et al., 2023), graph-based models (You et al., 2018; Jin et al., 2020b; Mercado et al., 2021; Yang et al., 2021; Maziarz et al., 2022; Vignac et al., 2023), GFlowNets (Bengio et al., 2023; 2021; Shen et al., 2023), genetic algorithms (GAs) (Mitchell, 1998; Jensen, 2019; Fu et al., 2022; Lee et al., 2024), and diffusion models (Lee et al., 2023; Igashov et al., 2024; Schneuing et al., 2023). However, many works do not explicitly consider an oracle budget (or use a very lenient budget) and focus mostly on showing that goal-directed generation is possible. The release of the PMO benchmark (Gao et al., 2022) highlighted that improvements in sample efficiency are vital to even consider the prospect of directly optimizing high-fidelity oracles, e.g., MD may take GPU hours per molecules (Moore et al., 2023). In the benchmark, the oracle budget is 10,000, but as we push towards high-fidelity oracles, a more stringent budget would be necessary. More recent works (Yang et al., 2021; Fu et al., 2022; Guo & Schwaller, 2024a;b; Lee et al., 2023; 2024; Shen et al., 2023) have enforced fixed oracle budgets when comparing performance against other methods. All the objective functions considered in these works include docking, which is used in *every single* experimentally validated structure-based generative design case study (Du et al., 2024)

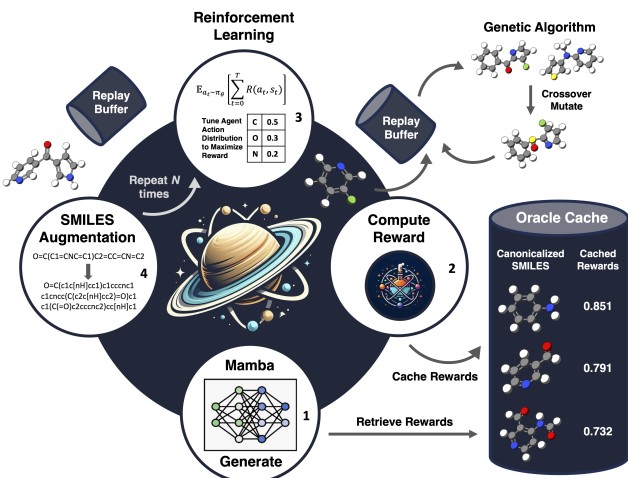

Figure 1: Saturn generative workflow. All generated SMILES and their rewards are stored in the Oracle Cache after canonicalization. A genetic algorithm can be optionally applied using the replay buffer as the parent population. Augmented Memory is used to update the Agent numerous times.

and in commercial drug discovery (Pun et al., 2023). Correspondingly, in this work, we consider a wide range of docking tasks under heavily-constrained oracle budgets (1,000 or 3,000).

**Language-based Molecular Generative Models.** Text is one of the most widely used molecular representations, with common ones being simplified molecular-input line-entry systems (SMILES) (Weininger, 1988) and self-referencing embedded strings (SELFIES) (Krenn et al., 2020; 2022). Recent work has shown that the former is generally more performant, despite not enforcing 100% validity (Gao et al., 2022; Skinnider, 2024). Leveraging advances in natural language processing (NLP), language-based molecular generative models are amongst the first and still widely used models, encompassing RNNsOlivecrona et al. (2017); Segler et al. (2018); Popova et al. (2018), transformers (Vaswani et al., 2017; Radford et al., 2019; Bagal et al., 2021; Wang et al., 2023; Feng et al., 2023; Mazuz et al., 2023; Hu et al., 2024; He et al., 2024), and recently SSM S4 (Özçelik et al., 2024). In early benchmarks (GuacaMol (Brown et al., 2019) and MOSES (Polykovskiy et al., 2020)), language-based models have been shown to essentially solve the validity, uniqueness, and novelty metrics. Subsequently, the non-injective syntax of SMILES confers advantageous properties for generative design. Specifically, a single molecule can be expressed as at least $N$ (number of heavy atoms) SMILES, in a process known as SMILES augmentation, enumeration, or randomization (Bjerrum, 2017). This mechanism can be exploited to pre-train models under low data regimes to generalize in chemical space (Arús-Pous et al., 2019; Moret et al., 2020; Skinnider et al., 2021), improve sample efficiency (Bjerrum et al., 2023; Guo & Schwaller, 2024a), and perform transfer learning with a single positive example (Ballarotto et al., 2023). Despite the recent trend towards 3D molecular generation (Igashov et al., 2024; Schneuing et al., 2023), language-based models have demonstrated the ability to generate molecules that satisfy 3D-dependent objectives, such as docking in a sample-efficient manner (Guo & Schwaller, 2024a;b). This suggests that language-based models are not entirely 3D-naive and can effectively explore relevant regions of the 3D chemical space. Finally, language models are amongst the most sample-efficient models (Gao et al., 2022; Polykovskiy et al., 2020; Brown et al., 2019) and most studies achieving experimental validation of a generated molecule incorporate SMILES-based models (Du et al., 2024).

## 3 METHOD

In this section, each component of Saturn (Fig. 1) is described: the language model backbone for molecular generation, the Augmented Memory (Guo & Schwaller, 2024a) RL algorithm, the GA, and specific details into key components responsible for sample efficiency and mitigating mode collapse.

**Autoregressive Language Model Backbone for Molecular Generation.** Molecules are represented as SMILES (Weininger, 1988) and the task of goal-directed generation is cast as an RL problem. Let

$S_t$ denote the state space representing all intermediate token sequences during molecular generation. The action space, $A_t(s_t)$, is defined as the conditional token distribution induced by the policy, $\pi_\theta$, and parameterized by a language model backbone. In this work, we investigated RNN, decoder transformer, and Mamba backbones with the latter chosen as the default after extensive experimentation. Therefore $\theta$ represents the parameters of the Mamba backbone (Gu & Dao, 2023; Chen, 2024), which is a state-space model (Gu et al., 2021b;a) and features four *learnable* matrices to propagate sequence information: $\bar{\mathbf{A}}, \bar{\mathbf{B}}, \bar{\mathbf{C}}, \bar{\mathbf{D}}$:

$$h_t = \bar{\mathbf{A}} h_{t-1} + \bar{\mathbf{B}}(x_t) x_t$$
$$y_t = \bar{\mathbf{C}}(x_t) h_t + \bar{\mathbf{D}} x_t$$

where $h$ is the state, $x$ is the input sequence (SMILES in this work), and $y$ is the output. Importantly, the input-dependent parameters confer a *selective* mechanism that can handle contextual importance, and notably differs from previous state-space models (Gu et al., 2021b;a). Like other language models, a linear projection transforms the Mamba output to a multinomial token distribution. This enables sequence generation, which we define as a Markov process, and thus, sampling a SMILES, $x$, is given by the product of conditional token probabilities (Eq. 1):

$$P(x) = \prod_{t=1}^{T} \pi_{\theta_{\text{Agent}}}(a_t \mid s_t) \tag{1}$$

where $\pi_\theta$ is the Mamba backbone and referred to as the *Agent* to match RL terminology and $a_t$ and $s_t$ are the token selected and token sequence so far, at time-step $t$, respectively. We couple RL to the generative process to enable multi-parameter optimization (MPO). The general objective in RL is to maximize the expected reward (Eq. 2):

$$J(\theta) = \mathbb{E}_{a_t \sim \pi_{\theta_{\text{Agent}}}} \left[ \sum_{t=1}^{T} R(a_t, s_t) \right] \tag{2}$$

$R$ is the reward function and can represent any arbitrary MPO objective and $\sigma$ is a scalar factor modulating its effect. Next, the Augmented Likelihood (Olivecrona et al., 2017) (Eq. 3) is defined, where the prior is the pre-trained model with *frozen* weights:

$$\log \pi_{\text{Augmented}}(x) = \log \pi_{\text{prior}}(x) + \sigma R(x) \tag{3}$$

The reward is defined as $\log \pi_{\text{Augmented}}$ - $\log \pi_{\theta_{\text{Agent}}}$. Following previous works (Olivecrona et al., 2017; Fialková et al., 2021; Guo & Schwaller, 2024a), maximizing Eq. 2 is equivalent (up to a factor) to minimizing the squared difference between the Augmented Likelihood and the Agent Likelihood (Eq. 4):

$$L(\theta) = \frac{1}{|B|} \left[ \sum_{a \in A^*} (\log \pi_{\text{Augmented}} - \log \pi_{\theta_{\text{Agent}}}) \right]^2 \tag{4}$$

$A^*$ is defined as the actions taken across all time-steps in a given batch. During optimization, the expected reward (Eq. 2) is approximated by sampling a batch, $B$, of SMILES. The batch size controls for variance as approximating the expectation with fewer samples is necessarily more noisy. See Appendix B.5 for full details on the algorithm and pseudo-code.

**Augmented Memory.** In Saturn, Augmented Memory maintains a replay buffer of the top 100 SMILES ranked by their rewards. At each generation epoch, the Agent is updated $N$ augmentation rounds times. *Each* augmentation round involves taking every SMILES in the buffer, augmenting (randomizing) (Bjerrum, 2017) them, and updating the Agent following Eq. 4. A Diversity Filter (DF) (Blaschke et al., 2020b) stores the Bemis-Murcko (Bemis & Murcko, 1996) scaffolds of every

SMILES generated. If a scaffold is generated more than a permitted threshold ($M = 10$ in this work), its reward is truncated to 0. Before executing Augmented Memory, scaffolds associated with penalized rewards are purged from the buffer, preventing mode collapse.

**Genetic Algorithm.** Saturn adapts GraphGA (Jensen, 2019) where the replay buffer is treated as the parent population. The motivation is to generate more high reward SMILES to *replace* the buffer SMILES, under the hypothesis that on average, these too, will be high reward (Appendix C.5).

**Differences to Previous Works.** Saturn adapts Augmented Memory (Guo & Schwaller, 2024a) but differs in several important ways. Firstly, unlike the original work, we elucidate the mechanism into *why* Augmented Memory can improve sample efficiency and *explicitly* show that it makes generating the replay buffer molecules likely. The following Results section will show that high sample efficiency can be achieved by *local sampling*, whereby the modeled distribution is strategically *overfit* on these replay buffer molecules. Precisely, this means making the Agent particularly likely, but *not* deterministic, to generate *any SMILES sequence form* of the replay buffer molecules. By nature of multinomial decoding, stochastic generation means that unique sampled molecules might only differ by a small number of tokens, which translates to the molecules differing by a small number of atoms. Secondly, we show that Mamba synergistically enhances this mechanism by nature of being a proficient distribution learner. Exactly because Mamba can overfit the distribution of replay buffer molecules, it displays the greatest degree of *local sampling*. To accommodate repeat generated molecules, we introduce an oracle cache under the assumption that oracle evaluations are *near deterministic* (for docking oracles, we fix the seed). If the same SMILES is generated at a later epoch, the reward is retrieved from the cache and does not impose an oracle call. Finally, by showing that strategic overfitting can be beneficial, we further demonstrate that scaling up architectures (Appendix F.6) can also improve sample efficiency. This offers discourse into benefits of architectural differences in the small molecule goal-directed generation regime. We show that there are benefits despite the modeled sequences being relatively short (< 80 tokens).

## 4 RESULTS AND DISCUSSION

The results section is comprised of three parts: formulating Saturn on a toy MPO task, demonstrating sample efficiency on an MPO docking (3 targets) task, and benchmarking against 22 models (including dataset screening baselines) on another MPO docking (5 targets) task which also considers synthesizability. **Every experiment was run across 10 seeds (0-9 inclusive)**, comprising > 5,000 experiments.

### 4.1 PART 1: ELUCIDATING THE OPTIMIZATION DYNAMICS OF SATURN

We begin by identifying the optimal architecture and hyperparameters for Saturn. First, we experiment with varying the batch size and augmentation rounds of the Augmented Memory algorithm (Guo & Schwaller, 2024a), and explicitly demonstrate the trade-off between sample efficiency and diversity. Unlike the original Augmented Memory work, which used an RNN backbone, we investigate more advanced architectures: decoder transformer (Vaswani et al., 2017; Radford et al., 2019) and Mamba (Gu & Dao, 2023). Our analysis elucidates *how* SMILES augmentation, combined with these architectures, synergistically improves sample efficiency in Saturn. The key mechanism is *local sampling* in chemical space, whereby relatively small atomic changes are made to high-reward replay buffer molecules.

**Experimental Details.** We define a toy experiment with the following MPO objective: molecular weight (MW) < 350 Da, number of rings $\geq 2$, and maximize topological polar surface area (tPSA) (Guo & Schwaller, 2024b). Optimizing this objective *requires* generating molecules with rings saturated with heteroatoms, which are dissimilar from the training data. Hence, it is also testing out-of-distribution optimization. All experiments in this section were run across 10 seeds (0-9 inclusive) with an oracle budget of 1,000, and the models were pre-trained with ChEMBL 33 (Gaulton et al., 2012) (Appendix C.1).

**Metrics.** The sample efficiency metrics are **Yield** and **Oracle Burden** (OB). Yield is the number of *unique* generated molecules above a reward threshold, and OB is the number of oracle calls required

to generate $N$ *unique* molecules above a reward threshold. The reward threshold in this experiment is 0.7 as molecules start to possess saturated heteroatom rings. Most configurations generate at least *some* molecules passing this threshold within the budget, enabling us to report statistics.

**Understanding the Limits of Augmented Memory.** Augmented Memory (Guo & Schwaller, 2024a) improves sample efficiency by repeated learning from high reward SMILES. With decreasing batch size, performance variance increases, as the approximation to the expected reward (Eq. 2) becomes more noisy. In return, fewer oracle calls are imposed, and the Agent learns from an increasingly smaller set of unique SMILES. We hypothesize that as long as unique high reward SMILES are still generated, sample efficiency can improve with decreasing batch size, at the expense of diversity. We perform a grid search and vary the batch size (64, 32, 16, 8) and augmentation rounds (0-20 inclusive) using the default RNN architecture (Appendix 5). We make the following key observations: with *increasing* augmentation rounds and *decreasing* batch size, sample efficiency improves, diversity decreases, and generating repeated SMILES becomes increasingly prevalent but is tolerable with oracle caching. The optimal augmentation rounds and batch size are 5-10 and 16, respectively, as pushing further introduces *too much* variance, such that apparent improvements are not statistically significant (at the 95% confidence level). In Appendix C.4, we explored the addition of Beam Enumeration (Guo & Schwaller, 2024b) but improvements were not consistently statistically significant. In Appendix C.5, we explored allocating a portion of the oracle budget to a GA, which decreases sample efficiency, but recovers diversity, in agreement with previous works (Liu et al., 2021; Lee et al., 2024). Finally, see Appendix C.2 for systematic ablation studies on the effect of every component of Saturn.

**Small Molecule Goal-directed Generation: Beyond RNNs.** In this section, we move beyond **RNN** (5.8M) to **Decoder** transformer (Vaswani et al., 2017; Radford et al., 2019) (6.3M) and **Mamba** (Gu & Dao, 2023) (5.2M) (see Appendix B.2 for Mamba details), and empirically show that varying the architecture can improve sample efficiency. Complete grid search results are presented in Appendix C.3. We make the following observations: Increasing augmentation rounds decreases diversity and *inconsistently* improves Yield and OB for RNN and transformer. Mamba *more consistently* benefits from increasing augmentation rounds to generate more high reward molecules and also faster. Across the Yield and OB metrics, Mamba consistently outperforms both the RNN and transformer backbones. Given Mamba's superior sample efficiency, we focus our analysis on comparing it to the RNN baseline in the remainder of this section (transformer results are provided in Appendix C.3)

**Mamba: Enhanced Maximum Likelihood.** Table 15 shows that the Mamba architecture notably generates repeated SMILES, which can be rationalized with the maximum likelihood objective. Mamba (5.2M) and RNN (5.8M) have similar parameter counts but during pre-training, the former converges to a lower loss during pre-training (Appendix C.1), indicating a better match to the data distribution. Accordingly, and during RL, Eq. 4 aims to make generating high reward SMILES *more likely*. Mamba generates repeated SMILES suggesting it overfits the data distribution. We demonstrate this by cross-referencing Fig. 2a, which shows that with high augmentation rounds, the average max conditional token probability (during generation) approaches 1, and near collapses to a Dirac delta function (less so for RNN). This makes it likely, but *not* deterministic, to generate the same SMILES repeatedly.

**Squeezing the Likelihood of Augmented SMILES.** While the original Augmented Memory work (Guo & Schwaller, 2024a) demonstrated its empirical benefits, we elucidate the underlying mechanism. To isolate its effect, we design a sub-experiment as follows: generate molecules until the buffer is full (100) and then save the Agent state before and after executing Augmented Memory (10 augmentation rounds) and save every augmented SMILES form. After execution, the (End) Agent becomes more likely to generate the set of augmented SMILES (Fig. 2b). The more *improbable* the SMILES (high NLL), the larger the $\Delta$NLL shift (Fig. 2c). According to the loss function (Eq. 4), a larger difference between the Augmented Likelihood (Eq. 3) and Agent Likelihood results in a higher loss. When these terms are near equal, the loss approaches 0 (Fig. 2c circles). The purpose of the Augmented Likelihood is to regularize the Agent, preventing it from deviating *too far* from the prior (Olivecrona et al., 2017). Improbable SMILES, which impose a large gradient update, adjust the Agent towards a higher probability of generating such sequences. However, already probable (low NLL) SMILES can also impose large loss magnitudes (Fig. 2c), but the $\Delta$NLL shift is small because the softmax function saturates, causing minimal changes to the softmax output when the logits are tuned. Taking these observations together, Augmented Memory squeezes the likelihood of augmented SMILES, making the Agent more likely to generate *any* SMILES representation of the

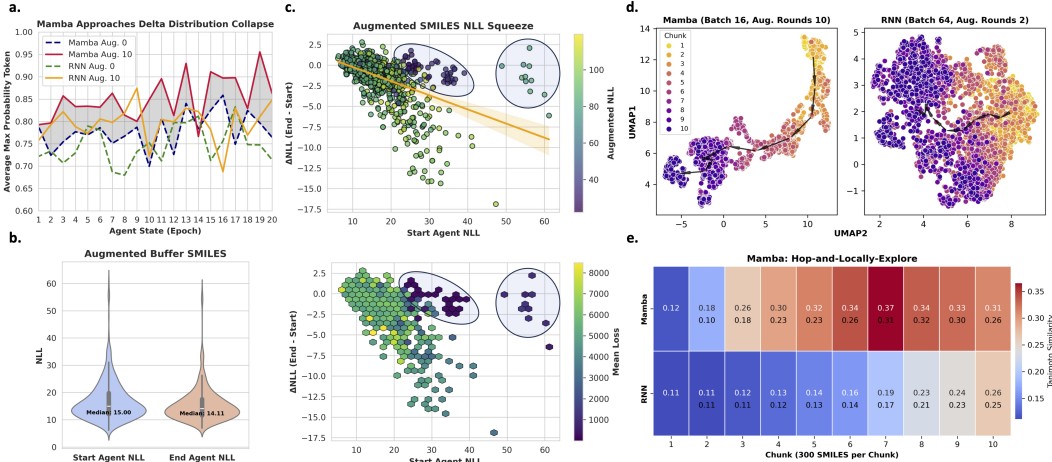

Figure 2: **a.** Average maximum token probability across Agent states. Augmentation pushes the Agent action distribution towards a delta distribution. **b.** Augmented Memory (10 augmentation rounds) makes the likelihood of generating SMILES in the buffer more likely. **c.** Top: On average, augmented forms of the buffer SMILES become more likely. Bottom: Similar loss magnitudes impose larger changes on improbable sequences and the Agent is driven towards generating these specific sequences. When the Augmented Likelihood is equal to the Agent likelihood, the loss approaches 0 (circles). **d.** 3,000 oracle budget test experiment chunked into 300 SMILES. UMAP embedding of the Agent chemical space traversal (arrows are the centroid of each chunk). Mamba exhibits a directional traversal while RNN (baseline Augmented Memory) continues to sample globally. **e.** Mamba exhibits a "hop-and-locally-explore" behavior where the intra-chunk Tanimoto similarity (top values) are higher than RNN. The bottom value is the inter-chunk similarity.

same molecular graph. We next demonstrate how the Mamba architecture synergistically leverages this mechanism to enhance sample efficiency.

**Mamba: Hop-and-Locally-Explore.** Mamba approaches Dirac delta function collapse (Fig. 2a) when learning from repeated augmented SMILES and in the previous section, we have shown that the Agent becomes increasingly likely to generate the buffer *molecules*. We hypothesized that Mamba exhibits a "hop-and-locally-explore" behavior: because it is likely to generate *some* SMILES representation of these molecules (*strategic overfitting*), small changes to any tokens in these set of augmented sequences equates to small changes to the *same* molecular graph, essentially performing a local exploration (similar molecules, on average, exhibit similar properties, provided the property landscape is not too rough (Aldeghi et al., 2022)). We verify our hypothesis with the following experiment: generate molecules (3,000 oracle budget) and separate the generated set into 10 chunks (each 300 SMILES). We trace the generation trajectory using UMAP (McInnes et al., 2018) and plot the chunk centroids, comparing Mamba and the baseline (vanilla Augmented Memory) (Fig. 2d). Mamba traverses chemical space in an increased directional manner and the chunks are more locally confined. Further analysis into the intra- and inter-chunk Tanimoto similarity reveals that *within* chunks, Mamba exhibits much greater similarity than the baseline, and similarity is always lower *between* chunks (Fig. 2e). Taking these observations together, Mamba (batch size 16) with Augmented Memory (10 augmentation rounds) and oracle caching synergistically improves sample efficiency via "hop-and-locally-explore" behavior (see Appendix D for further quantitative and qualitative analyses). From here on, this model configuration will be referred to as **Saturn** and hyperparameters are *fixed* such that all performance metrics in the following sections are out-of-the-box.

### 4.2 PART 2: TRANSFERABILITY OF SAMPLE EFFICIENCY TO PHYSICS-BASED ORACLES

In this section, we demonstrate that Saturn's sample efficiency transfers to an MPO objective involving docking against targets related to neurodegeneration (DRD2 (Wang et al., 2018) and AChE (Kryger et al., 1999)) and inflammation (MK2 kinase (Argiriadi et al., 2010)). The optimization objective is to constrain MW < 500 Da, maximize the quantitative estimate of drug-likeness (QED) (Bickerton

et al., 2012), and minimize AutoDock Vina (Trott & Olson, 2010) docking score (see Appendix E.1 for details on the docking protocol). All experiments were run across 10 seeds (0-9 inclusive) and with a 1,000 oracle budget. We compare Saturn (with and without GA) to baseline Augmented Memory (Guo & Schwaller, 2024a) using the Yield and OB metrics. Saturn generates more high reward molecules and faster, given the fixed oracle budget (Table 1). This holds even for the more challenging MK2 kinase target where the pre-training data (ChEMBL 33 (Gaulton et al., 2012)) is less suited. Furthermore, in agreement with the results from the test experiments, adding a GA on the buffer does not improve sample efficiency but recovers diversity, which can be useful in certain cases.

Table 1: Docking MPO with 1,000 oracle budget. Baseline is vanilla Augmented Memory (Guo & Schwaller, 2024a). IntDiv1 (Polykovskiy et al., 2020) is the internal diversity, Scaffolds is the number of unique Bemis-Murcko (Bemis & Murcko, 1996) scaffolds, OB is Oracle Burden (oracle calls required to generate $N$ unique molecules). All metrics are computed at the 0.8 reward threshold. The number in parentheses in the OB statistics represents how many runs out of 10 were successful. The mean and standard deviation across 10 seeds (0-9 inclusive) is reported. Best models (statistically significant at the 95% confidence level) are bolded.

| Target | Model | Yield (↑) | IntDiv1 (↑) | Scaffolds (↑) | OB 1 (↓) | OB 10 (↓) | OB 100 (↓) |
|---|---|---|---|---|---|---|---|
| DRD2 | Augmented Memory | $22 \pm 7$ | $0.774 \pm 0.019$ | $22 \pm 7$ | $143 \pm 75(10)$ | $733 \pm 120(10)$ | Failed |
| | **Saturn** | $369 \pm 62$ | $0.671 \pm 0.050$ | $310 \pm 70$ | $93 \pm 53(10)$ | $391 \pm 56(10)$ | $663 \pm 55(10)$ |
| | Saturn-GA | $209 \pm 55$ | $0.745 \pm 0.041$ | $189 \pm 57$ | $96 \pm 56(10)$ | $403 \pm 75(10)$ | $806 \pm 84(10)$ |
| AChE | Augmented Memory | $173 \pm 19$ | $0.843 \pm 0.009$ | $170 \pm 18$ | $57 \pm 2(10)$ | $189 \pm 52(10)$ | $776 \pm 58(10)$ |
| | **Saturn** | $480 \pm 79$ | $0.757 \pm 0.020$ | $400 \pm 96$ | $32 \pm 24(10)$ | $185 \pm 82(10)$ | $508 \pm 80(10)$ |
| | Saturn-GA | $343 \pm 57$ | $0.809 \pm 0.013$ | $287 \pm 50$ | $32 \pm 25(10)$ | $187 \pm 80(10)$ | $565 \pm 80(10)$ |
| MK2 | Augmented Memory | $0.2 \pm 0.4$ | — | $0.2 \pm 0.4$ | $836 \pm 186(2)$ | Failed | Failed |
| | **Saturn** | $14.9 \pm 14.1$ | $0.454 \pm 0.212$ | $14.1 \pm 13.2$ | $677 \pm 186(9)$ | $861 \pm 108(6)$ | Failed |
| | Saturn-GA | $6.1 \pm 6.5$ | $0.415 \pm 0.202$ | $5.5 \pm 5.5$ | $678 \pm 140(9)$ | $911 \pm 11(2)$ | Failed |

## 4.3 Part 3: Benchmarking Saturn and Demonstrating Enhanced Optimization

In this section, we compare Saturn's performance to previous works, including the state-of-the-art Goal-aware fragment Extraction, Assembly, and Modification (GEAM) proposed by Lee et al. (Lee et al., 2024), which recently reported impressive results on a docking MPO task that considers synthesizability, outperforming baselines by a large margin.

**Experimental Details.** We facilitate an exact comparison with GEAM (Lee et al., 2024) by extracting their oracle code for our experiments, pre-training on the provided ZINC 250k (Sterling & Irwin, 2015) dataset (Appendix F,) and used their MPO objective function (Eq. 5),

$$R(x) = \widehat{DS}(x) \times QED(x) \times \widehat{SA}(x) \in [0, 1], \tag{5}$$

where $\widehat{DS}$ is the normalized QuickVina 2 (Alhossary et al., 2015) docking score and $\widehat{SA}$ is the normalized synthetic accessibility score (Ertl & Schuffenhauer, 2009) (see Appendix F for normalization details). Following GEAM (Lee et al., 2024), docking was performed against 5 targets: **parp1**, **fa7**, **5ht1b**, **braf**, and **jak2**. We ran GEAM and Saturn across 10 seeds (0-9 inclusive) with an oracle budget of 3,000. We note that GEAM's pre-training requires the *labeled* ZINC 250k with all docking values already pre-computed, so there is a large up-front oracle cost. We also emphasize that we *do not tune* Saturn's hyperparameters for this task and the results in this section are out-of-the-box.

**Metrics.** Following Lee et al. (Lee et al., 2023; 2024), we assess the **Hit Ratio (%)** (molecules with a better docking score than the median of known actives, QED > 0.5, SA < 5) and **Novel Hit Ratio (%)** (with the additional constraint of maximum Tanimoto similarity of 0.4 to the training data). We further propose **Strict Hit Ratio (%)** and **Strict Novel Hit Ratio (%)** which filter for the more stringent criteria of QED > 0.7 (based on DrugStore dataset of marketed drugs (Bickerton et al., 2012)) and SA < 3 (based on off-the-shelf catalog molecules (Ertl & Schuffenhauer, 2009)). While drug candidates need not necessarily meet these stricter thresholds, this metric assesses *optimization capability*, which becomes pertinent when jointly optimizing all components is especially crucial. From an optimization perspective, the objective function (Eq. 5) aims to maximize QED and minimize SA and docking score simultaneously. Therefore, achieving high QED and low SA is part of the goal itself. We additionally measure molecular diversity using **IntDiv1** (Polykovskiy et al., 2020) and #**Circles** (Xie et al., 2023) with distance threshold 0.75.

Table 2: Hit Ratio (%). Results are from Lee et al. (Lee et al., 2023) except Augmented Memory, GEAM, datasets, and Saturn which we ran across 10 seeds (0-9 inclusive). The mean and standard deviation are reported. Best results (statistically significant at the 95% confidence level) are bolded.

| Method | Target Protein | | | | |
|---|---|---|---|---|---|
| | parp1 | fa7 | 5ht1b | braf | jak2 |
| **Datasets** | | | | | |
| ZINC 250k (Sterling & Irwin, 2015) | $3.993 \pm 0.355$ | $1.097 \pm 0.192$ | $24.260 \pm 0.622$ | $1.020 \pm 0.193$ | $6.183 \pm 0.344$ |
| ChEMBL 33 (Gaulton et al., 2012) | $6.077 \pm 0.453$ | $1.830 \pm 0.240$ | $24.163 \pm 0.715$ | $2.073 \pm 0.181$ | $9.013 \pm 0.562$ |
| **Generative Models** | | | | | |
| REINVENT (Olivecrona et al., 2017) | $4.693 \pm 1.776$ | $1.967 \pm 0.661$ | $26.047 \pm 2.497$ | $2.207 \pm 0.800$ | $5.667 \pm 1.067$ |
| JT-VAE (Jin et al., 2018) | $3.200 \pm 0.348$ | $0.933 \pm 0.152$ | $18.044 \pm 0.747$ | $0.644 \pm 0.157$ | $5.856 \pm 0.204$ |
| GraphAF (Shi et al., 2020) | $0.822 \pm 0.113$ | $0.011 \pm 0.016$ | $6.978 \pm 0.952$ | $1.422 \pm 0.556$ | $1.233 \pm 0.284$ |
| MORLD (Jeon & Kim, 2020) | $0.047 \pm 0.050$ | $0.007 \pm 0.013$ | $0.893 \pm 0.758$ | $0.047 \pm 0.040$ | $0.227 \pm 0.118$ |
| HierVAE (Jin et al., 2020a) | $1.180 \pm 0.182$ | $0.033 \pm 0.030$ | $0.740 \pm 0.371$ | $0.367 \pm 0.187$ | $0.487 \pm 0.183$ |
| GraphDF (Luo et al., 2021) | $0.044 \pm 0.031$ | $0.000 \pm 0.000$ | $0.000 \pm 0.000$ | $0.011 \pm 0.016$ | $0.011 \pm 0.016$ |
| FREED (Yang et al., 2021) | $4.860 \pm 1.415$ | $1.487 \pm 0.242$ | $14.227 \pm 5.116$ | $2.707 \pm 0.721$ | $6.067 \pm 0.790$ |
| FREED-QS (Yang et al., 2021) | $5.960 \pm 0.902$ | $1.687 \pm 0.177$ | $23.140 \pm 2.422$ | $3.880 \pm 0.623$ | $7.653 \pm 1.373$ |
| LIMO (Eckmann et al., 2022) | $0.456 \pm 0.057$ | $0.044 \pm 0.016$ | $1.200 \pm 0.178$ | $0.278 \pm 0.134$ | $0.711 \pm 0.329$ |
| GDSS (Jo et al., 2022) | $2.367 \pm 0.316$ | $0.467 \pm 0.112$ | $6.267 \pm 0.287$ | $0.300 \pm 0.198$ | $1.367 \pm 0.258$ |
| MOOD (Lee et al., 2023) | $7.260 \pm 0.764$ | $0.787 \pm 0.128$ | $21.427 \pm 0.502$ | $5.913 \pm 0.311$ | $10.367 \pm 0.616$ |
| Aug. Mem. (Guo & Schwaller, 2024a) | $16.966 \pm 3.224$ | $2.637 \pm 0.860$ | $52.016 \pm 2.302$ | $8.307 \pm 1.714$ | $21.548 \pm 4.938$ |
| GEAM (Lee et al., 2024) | $\mathbf{45.158 \pm 2.408}$ | $\mathbf{20.552 \pm 2.357}$ | $47.664 \pm 1.198$ | $\mathbf{30.444 \pm 1.610}$ | $46.129 \pm 2.073$ |
| Saturn (ours) | $\mathbf{57.981 \pm 18.537}$ | $14.527 \pm 9.961$ | $\mathbf{68.185 \pm 3.400}$ | $38.999 \pm 10.114$ | $\mathbf{60.827 \pm 11.502}$ |

Table 3: Novel Hit Ratio (%). Results are from Lee et al. (Lee et al., 2024) except GEAM and Saturn which we ran across 10 seeds (0-9 inclusive). The mean and standard deviation are reported. Best results (statistically significant at the 95% confidence level) are bolded.

| Method | Target Protein | | | | |
|---|---|---|---|---|---|
| | parp1 | fa7 | 5ht1b | braf | jak2 |
| REINVENT (Olivecrona et al., 2017) | $0.480 \pm 0.344$ | $0.213 \pm 0.081$ | $2.453 \pm 0.561$ | $0.127 \pm 0.088$ | $0.613 \pm 0.167$ |
| GCPN (You et al., 2018) | $0.056 \pm 0.016$ | $0.444 \pm 0.333$ | $0.444 \pm 0.150$ | $0.033 \pm 0.027$ | $0.256 \pm 0.087$ |
| JT-VAE (Jin et al., 2018) | $0.856 \pm 0.211$ | $0.289 \pm 0.016$ | $4.656 \pm 1.406$ | $0.144 \pm 0.068$ | $0.815 \pm 0.044$ |
| GraphAF (Shi et al., 2020) | $0.689 \pm 0.166$ | $0.011 \pm 0.016$ | $3.178 \pm 0.393$ | $0.956 \pm 0.319$ | $0.767 \pm 0.098$ |
| GraphGA (Jensen, 2019) | $4.811 \pm 1.661$ | $0.422 \pm 0.193$ | $7.011 \pm 2.732$ | $3.767 \pm 1.498$ | $5.311 \pm 1.667$ |
| MORLD (Jeon & Kim, 2020) | $0.047 \pm 0.050$ | $0.007 \pm 0.013$ | $0.880 \pm 0.735$ | $0.047 \pm 0.040$ | $0.227 \pm 0.118$ |
| HierVAE (Jin et al., 2020a) | $0.553 \pm 0.214$ | $0.007 \pm 0.013$ | $0.507 \pm 0.278$ | $0.207 \pm 0.220$ | $0.227 \pm 0.127$ |
| RationaleRL (Jin et al., 2020b) | $4.267 \pm 0.450$ | $0.900 \pm 0.098$ | $2.967 \pm 0.307$ | $0.000 \pm 0.000$ | $2.967 \pm 0.196$ |
| GA+D (Nigam et al., 2020) | $0.044 \pm 0.042$ | $0.011 \pm 0.016$ | $1.544 \pm 0.273$ | $0.800 \pm 0.864$ | $0.756 \pm 0.204$ |
| MARS (Xie et al., 2021) | $1.178 \pm 0.299$ | $0.367 \pm 0.072$ | $6.833 \pm 0.706$ | $0.478 \pm 0.083$ | $2.178 \pm 0.545$ |
| GEGL (Ahn et al., 2020) | $0.789 \pm 0.150$ | $0.256 \pm 0.083$ | $3.167 \pm 0.260$ | $0.244 \pm 0.016$ | $0.933 \pm 0.072$ |
| GraphDF (Luo et al., 2021) | $0.044 \pm 0.031$ | $0.000 \pm 0.000$ | $0.000 \pm 0.000$ | $0.011 \pm 0.016$ | $0.011 \pm 0.016$ |
| FREED (Yang et al., 2021) | $4.627 \pm 0.727$ | $1.332 \pm 0.113$ | $16.767 \pm 0.897$ | $2.940 \pm 0.359$ | $5.800 \pm 0.295$ |
| LIMO (Eckmann et al., 2022) | $0.455 \pm 0.057$ | $0.044 \pm 0.016$ | $1.189 \pm 0.181$ | $0.278 \pm 0.134$ | $0.689 \pm 0.319$ |
| GDSS (Jo et al., 2022) | $1.933 \pm 0.208$ | $0.368 \pm 0.103$ | $4.667 \pm 0.306$ | $0.167 \pm 0.134$ | $1.167 \pm 0.281$ |
| PS-VAE (Kong et al., 2022) | $1.644 \pm 0.389$ | $0.478 \pm 0.140$ | $12.622 \pm 1.437$ | $0.367 \pm 0.047$ | $4.178 \pm 0.933$ |
| MOOD (Lee et al., 2023) | $7.017 \pm 0.428$ | $0.733 \pm 0.141$ | $18.673 \pm 0.423$ | $5.240 \pm 0.285$ | $9.200 \pm 0.524$ |
| GEAM (Lee et al., 2024) | $39.159 \pm 2.790$ | $\mathbf{19.540 \pm 2.347}$ | $40.123 \pm 1.611$ | $\mathbf{27.467 \pm 1.374}$ | $41.765 \pm 3.412$ |
| Saturn (ours) | $3.839 \pm 3.316$ | $0.470 \pm 0.272$ | $5.731 \pm 6.166$ | $3.652 \pm 3.777$ | $6.129 \pm 5.449$ |
| Saturn-Tanimoto (ours) | $\mathbf{50.552 \pm 9.530}$ | $\mathbf{20.181 \pm 5.598}$ | $\mathbf{54.260 \pm 6.722}$ | $19.820 \pm 10.120$ | $\mathbf{47.785 \pm 14.041}$ |

**Saturn and GEAM Outperform all Baselines.** We evaluate the Hit Ratio and include random sampling of 3,000 molecules from ZINC 250k (Sterling & Irwin, 2015) and ChEMBL 33 (Gaulton et al., 2012) as baselines (Table 2). The results show that only Augmented Memory (Guo & Schwaller, 2024a), GEAM (Lee et al., 2024), and Saturn outperform these baselines, with GEAM and Saturn displaying similar performance. However, Saturn exhibits higher variance, likely due to the small batch size (16) used to approximate the expected reward (Eq. 2). For the Novel Hit Ratio (Table 3), Saturn performs worse than GEAM. However, this is expected since the Mamba backbone excels at maximum likelihood estimation and fits the ZINC 250k training distribution well (Appendix F.1). It is then unsurprising that generated molecules are not particularly dissimilar to ZINC. We highlight that this 0.4 threshold is arbitrary and that modeling distributions *well* is the fundamental *goal* of generative models. However, to demonstrate how to satisfy this "Novel" metric, we divide the task into two phases, akin to curriculum learning (Guo et al., 2022). Firstly, we task the base Saturn model to generate molecules with high Tanimoto *dissimilarity* (this is the only optimization objective) to the training data. We run this process for 1,500 oracle calls (see Appendix F.4 for more details). This new model checkpoint (Saturn-Tanimoto) now generates molecules that are dissimilar to ZINC 250k and is the starting point for GEAM's MPO task. Table 3 shows that performance immediately recovers and matches GEAM. We believe this is still a fair assessment as computing Tanimoto similarity is cheap (this process took minutes) and also shows the flexibility of Saturn.

Table 4: Strict Hit Ratio (%). GEAM and Saturn results are across 10 seeds (0-9 inclusive). OB is Oracle Burden (oracle calls required to generate $N$ unique molecules). The number in parentheses in the OB statistics represents how many runs out of 10 were successful. The mean and standard deviation are reported. Best results (statistically significant at the 95% confidence level) are bolded.

| Method | Target Protein | | | | |
|---|---|---|---|---|---|
| | parp1 | fa7 | 5ht1b | braf | jak2 |
| **GEAM** (Lee et al., 2024) | | | | | |
| Strict Hit Ratio ($\uparrow$) | $6.510 \pm 1.087$ | $2.106 \pm 0.958$ | $8.719 \pm 0.903$ | $3.685 \pm 0.524$ | $7.944 \pm 1.157$ |
| OB (1) ($\downarrow$) | $250 \pm 157(10)$ | $433 \pm 209(10)$ | $114 \pm 112(10)$ | $355 \pm 96(10)$ | $230 \pm 117(10)$ |
| OB (10) ($\downarrow$) | $743 \pm 52(10)$ | $1446 \pm 404(10)$ | $531 \pm 38(10)$ | $892 \pm 144(10)$ | $537 \pm 70(10)$ |
| OB (100) ($\downarrow$) | $2106 \pm 202(10)$ | $2927 \pm 0(1)$ | $1527 \pm 110(10)$ | $2674 \pm 163(6)$ | $1606 \pm 218(10)$ |
| **IntDiv1** ($\uparrow$) | $0.766 \pm 0.017$ | $0.709 \pm 0.043$ | $0.799 \pm 0.017$ | $0.751 \pm 0.023$ | $0.763 \pm 0.021$ |
| **#Circles** ($\uparrow$) | $14 \pm 3$ | $7 \pm 2$ | $25 \pm 3$ | $11 \pm 2$ | $18 \pm 2$ |
| **Saturn (ours)** | | | | | |
| **Strict Hit Ratio** | $55.102 \pm 18.027$ | $13.887 \pm 9.723$ | $64.730 \pm 3.717$ | $37.250 \pm 9.615$ | $55.903 \pm 13.613$ |
| **OB (1)** ($\downarrow$) | $139 \pm 96(10)$ | $352 \pm 206(10)$ | $21 \pm 7(10)$ | $291 \pm 143(10)$ | $88 \pm 56(10)$ |
| **OB (10)** ($\downarrow$) | $518 \pm 92(10)$ | $924 \pm 247(10)$ | $105 \pm 23(10)$ | $581 \pm 123(10)$ | $348 \pm 96(10)$ |
| **OB (100)** ($\downarrow$) | $956 \pm 259(10)$ | $1776 \pm 551(10)$ | $441 \pm 44(10)$ | $1057 \pm 187(10)$ | $785 \pm 191(10)$ |
| IntDiv1 ($\uparrow$) | $0.596 \pm 0.049$ | $0.592 \pm 0.066$ | $0.685 \pm 0.021$ | $0.597 \pm 0.042$ | $0.638 \pm 0.034$ |
| #Circles ($\uparrow$) | $5 \pm 0$ | $3 \pm 1$ | $17 \pm 3$ | $4 \pm 0$ | $7 \pm 1$ |

**Saturn: Enhanced MPO.** Due to the superior performance of Saturn and GEAM, we further investigate their optimization capability by applying a strict filter for QED > 0.7 and SA < 3 (Table 4). The results show that GEAM's Hit Ratios drop drastically while Saturn's remain relatively unchanged, which demonstrates that Saturn optimizes the MPO objective to a much greater degree (see Appendix F for *Novel* Strict Filter results). Importantly, Saturn finds molecules passing this strict filter with much fewer oracle calls (OB metrics in Table 4), trading off diversity to do so. For **fa7** and **braf**, GEAM does not find 100 molecules passing the strict filter in 9/10 and 4/10 replicates, respectively, while Saturn is successful in 10/10 for both (Table 4). Finding desirable molecules under minimal oracle calls is practically relevant when moving to computationally expensive high-fidelity oracles, so as to identify a small set of *excellent* candidates satisfying the MPO objective.

## 5 CONCLUSION

In this work, we present **Saturn**, a framework for sample-efficient *de novo* molecular design using memory manipulation. We demonstrate the first application of the Mamba (Gu & Dao, 2023) architecture for generative molecular design with reinforcement learning and show how it synergistically leverages SMILES augmentation and experience replay for enhanced sample efficiency. Through systematic study, we elucidate the mechanism of Augmented Memory (original work only showed its empirical benefits) and show it squeezes sequence generation likelihoods such that it becomes increasingly likely to generate *some* SMILES representation of the replay buffer molecular graphs. Next, we show *how* Mamba leverages this mechanism to improve sample efficiency through "hop-and-locally-explore" behavior. With the optimal architecture and hyperparameters identified for sample efficiency in a test experiment, we apply Saturn on two sets of MPO tasks relevant to drug discovery, outperforming all baseline models and the recent GEAM (Lee et al., 2024) model which, when released, outperformed all baselines by a large margin. Compared to GEAM, we further show that Saturn achieves superior MPO, finding desirable molecules faster with fewer oracle calls, albeit with a trade-off in diversity. Our work opens up the prospect of *directly* optimizing expensive high-fidelity oracles (beyond docking), which are more correlated with relevant drug discovery end-points. Recent work has applied multi-fidelity learning (Eckmann et al., 2024) or active learning (Loeffler et al., 2024a; Dodds et al., 2024) to enable on-the-fly update of a surrogate model to predict such oracle evaluations for generative design. These workflows can be applied directly with Saturn, but importantly, we may be *sufficiently efficient* to directly optimize these oracles, mitigating surrogate out-of-domain concerns. Moreover, it is straightforward to augment Saturn with known strategies to improve sample efficiency, such as curriculum learning (Guo et al., 2022) as we have shown in Part 3. Correspondingly, future work will stress-test Saturn on high-fidelity oracles and interrogate the prospect of directly optimizing QM/MM and free energy (Wang et al., 2019; Moore et al., 2022; 2023; Crivelli-Decker et al., 2024) protocols with modest computational resources.

## 6 REPRODUCIBILITY STATEMENT

The code is provided in the figshare link in the Abstract and also provided here: `https://figshare.com/s/21059896530e222b9cd5`. The repository contains a README along with prepared files to reproduce all experiments.

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

## A    APPENDIX

The Appendix contains full details on Saturn, grid-search results, ablation studies, algorithmic details, and supplementary results for additional experiments including architecture scaling studies. The code is available at `https://figshare.com/s/21059896530e222b9cd5`.

## B    WHAT IS SATURN?

Saturn is a language-based generative molecular design framework which features minimal implementations of Augmented Memory (Guo & Schwaller, 2024a) and Beam Enumeration (Guo & Schwaller, 2024b). These two methods were first implemented here: `https://github.com/schwallergroup/augmented_memory`, which in turn was built on REINVENT version 3.2 (Olivecrona et al., 2017; Blaschke et al., 2020a): `https://github.com/MolecularAI/Reinvent`. REINVENT is still under active development and version 4 (Loeffler et al., 2024b) was recently released, supporting a wide range of generative tasks including small molecule design (Olivecrona et al., 2017; Blaschke et al., 2020a), library design (Fialková et al., 2021), linker design

(Guo et al., 2023), proposing small modifications (He et al., 2021), and sampling nearest neighbors (Tibo et al., 2023).

Saturn (at the moment) focuses only on generative small molecule design and **research development is on sample efficiency**. It is a much smaller code-base than REINVENT 4 and with focus on minimal implementation. That being said, the key new additions to Saturn include: extending small molecule generative architecture from just RNN in REINVENT to Decoder transformer (Vaswani et al., 2017; Radford et al., 2019) and Mamba (Gu & Dao, 2023). Secondly, allowing oracle caching to track repeated generations and allow pre-screening specified oracles (in an MPO objective, some oracle components may be computationally inexpensive and it would be practical to first screen a molecules through these oracles before any expensive components). Thirdly, implementation of a genetic algorithm which couples GraphGA (Jensen, 2019) on the replay buffer such that new molecules can be generated from the replay buffer parent sequences. In the ensuing subsections, we describe in detail these key new additions.

## B.1 GENERATIVE ARCHITECTURE

Many initial language-based molecular generative models were RNN-based (Olivecrona et al., 2017; Segler et al., 2018; Neil et al., 2018; Popova et al., 2018). Early benchmarks (GuacaMol (Brown et al., 2019) and MOSES (Polykovskiy et al., 2020)) assessed whether generated molecules were valid (RDKit parsable), unique, and novel (not in the training data). RNNs satisfy these metrics and can learn distributions well (Flam-Shepherd et al., 2022). More recently, with the prevalence of the transformer (Vaswani et al., 2017; Radford et al., 2019) architecture, many works (Bagal et al., 2021; Wang et al., 2023; Feng et al., 2023; Mazuz et al., 2023; Hu et al., 2024; He et al., 2024; Thomas et al., 2022; Yang et al., 2020) have suggested a replacement of RNNs for generative design. However, many performance assessments only focus on validity, uniqueness, novelty, and optimizing for permissive oracles such as logP, QED (Bickerton et al., 2012) ("drug-likeness"), and the SA score (Ertl & Schuffenhauer, 2009). Some works show that transformers can learn longer SMILES sequences better than RNNs (Feng et al., 2023) (such as natural products). However, often, one actually *wants* to limit sequence length to constrain design to small molecules. Furthermore, recent works have coupled transformers with reinforcement learning (RL) (Feng et al., 2023; Mazuz et al., 2023; Hu et al., 2024; He et al., 2024; Thomas et al., 2022) but the performance is not necessarily better than RNNs. Consequently, it is unclear whether the benefits of transformers are strictly advantageous for small molecule generation.

In this work, we extend Augmented Memory (Guo & Schwaller, 2024a) to Decoder transformer (Vaswani et al., 2017; Radford et al., 2019) and Mamba (Gu & Dao, 2023). Our results show that transformers display similar performance to RNNs for small molecule generation, in agreement with previous literature findings (Thomas et al., 2022). We further demonstrate the first application of Mamba (Gu & Dao, 2023) for goal-directed generation, supplementing recent work investigating S4 models for transfer learning (Özçelik et al., 2024).

## B.2 MAMBA ARCHITECTURE

In Saturn, we empirically find that the Mamba (Gu & Dao, 2023) architecture is the most parameter-efficient in our RL framework when tuning for sample efficiency. Note that in Appendix F.6, we find that scaling up the decoder transformer (25.3M) Vaswani et al. (2017); Radford et al. (2019) to about 5x the size of the Mamba (5.2M) results in similar performance. However, due to less GPU load for smaller models, we chose Mamba as the default architecture in Saturn. Mamba was recently proposed as an alternative to transformers, with linear time training in contrast to the quadratic attention scaling. In decoder transformers, self-attention is the key component:

$$\text{Attention}(Q, K, V) = \text{softmax}\left(\frac{QK^\top}{\sqrt{d_k}}\right) V$$

where embedded input (SMILES (Weininger, 1988) sequences in our case), $X$, is multiplied with the weight matrices, $W_Q$, $W_K$, $W_V$ to produce the Query (Q), Key (K), and Value (V) matrices, respectively. $QK^T$ results in an $N$ x $N$ (sequence length) matrix and leads to an overall $O(N^2)$

scaling. However, an advantage of self-attention is that during inference, the entire context is available without any compression.

We now contrast the information flow in Mamba, with all information adapted from, and following the convention in preceding work on state-space models Gu et al. (2021b), structured S4 (Gu et al., 2021a), the original Mamba (Gu & Dao, 2023) work and a technical blog (Chen, 2024). Mamba builds on state-space models, which propagate information through four learnable matrices: $\mathbf{A}$, $\mathbf{B}$, $\mathbf{C}$, and $\mathbf{D}$:

$$h'(t) = \mathbf{A}h(t) + \mathbf{B}x(t)$$
$$y(t) = \mathbf{C}h(t) + \mathbf{D}x(t)$$

where $h$ is the state, $x$ is the input sequence, and $y$ is the output. Therefore $\mathbf{A}$ and $\mathbf{B}$ dictate how the state changes as a function of the current state and input, respectively. This is similar for $\mathbf{C}$ and $\mathbf{D}$ on the output. As $\mathbf{A}$, $\mathbf{B}$, $\mathbf{C}$, and $\mathbf{D}$ are fixed for all $t$, this is *time-invariant*. As we are working with discrete data (SMILES tokens in our case), the continuous form is discretized (Gu et al., 2021b;a):

$$h_t = \bar{\mathbf{A}}h_{t-1} + \bar{\mathbf{B}}x_t$$
$$y_t = \bar{\mathbf{C}}h_t + \bar{\mathbf{D}}x_t$$

The original works (Gu et al., 2021b;a) show that the model can be trained efficiently using a continuous *convolution* while offering efficient inference through a *recurrent* mode. Note that in some references, $\mathbf{D}$ is omitted and does not transform the input, $x$ (Gu, 2023). In the current formulation which is time-invariant, all states and input are transformed in the same way. However, it would be advantageous to process information differently depending on whether it is *more relevant*, similar to self-attention. Correspondingly, Mamba (Gu & Dao, 2023) extends this framework and removes the time-invariance constraint by making the $\mathbf{B}$, $\mathbf{C}$, and $\mathbf{D}$ matrices dependent on the input (the following notation follows (Chen, 2024)):

$$h_t = \bar{\mathbf{A}}h_{t-1} + \bar{\mathbf{B}}(x_t)x_t$$
$$y_t = \bar{\mathbf{C}}(x_t)h_t + \bar{\mathbf{D}}x_t$$

The input-dependent parameters allow Mamba to *selectively* propagate information. Removing time-invariance prevents efficient training with continuous convolution and the Mamba (Gu & Dao, 2023) authors propose an efficient recurrent *scan* in place. In Saturn, we use their optimized training and inference by adapting the code from the official Mamba repository: `https://github.com/state-spaces/mamba`.

We end this section by conveying that we were not particularly worried about the training and inference speed, since we are working with small molecules with relatively short sequences (typically $< 80$ tokens). The bottleneck is the reward computation, especially if the oracle is expensive. We were interested in studying the optimization dynamics and how efficiently each model can be tuned via RL.

### B.3 ORACLE CACHING

In many reinforcement learning (RL) set-ups, the reward is assumed to be *stationary*, i.e., it does not change on repeat evaluation. This is an assumption that is not always true for physics-based oracles relevant in drug discovery. For example, docking depends on the initial conformer generated, and even more so for molecular dynamics simulations. However, it is reasonable to assume that the reward is *near deterministic* given a reasonably well behaved protein system (in which preliminary studies were made to verify the oracle stability). In effect, the reward for repeat molecules can be retrieved from a cache, thus not imposing additional oracle evaluations. In this work, we show that under this assumption, Saturn can leverage the Mamba (Gu & Dao, 2023) architecture for enhanced sample efficiency. In particular, Mamba displays low uniqueness, but we show this is not detrimental.

As any given molecule can have numerous SMILES representations (via augmentation (Bjerrum, 2017)), it is important to store the *canonical* SMILES in the cache, and also to canonicalize sampled

batches when querying the cache. Canonicalization is simply a pre-defined traversal and can differ depending on the method used. As long as all canonicalization operations are performed with the same method, consistency can be guaranteed. In this work, we use RDKit.

### B.4 GENETIC ALGORITHM

Genetic algorithms (GAs) by themselves can be sample-efficient molecular optimizers (Gao et al., 2022; Jensen, 2019; Tripp & Hernández-Lobato, 2023). Previous work has shown that GAs can improve diversity of the generated set (Liu et al., 2021). Recently, Lee et al. (Lee et al., 2024) proposed Goal-aware fragment Extraction, Assembly, and Modification (GEAM) which combines RL with a GraphGA (Jensen, 2019) and achieves impressive results on generating diverse hits. In Saturn, we implement GraphGA on the replay buffer itself, treating the highest rewarding molecules generated in the entire run so far, as the parent population. Following GEAM (Lee et al., 2024), sampling the parents is done with probability proportion to their corresponding rewards. New molecules from crossover and mutation operations are deposited into the Buffer if they are also high rewarding, essentially *refreshing* the buffer, such that Augmented Memory (Guo & Schwaller, 2024a) can learn from these new SMILES. The motivation was to leverage the GA to counteract decreases in diversity and potentially improve sample efficiency. In the results in the main text and in the following sections, we show that applying the GA does not lead to improved sample efficiency but does indeed recover diversity. We believe that this can be a useful modification to the optimization algorithm in cases where relatively expensive oracles are used and diversity is important due to prevalence of false positives. Concretely, higher-fidelity oracles should in principle model physical behavior more accurately, such that true positives are more common. This can be shown in previous works where using free energy simulations provide better correlations with binding affinity (Eckmann et al., 2024; Crivelli-Decker et al., 2024). In such a case, sample efficiency becomes increasingly important, as the goal is to simply generate molecules satisfying this simulation and lower diversity is not detrimental. However, when using lower-fidelity oracles, more false positives means it is beneficial to have more diverse ideas for downstream triaging. Finally, we note that applying the GA and generating new molecules strictly means they were generated off-policy (in the RL context). Therefore, more meaningful updates to the Agent *may* be achieved with importance sampling (Schlegel et al., 2019), which we did not explore in the current work.

### B.5 FULL ALGORITHM DETAILS AND PSEUDO-CODE

In this section, we derive Saturn's loss function with particular focus on showing its equivalency to maximizing the expected reward. The derivation follows previous works (Olivecrona et al., 2017; Fialková et al., 2021; Guo & Schwaller, 2024a) but with added discussion around implications of the loss function. Specifically, Saturn adapts the Augmented Memory (Guo & Schwaller, 2024a) algorithm which is in turn based on REINVENT (Olivecrona et al., 2017; Blaschke et al., 2020a; Loeffler et al., 2024b). The algorithm itself is reinforcement learning based and can be seen as a modified REINFORCE (Williams, 1992) algorithm. However, while **Saturn (using Mamba with batch size 16 and 10 augmentation rounds)** adapts Augmented Memory, the optimization trajectory is quite different from the original Augmented Memory work due to the "hop-and-locally-explore" sampling behavior. We will focus on highlighting specific points related to this.

**Saturn's Loss Function.** We begin by presenting *how* Saturn generates SMILES (Weininger, 1988), which is the data representation used. SMILES are sequences of alphanumeric characters that can be parsed and mapped to a molecular graph, i.e., a molecule. As SMILES are text-based, it is straightforward to tokenize them, and pre-training Saturn follows next-token prediction. Saturn generates SMILES in an autoregressive manner and thus, SMILES are generated token-by-token from time-step, $t$ to $T$. This can be viewed from a reinforcement learning perspective by defining $S_t$ as the state space representing all intermediate token sequences during molecular generation. $A_t(s_t)$ is the action space which involves sampling a token from a conditional probability distribution, given a token sequence so far, i.e., the current state. Mathematically, the probability of sampling a SMILES, $x$ is given by:

$$P(x) = \prod_{t=1}^{T} \pi_{\theta_{\text{Agent}}} (a_t \mid s_t) \tag{6}$$

Just generating SMILES is often not useful because they should satisfy the target objective. Thus, the base pre-trained model needs to be tuned somehow to achieve this. The end goal is to find a **Policy** (in the reinforcement learning perspective) which dictates with *what* probability SMILES should be generated to optimize an objective function. To this end, we define the **Prior** and the **Agent** which share the same architecture (Mamba) and whose weights are exactly the same at the beginning of a generative experiment. The Prior and Agent are general terms to describe the model states but they both are policies as they both induce a probability of sampling SMILES. However, what is different is that the Prior's weights are frozen so it is *never* updated. By contrast, the Agent *is* updated and is the model that is learning how to generate "good" SMILES. We now discuss how this is achieved. We define the Augmented Likelihood (Olivecrona et al., 2017) of a SMILES, $x$, which is a linear combination between the Prior and a reward term:

$$\log \pi_{\text{Augmented}}(x) = \log \pi_{\text{Prior}}(x) + \sigma R(x) \tag{7}$$

$\log \pi_{\text{Prior}}(x)$ is the log-probability of generating a given SMILES, $x$, under the Prior. Since the Prior's weights are fixed, the probability of sampling a given SMILES *never* changes. Models are typically parameterized by its weights, $\theta$. We take care here and omit $\theta$ because the Prior, as stated previously, is not updated. Next, $R$ is the reward function which defines the target objective, e.g., minimize docking score. Note that the reward function can contain multiple objectives, in which case, constituting a multi-parameter optimization objective. For example, in Experiment 3 of the main text, $R$ is comprised of minimizing docking score, maximizing QED score (Bickerton et al., 2012), and minimizing SA score (Ertl & Schuffenhauer, 2009). $R$ takes as input a SMILES, $x$, and returns a scalar reward $\in [0, 1]$. $\sigma$ is a hyperparameter that scales the contribution of the reward function. Importantly, given a SMILES, $x$, a low $\sigma$ means the Augmented Likelihood converges to the Prior likelihood while a high $\sigma$ means the Augmented Likelihood is dominated by the reward. In this work, $\sigma$ is never changed and is 128 as this was found to work well in the original REINVENT work (Olivecrona et al., 2017).

The loss function is defined as the squared difference between the Augmented Likelihood and the Agent Likelihood:

$$L(\theta) = (\log \pi_{\text{Augmented}}(x) - \log \pi_{\theta_{\text{Agent}}}(x))^2 \tag{8}$$

$\log \pi_{\text{Agent}}(x)$ is the log-probability of generating a given SMILES, $x$, under the Agent. Importantly, we explicitly include $\theta$ here because the Agent *is* updated. We stop here for a moment to discuss the implications of the loss function. The loss function tries to minimize the distance between the Augmented Likelihood and the Agent likelihood. Since the Augmented Likelihood (Eq. 7 is a linear combination of the Prior likelihood and the reward function, if the Agent generates "bad" SMILES, then the reward goes to 0 and the Augmented Likelihood converges to the Prior Likelihood. In this event, the Agent's weights actually regress back towards the Prior. This is because the Prior is pre-trained on a general dataset containing bio-active molecules (such as ChEMBL (Gaulton et al., 2012) and ZINC 250k (Sterling & Irwin, 2015). The implicit assumption during pre-training is that these general datasets might actually already contain "good" molecules. Therefore, in the event that "bad" molecules are generated, the Prior acts as a "fall-back". On the other hand, when the reward is not 0, the Prior still "anchors" the Agent and does not let its weights deviate *too far* from the Prior (this is controlled by $\sigma$). The reason for this is also because the Prior is assumed to potentially already contain "good" molecules. In practice, the Agent can deviate quite far from the Prior (Loeffler et al., 2024b). We now discuss an important implication of this loss function in Saturn. Saturn heavily leverages SMILES augmentation (Bjerrum, 2017) as a data augmentation method to learn from the same molecular graph multiple times. Alternative SMILES sequences, while mapping to the same molecular graph, can have drastically different likelihoods. This is shown in Figure 2 in the main text where Saturn is trained to make it likely to generate all of these alternative SMILES forms. However, this does not always work. Because alternative SMILES forms have different likelihoods, there is the possibility that with the right combination of terms in the Augmented Likelihood, that it equals the Agent likelihood. In this case, the loss contribution is 0 so the Agent actually is not tuned to generate that particular SMILES form with higher likelihood. This is a contributing factor to Saturn's "hop-and-locally-explore" behavior. Given a set of augmented SMILES, if some of these SMILES cancel out in the loss function, then there is a smaller set of augmented SMILES that contribute to the

loss function. With a smaller set, overfitting becomes more prone but we show that this mechanism actually benefits sample efficiency.

Finally, Saturn does not generate individual SMILES but rather, batches of SMILES. Therefore, the loss function is a batched loss:

$$L(\theta) = \frac{1}{|B|} \left[ \sum_{a \in A^*} (\log \pi_{\text{Augmented}} - \log \pi_{\theta_{\text{Agent}}}) \right]^2 \tag{9}$$

The loss magnitude is the mean loss for a given batch, $B$, of sampled SMILES constructed following the actions, $a \in A^*$.

**Minimizing the loss function is equivalent to maximizing the expected reward.** In reinforcement learning, the general objective is to maximize the expected reward. In this section, we show how maximizing the expected reward is equivalent to minimizing the loss function. We first further define some preliminaries: sampling trajectories means sampling SMILES in our context. While there are often *intermediate* rewards during trajectory sampling, e.g., a drone tasked to fly to a target location might receive various rewards for how balanced it is during the flight, we set all intermediate rewards to 0. This is because rewards are only meaningful if the SMILES is a valid molecule. Technically, since the reward is directly the reward from the full trajectory, it is actually the **Return** in reinforcement learning terminology, but we use the term reward to match existing literature. Mathematically, the cost function (in reinforcement learning, $J$ is used and we follow this convention) describes the expected reward when taking actions from a policy that is parameterized by a neural network (Mamba in our case):

$$J(\theta) = \mathbb{E}_{a_t \sim \pi_{\theta_{\text{Agent}}}} \left[ \sum_{t=1}^{T} R(a_t, s_t) \right] \tag{10}$$

Since the expectation is in discrete space (sampling tokens is a discrete action), the cost function can be rewritten by transforming the expectation to a sum:

$$J(\theta) = \sum_{t=1}^{T} \sum_{a \in A_t} R(a_t, s_t) \pi_{\theta_{\text{Agent}}}(a_t|s_t) \tag{11}$$

The double summation is over all time-steps and actions (which token sampled) following the policy, $\pi_\theta$. Since we want to maximize the cost function, we take the derivative:

$$\nabla_\theta J(\theta) = \sum_{t=1}^{T} \sum_{a \in A_t} R(a_t, s_t) \nabla_\theta \pi_{\theta_{\text{Agent}}}(a_t|s_t) \tag{12}$$

Next, the log-derivative trick:

$$\nabla_\theta J(\theta) = \sum_{t=1}^{T} \sum_{a \in A_t} R(a_t, s_t) \pi_{\theta_{\text{Agent}}}(a_t|s_t) \nabla_\theta \log \pi_\theta(a_t|s_t) \tag{13}$$

Using the definition of expectation for discrete space again, the cost function is rewritten:

$$\nabla_\theta J(\theta) = \mathbb{E}_{a_t \sim \pi_{\theta_{\text{Agent}}}} \left[ \sum_{t=1}^{T} R(a_t, s_t) \nabla_\theta \log \pi_{\theta_{\text{Agent}}}(a_t|s_t) \right] \tag{14}$$

Computing the expectation exactly is intractable. This would involve sampling every single SMILES and computing their rewards. Therefore, the expectation is approximated by sampling a batch, $B$, of

SMILES. Next, the set of actions taken in a batch at every time-step, is denoted $A^*$, which yield the specific SMILES generated:

$$\nabla_\theta J(\theta) = \frac{1}{|B|} \left[ \sum_{a \in A^*} R(a_t, s_t) \nabla_\theta \log \pi_{\theta_{\text{Agent}}}(a_t|s_t) \right] \tag{15}$$

The reward, $R$ is defined according to previous works (Olivecrona et al., 2017; Fialková et al., 2021; Guo & Schwaller, 2024a):

$$R(a_t, s_t) = \log \pi_{\text{Augmented}} - \log \pi_{\theta_{\text{Agent}}} \tag{16}$$

Substituting the reward function:

$$\nabla_\theta J(\theta) = \frac{1}{|B|} \left[ \sum_{a \in A^*} \log \pi_{\text{Augmented}} - \log \pi_{\theta_{\text{Agent}}} \right] \sum_{a \in A^*} \nabla_\theta \log \pi_{\theta_{\text{Agent}}}(a_t|s_t) \tag{17}$$

Recalling the loss function:

$$L(\theta) = \frac{1}{|B|} \left[ \sum_{a \in A^*} (\log \pi_{\text{Augmented}} - \log \pi_{\theta_{\text{Agent}}}) \right]^2 \tag{18}$$

Minimizing the loss function requires taking the derivative with respect to $\theta$:

$$\nabla_\theta L(\theta) = -2 \frac{1}{|B|} \left[ \sum_{a \in A^*} \log \pi_{\text{Augmented}} - \log \pi_{\theta_{\text{Agent}}} \right] \sum_{a \in A^*} \nabla_\theta \log \pi_{\theta_{\text{Agent}}} \tag{19}$$

The cost function (Eq. 17) is equivalent to the loss function (Eq. 19) up to a factor.

**Saturn Pseudo-code.**

## C  SATURN: IDENTIFYING OPTIMAL HYPERPARAMETERS AND ARCHITECTURE

In this section, we present results from all hyperparameter investigations for Saturn. In particular, we formulated four questions (each devoted to one subsection) which we answer with empirical results and discussion on the test experiment which has the following multi-parameter optimization (MPO) objective: molecular weight (MW) < 350 Da, number of rings $\geq 2$, and maximize topological polar surface area (tPSA).

**Metrics.** Following Guo et al. (Guo & Schwaller, 2024b), the sample efficiency metrics are **Yield** and **Oracle Burden** (OB). Yield (Eq. 20) is the number of *unique* generated molecules above a reward threshold, $T$.

$$Yield = \sum_{g=1}^{G} \mathbb{I}[R(g) > T] \tag{20}$$

Oracle Burden (Eq. 21) is the number of oracle calls ($c$) required to generate $N$ *unique* molecules above a reward threshold, $T$.

$$Oracle\ Burden = c \mid \sum_{g=1}^{G} \mathbb{I}[R(g) > T] = N \tag{21}$$

**Algorithm 1:** Saturn Goal-directed Generation

**Input:** Oracle Budget $Budget$, Prior $\pi_{\text{Prior}}$, Augmentation Rounds $A$, Reward Function $R$, Sigma $\sigma$, Replay Buffer Size $K$, Genetic Algorithm $GA$

**Output:** Fine-tuned Agent Policy $\pi_{\theta_{\text{Agent}}}$, Generated Set $G$

**Initialization:**

1. Generative Agent $\pi_{\theta_{\text{Agent}}} = \pi_{\text{Prior}}$

2. Diversity Filter $DF$

3. Replay Buffer $RB = \{\}$

4. Oracle Calls $Calls = 0$

5. Oracle Cache $Cache = \{\}$

6. Generated Set $G = \{\}$

**while** $C < Budget$ **do**

Sample batch of SMILES $X = \{x_1, \ldots, x_b\}$ with $x_i \sim \pi_{\theta_{\text{Agent}}}$;

(Optionally) Generate SMILES using the Genetic Algorithm $X_{\text{GA}} = GA(RB)$;

$X = X \cup X_{\text{GA}}$;

**if** $X$ $in$ $Cache$ **then**

Retrieve rewards $R_{\text{Cached}}$

Compute reward for *new* SMILES $R(X_{\text{New}})$;

Update Generated Set tracking $G = G \cup (X_{\text{New}}, R(X_{\text{New}}))$;

Update Oracle Cache $Cache = ((X_{\text{New}}, R_{\text{New}}) \cup Cache)$;

Update Oracle Calls $C = C + |X_{\text{New}}|$;

$R(X) = R_{\text{Cached}} \cup R(X_{\text{New}})$;

Modify rewards based on the Diversity Filter $R(X) = DF(X, R(X))$;

Update Replay Buffer $RB = TopK(X \cup RB)$;

Compute Augmented Likelihood $\log \pi_{\text{Augmented}}(X) = \log \pi_{\text{Prior}}(X) + \sigma R(X)$;

Compute loss $J(\theta) = (\log \pi_{\text{Augmented}} - \log \pi_{\theta_{\text{Agent}}}(X))^2$;

Update the Agent $\pi_{\theta_{\text{Agent}}}$;

Purge Replay Buffer;

**for** $i \leftarrow 1$ **to** $A$ **do**

Augment sampled **and** Replay Buffer SMILES $X_{\text{Augmented}}$;

Compute Augmented Likelihood of augmented SMILES (reward is unchanged) $\log \pi_{\text{Augmented}} = \log \pi_{\text{Prior}}(X_{\text{Augmented}}) + \sigma R(X_{\text{Augmented}})$;

Compute loss $J(\theta)_{\text{Augmented}} = (\log \pi_{\text{Augmented}} - \log \pi_{\theta_{\text{Agent}}}(X_{\text{Augmented}}))^2$;

Update the Agent $\pi_{\theta_{\text{Agent}}}$;

**The Yield and OB metrics are used to assess sample efficiency at the 0.7 reward threshold. In all tables, the number after OB parentheses is the number of successful replicates out of 10. All metrics other than IntDiv1 (Polykovskiy et al., 2020) are rounded to the nearest integer. All individual experiments were run across 10 seeds (0-9 inclusive) and with a 1,000 oracle budget. All experiments were run sequentially on a workstation equipped with an NVIDIA RTX 3090 GPU and AMD Ryzen 9 5900X 12-Core CPU.**

### C.1  DATA PRE-PROCESSING AND PRE-TRAINING

Before presenting grid-search results, we first describe the full data pre-processing pipeline and design decisions made. The pre-training data for all experiments except **Part 3: Benchmarking Physics-based MPO Objective** in the main text (ZINC 250k (Sterling & Irwin, 2015) instead), was ChEMBL 33 (Gaulton et al., 2012). We first downloaded the raw ChEMBL 33 from: `https://ftp.ebi.ac.uk/pub/databases/chembl/ChEMBLdb/releases/chembl_33/`. There was no particular reason version 33 was chosen, other than it was the latest version at the time of experiments. We note that very recently (March 2024), version 34 was released.

The exact pre-processing steps along with the SMILES remaining after each step are:

1. Raw ChEMBL 33 - 2,372,674

2. Standardization (charge and isotope handling) based on `https://github.com/MolecularAI/ReinventCommunity/blob/master/notebooks/Data_Preparation.ipynb`. All SMILES that could not be parsed by RDKit were removed - 2,312,459

3. Kept only the unique SMILES - 2,203,884

4. Tokenize all SMILES based on REINVENT's tokenizer: `https://github.com/MolecularAI/reinvent-models/blob/main/reinvent_models/reinvent_core/models/vocabulary.py`

5. Keep SMILES $\leq$ 80 tokens - 2,065,099

6. $150 \leq$ molecular weight $\leq 600$ - 2,016,970

7. Number of heavy atoms $\leq 40$ - 1,975,282

8. Number of rings $\leq 8$ - 1,974,522

9. Size of largest ring $\leq 8$ - 1,961,690

10. Longest aliphatic carbon chain $\leq 5$ - 1,950,213

11. Removed SMILES containing the following tokens (due to undesired chemistry and low token frequency): [S+], [C-], [s+], [O], [S@+], [S@@+], [S-], [o+], [NH+], [n-], [N@], [N@@], [N@+], [N@@+], [S@@], [C+], [S@], [c+], [NH2+], [SH], [NH-], [cH-], [O+], [c-], [CH], [SH+], [CH2-], [OH+], [nH+], [SH2] - **1,942,081**

The final vocabulary contained 37 tokens (2 extra tokens were added, indicating <START> and <END>). We note that stereochemistry tokens were kept (this is not the case for REINVENT (Blaschke et al., 2020a)).

In this work, we investigated LSTM (Hochreiter & Schmidhuber, 1997) RNN, Decoder transformer (Vaswani et al., 2017; Radford et al., 2019), and Mamba (Gu & Dao, 2023). Given a vocabulary of 37, the model parameters were as follows:

1. RNN: 5,807,909 (based on REINVENT (Blaschke et al., 2020a))

2. Decoder: 6,337,061 (based on recent work (Hu et al., 2024) that applied this model size and used a similar loss function to REINVENT)

3. Mamba: 5,265,920 (based on similar size to RNN)

The exact hyperparameters of each architecture are the default arguments in the codebase. Each training step consisted of a full pass through the dataset. The key pre-training parameters were:

1. Max training steps = 20

2. Seed = 0

3. Batch size = 512

4. Learning rate = 0.0001

5. Randomize (Bjerrum, 2017) every batch of SMILES

The following model checkpoints were used:

1. RNN: Epoch 18, NLL = 34.61, Validity (10k) = 94.48%

2. Decoder: Epoch 20, NLL = 33.38, Validity (10k) = 96.04%

3. Mamba: Epoch 18, NLL = 32.21, Validity (10k) = 95.60%

### C.2 UNDERSTANDING THE LIMITS OF AUGMENTED MEMORY

Augmented Memory (Guo & Schwaller, 2024a) improves sample efficiency by repeated learning on the high reward SMILES stored in the replay buffer (referred to as Buffer from here on). For completeness, we first describe *how* repeated learning can be achieved via data augmentation. SMILES are string representations resulting from performing a depth-first search (DFS) on a molecular graph (as is done in RDKit). Depending on the starting node (atom in the molecule), a different SMILES representation results from DFS. In Augmented Memory, SMILES augmentation is performed by shuffling the atom order and yielding different SMILES representations of the *same* molecular graph. This is useful as data augmentation because all of these SMILES representations map to the same molecule, yet their sequence likelihoods are different. Since they map to the same molecule, the same reward can be assigned to all augmented SMILES.

In the original work, ablation experiments showed that updating the Agent with *only* the Buffer resulted in minimal difference. This suggests that a viable way to exploiting the gains from Augmented Memory is to simply have *new* examples of high reward SMILES being added to the Buffer. In the original work, the number of augmentation rounds was capped at two to mitigate mode collapse. In this work, we assume *near deterministic* rewards and use caching to handle repeated generations. Under this assumption, our hypothesis in this subsection is: as long as unique high reward SMILES are generated, increasing augmentation rounds can further improve sample efficiency. Correspondingly, we perform a grid search using Augmented Memory's default generator architecture (LSTM (Hochreiter & Schmidhuber, 1997) RNN) and vary the batch size (64, 32, 16, 8) and augmentation rounds (0-20 inclusive except 1) where 0 augmentation rounds is equivalent to REINVENT (Olivecrona et al., 2017; Blaschke et al., 2020a). The results are shown in Tables 5, 6, 7, and 8.

**Increasing augmentation rounds:**

1. Decreases diversity, as expected.

2. Increases the number of repeated SMILES.

**Decreasing batch size:**

1. Monotonically improves sample efficiency (though not always significant at the 95% confidence level).

2. Benefits Augmented memory more than REINVENT (0 augmentation rounds).

3. Increases the number of repeated SMILES.

4. Increases variance, as expected (since the expected reward is being approximated with a smaller batch size so it is more noisy).

5. Decreases diversity.

**Taking these observations together, increasing augmentation rounds and decreasing batch size *can* trade-off diversity for sample efficiency (inconsistently and with higher variance).**

Table 5: RNN batch size 64.

| Model | Aug. Rounds | Yield | IntDiv1 | Scaffolds | OB 1 | OB 10 | OB 100 | Repeats |
|-------|-------------|-------|---------|-----------|------|-------|--------|---------|
| RNN | 0 | 0±0 | — | 0±0 | 584±251 (5) | Failed (0) | Failed (0) | 1±1 |
| RNN | 2 | 15±9 | 0.775±0.073 | 15±9 | 644±173 (10) | 941±58 (8) | Failed (0) | 0±0 |
| RNN | 3 | 33±42 | 0.788±0.043 | 32±40 | 613±96 (10) | 927±128 (9) | 993±0 (1) | 0±0 |
| RNN | 4 | 32±16 | 0.813±0.024 | 31±16 | 527±198 (10) | 880±90 (10) | Failed (0) | 0±0 |
| RNN | 5 | 40±14 | 0.812±0.023 | 39±13 | 459±177 (10) | 862±68 (10) | Failed (0) | 0±0 |
| RNN | 6 | 41±32 | 0.805±0.032 | 39±28 | 492±184 (10) | 852±99 (9) | 1041±0 (1) | 0±0 |
| RNN | 7 | 47±25 | 0.814±0.019 | 46±24 | 543±188 (10) | 842±93 (10) | 1055±0 (1) | 0±0 |
| RNN | 8 | 28±16 | 0.801±0.032 | 27±16 | 557±173 (10) | 912±82 (9) | Failed (0) | 0±0 |
| RNN | 9 | 21±13 | 0.742±0.124 | 21±13 | 596±215 (10) | 918±61 (8) | Failed (0) | 1±2 |
| RNN | 10 | 27±18 | 0.796±0.046 | 27±18 | 511±266 (10) | 859±65 (8) | Failed (0) | 0±0 |
| RNN | 11 | 20±14 | 0.749±0.115 | 20±14 | 611±235 (10) | 938±85 (8) | Failed (0) | 1±2 |
| RNN | 12 | 48±18 | 0.813±0.022 | 46±18 | 468±206 (10) | 851±55 (10) | Failed (0) | 1±1 |
| RNN | 13 | 57±43 | 0.808±0.027 | 54±39 | 446±213 (10) | 822±144 (10) | 952±0 (1) | 1±2 |
| RNN | 14 | 33±13 | 0.801±0.024 | 32±13 | 587±175 (10) | 884±79 (10) | Failed (0) | 1±1 |
| RNN | 15 | 47±32 | 0.797±0.037 | 46±32 | 532±196 (10) | 836±122 (10) | 1052±0 (1) | 2±2 |
| RNN | 16 | 34±32 | 0.783±0.026 | 33±30 | 647±208 (10) | 918±97 (10) | 1034±0 (1) | 3±4 |
| RNN | 17 | 31±29 | 0.769±0.06 | 30±29 | 645±176 (10) | 870±99 (7) | Failed (0) | 3±4 |
| RNN | 18 | 35±28 | 0.774±0.035 | 32±24 | 673±125 (10) | 898±88 (8) | 1053±0 (1) | 7±5 |
| RNN | 19 | 43±41 | 0.781±0.034 | 40±36 | 659±183 (10) | 875±111 (8) | 949±0 (1) | 7±9 |
| RNN | 20 | 51±29 | 0.792±0.03 | 48±28 | 583±187 (10) | 837±133 (10) | 1056±0 (1) | 3±2 |

Table 6: RNN batch size 32.

| Model | Aug. Rounds | Yield | IntDiv1 | Scaffolds | OB 1 | OB 10 | OB 100 | Repeats |
|-------|-------------|-------|---------|-----------|------|-------|--------|---------|
| RNN | 0 | 0±0 | — | 0±0 | 798±101 (5) | Failed (0) | Failed (0) | 1±1 |
| RNN | 2 | 43±25 | 0.825±0.029 | 42±24 | 608±151 (10) | 844±90 (9) | Failed (0) | 0±0 |
| RNN | 3 | 52±34 | 0.810±0.059 | 51±32 | 522±141 (10) | 789±100 (9) | 1018±0 (2) | 0±1 |
| RNN | 4 | 87±33 | 0.820±0.018 | 83±31 | 466±120 (10) | 740±77 (10) | 987±30 (4) | 1±3 |
| RNN | 5 | 98±57 | 0.817±0.027 | 89±50 | 408±184 (10) | 714±136 (10) | 915±20 (4) | 1±2 |
| RNN | 6 | 76±50 | 0.808±0.028 | 71±43 | 476±159 (10) | 783±99 (10) | 927±30 (2) | 1±3 |
| RNN | 7 | 78±40 | 0.805±0.027 | 72±40 | 478±90 (10) | 760±70 (10) | 942±26 (2) | 3±7 |
| RNN | 8 | 89±72 | 0.798±0.036 | 78±58 | 529±165 (10) | 767±146 (10) | 899±48 (3) | 9±13 |
| RNN | 9 | 57±52 | 0.781±0.046 | 50±42 | 608±186 (10) | 811±143 (9) | 977±36 (3) | 5±4 |
| RNN | 10 | 90±65 | 0.788±0.031 | 82±55 | 549±158 (10) | 769±142 (10) | 977±66 (5) | 9±14 |
| RNN | 11 | 60±43 | 0.755±0.105 | 57±43 | 593±207 (10) | 781±83 (8) | 969±52 (2) | 2±2 |
| RNN | 12 | 103±83 | 0.790±0.021 | 90±72 | 534±168 (10) | 763±158 (10) | 930±105 (4) | 10±23 |
| RNN | 13 | 72±57 | 0.749±0.065 | 62±52 | 578±155 (10) | 765±134 (8) | 958±54 (3) | 12±9 |
| RNN | 14 | 95±55 | 0.779±0.027 | 83±47 | 463±173 (10) | 758±110 (10) | 964±28 (5) | 16±15 |
| RNN | 15 | 74±60 | 0.784±0.036 | 66±52 | 554±92 (10) | 820±124 (10) | 963±54 (4) | 22±20 |
| RNN | 16 | 84±60 | 0.758±0.07 | 70±44 | 544±209 (10) | 768±105 (9) | 957±42 (5) | 17±19 |
| RNN | 17 | 112±74 | 0.765±0.067 | 96±56 | 474±131 (10) | 729±105 (10) | 908±96 (4) | 21±21 |
| RNN | 18 | 77±49 | 0.774±0.039 | 67±43 | 533±100 (10) | 779±102 (10) | 927±12 (2) | 35±32 |
| RNN | 19 | 84±56 | 0.749±0.037 | 68±50 | 535±181 (10) | 788±127 (10) | 951±61 (3) | 33±44 |
| RNN | 20 | 76±77 | 0.717±0.094 | 64±61 | 653±200 (10) | 810±121 (9) | 919±76 (3) | 56±64 |

Table 7: RNN batch size 16.

| Model | Aug. Rounds | Yield | IntDiv1 | Scaffolds | OB 1 | OB 10 | OB 100 | Repeats |
|-------|-------------|-------|---------|-----------|------|-------|--------|---------|
| RNN | 0 | 8±9 | 0.700±0.126 | 8±9 | 546±263 (8) | 837±144 (3) | Failed (0) | 1±1 |
| RNN | 2 | 86±40 | 0.819±0.026 | 82±38 | 409±158 (10) | 709±86 (10) | 907±14 (2) | 2±4 |
| RNN | 3 | 103±47 | 0.831±0.027 | 100±44 | 406±157 (10) | 706±98 (10) | 942±45 (5) | 2±3 |
| RNN | 4 | 90±62 | 0.828±0.017 | 83±53 | 440±152 (10) | 741±102 (10) | 916±76 (3) | 1±1 |
| RNN | 5 | 107±58 | 0.814±0.036 | 101±54 | 480±118 (10) | 721±109 (10) | 916±53 (4) | 7±7 |
| RNN | 6 | 121±80 | 0.791±0.040 | 107±68 | 493±214 (10) | 713±156 (10) | 895±107 (5) | 12±11 |
| RNN | 7 | 144±107 | 0.776±0.026 | 117±86 | 467±186 (10) | 684±136 (10) | 871±116 (6) | 38±82 |
| RNN | 8 | 120±95 | 0.734±0.128 | 104±85 | 481±288 (10) | 653±145 (8) | 854±54 (5) | 18±28 |
| RNN | 9 | 141±104 | 0.783±0.048 | 112±72 | 453±211 (10) | 654±154 (9) | 871±104 (6) | 59±95 |
| RNN | 10 | 106±76 | 0.760±0.0560 | 84±63 | 510±201 (10) | 733±122 (9) | 913±64 (5) | 43±47 |
| RNN | 11 | 120±105 | 0.764±0.032 | 95±81 | 500±220 (10) | 741±199 (10) | 829±99 (4) | 42±37 |
| RNN | 12 | 171±140 | 0.769±0.028 | 124±109 | 389±209 (10) | 662±186 (10) | 774±128 (5) | 39±30 |
| RNN | 13 | 133±106 | 0.767±0.038 | 106±93 | 510±186 (10) | 690±162 (10) | 826±131 (4) | 83±88 |
| RNN | 14 | 166±130 | 0.769±0.045 | 129±93 | 413±237 (10) | 659±195 (10) | 777±94 (5) | 93±69 |
| RNN | 15 | 154±89 | 0.732±0.064 | 127±78 | 504±162 (10) | 647±124 (9) | 861±59 (7) | 94±75 |
| RNN | 16 | 156±155 | 0.716±0.094 | 109±109 | 517±196 (10) | 682±202 (9) | 838±182 (6) | 143±120 |
| RNN | 17 | 141±82 | 0.737±0.059 | 98±49 | 444±181 (10) | 696±128 (10) | 894±71 (7) | 198±163 |
| RNN | 18 | 189±136 | 0.727±0.044 | 152±119 | 469±212 (10) | 657±174 (10) | 832±141 (7) | 247±210 |
| RNN | 19 | 162±121 | 0.654±0.165 | 119±98 | 507±257 (10) | 625±137 (8) | 836±109 (7) | 210±128 |
| RNN | 20 | 139±110 | 0.732±0.045 | 91±67 | 492±188 (10) | 720±157 (10) | 847±110 (5) | 262±179 |

Table 8: RNN batch size 8.

| Model | Aug. Rounds | Yield | IntDiv1 | Scaffolds | OB 1 | OB 10 | OB 100 | Repeats |
|-------|-------------|-------|---------|-----------|------|-------|--------|---------|
| RNN | 0 | 21±21 | 0.645±0.133 | 17±18 | 481±291 (10) | 826±95 (6) | Failed (0) | 16±15 |
| RNN | 2 | 136±100 | 0.807±0.028 | 113±73 | 428±169 (10) | 665±159 (10) | 849±113 (5) | 8±9 |
| RNN | 3 | 143±97 | 0.793±0.037 | 131±85 | 395±169 (10) | 667±126 (10) | 863±109 (6) | 27±33 |
| RNN | 4 | 152±115 | 0.785±0.022 | 129±96 | 379±212 (10) | 680±179 (10) | 865±124 (7) | 44±47 |
| RNN | 5 | 164±84 | 0.786±0.038 | 123±56 | 350±158 (10) | 643±121 (10) | 876±81 (8) | 40±41 |
| RNN | 6 | 224±104 | 0.790±0.041 | 181±79 | 352±176 (10) | 584±159 (10) | 782±56 (8) | 49±40 |
| RNN | 7 | 185±111 | 0.751±0.070 | 151±96 | 435±224 (10) | 608±127 (9) | 814±86 (7) | 116±119 |
| RNN | 8 | 159±128 | 0.775±0.050 | 128±114 | 460±195 (10) | 646±145 (9) | 858±140 (7) | 105±77 |
| RNN | 9 | 198±164 | 0.732±0.072 | 151±121 | 451±227 (10) | 641±158 (9) | 782±168 (6) | 285±396 |
| RNN | 10 | 139±127 | 0.728±0.078 | 100±73 | 512±212 (8) | 702±124 (7) | 867±145 (4) | 112±61 |
| RNN | 11 | 205±173 | 0.753±0.062 | 151±120 | 444±267 (10) | 652±234 (10) | 737±167 (6) | 254±320 |
| RNN | 12 | 261±165 | 0.762±0.057 | 211±135 | 320±246 (10) | 579±210 (10) | 775±168 (9) | 518±760 |
| RNN | 13 | 231±198 | 0.753±0.061 | 155±101 | 444±184 (9) | 601±235 (9) | 790±214 (8) | 351±289 |
| RNN | 14 | 158±103 | 0.718±0.091 | 108±60 | 526±208 (10) | 681±127 (9) | 845±80 (6) | 374±308 |
| RNN | 15 | 221±128 | 0.731±0.043 | 150±129 | 439±196 (10) | 618±168 (10) | 826±153 (9) | 461±292 |
| RNN | 16 | 196±145 | 0.725±0.043 | 136±101 | 470±228 (10) | 683±198 (10) | 813±141 (7) | 694±495 |
| RNN | 17 | 258±130 | 0.689±0.119 | 193±94 | 467±210 (10) | 576±139 (9) | 787±115 (9) | 796±600 |
| RNN | 18 | 253±114 | 0.727±0.047 | 195±98 | 394±175 (10) | 605±124 (10) | 764±82 (8) | 1112±974 |
| RNN | 19 | 268±159 | 0.714±0.052 | 204±132 | 418±161 (10) | 579±167 (10) | 745±153 (8) | 817±811 |
| RNN | 20 | 292±153 | 0.713±0.039 | 220±121 | 397±205 (10) | 574±188 (10) | 776±173 (10) | 1406±1391 |

## C.3 DO ARCHITECTURES DIFFER IN BEHAVIOR?

RNNs essentially solve the validity, uniqueness, and novelty metrics (Brown et al., 2019; Polykovskiy et al., 2020) and can learn molecular distributions well (Flam-Shepherd et al., 2022) for small molecule design. In this subsection, we extend Augmented Memory to Decoder transformer (Vaswani et al., 2017; Radford et al., 2019) and Mamba (Gu & Dao, 2023) to investigate the RL dynamics and empirically investigate potential benefits. Our hypothesis is that since self-attention (Vaswani et al., 2017) and selective scanning (Gu & Dao, 2023) *can* capture different structural elements (Özçelik et al., 2024) (via focusing on different aspects of the sequence), benefits *may* arise from capturing and focusing on favorable moieties. Our analysis is focused solely on sample efficiency metrics and not validity, uniqueness, and novelty.

Similar to the previous subsection, we perform a grid-search over batch size (64, 32, 16, 8) and augmentation rounds (0-20 inclusive except 1). As the results for RNN were presented in the previous subsection, this subsection only shows Decoder and Mamba results (Tables 9, 10, 11, 12, 13, 14, 15, and 16).

**The following observations are similar to RNN. Increasing augmentation rounds**:

1. Decreases diversity, as expected.

2. Increases the number of repeated SMILES.

**Decreasing batch size:**

1. Monotonically improves sample efficiency (though not always significant at the 95% confidence level).

2. Benefits Augmented memory more than REINVENT (0 augmentation rounds).

3. Increases the number of repeated SMILES.

4. Increases variance, as expected (since the expected reward is being approximated with a smaller batch size so it is more noisy).

5. Decreases diversity.

**The following observations contrast RNN with Decoder and Mamba**:

1. Mamba > Decoder > RNN in terms of NLL convergence (end of Appendix C.1).

2. Propensity to generate repeated SMILES follows the same trend and is further supported with the IntDiv1 generally being lower than RNN for the same number of augmentation rounds across all batch sizes.

Table 9: Decoder batch size 64.

| Model | Aug. Rounds | Yield | IntDiv1 | Scaffolds | OB 1 | OB 10 | OB 100 | Repeats |
|---|---|---|---|---|---|---|---|---|
| Decoder | 0 | 1±1 | 0.548±0.129 | 1±1 | 691±266 (6) | Failed (0) | Failed (0) | 2±1 |
| Decoder | 2 | 26±19 | 0.800±0.061 | 26±18 | 524±128 (10) | 868±76 (8) | Failed (0) | 0±0 |
| Decoder | 3 | 37±24 | 0.801±0.031 | 36±23 | 629±154 (10) | 849±85 (9) | Failed (0) | 0±0 |
| Decoder | 4 | 49±38 | 0.797±0.055 | 48±37 | 590±142 (10) | 851±89 (9) | 984±0 (1) | 0±0 |
| Decoder | 5 | 63±35 | 0.821±0.014 | 62±35 | 545±136 (10) | 814±84 (10) | 997±21 (2) | 1±1 |
| Decoder | 6 | 43±34 | 0.794±0.033 | 40±32 | 649±155 (10) | 881±127 (10) | 1045±0 (1) | 2±4 |
| Decoder | 7 | 42±29 | 0.800±0.039 | 41±29 | 585±175 (10) | 859±116 (9) | 1042±0 (1) | 4±3 |
| Decoder | 8 | 22±28 | 0.719±0.119 | 21±28 | 717±157 (10) | 939±104 (7) | 1051±0 (1) | 6±6 |
| Decoder | 9 | 23±22 | 0.704±0.156 | 19±16 | 618±233 (10) | 889±92 (7) | Failed (0) | 10±5 |
| Decoder | 10 | 43±48 | 0.768±0.056 | 41±47 | 643±110 (10) | 788±104 (6) | 980±0 (1) | 10±7 |
| Decoder | 11 | 36±45 | 0.756±0.068 | 34±44 | 698±116 (10) | 881±108 (8) | 891±0 (1) | 9±7 |
| Decoder | 12 | 47±28 | 0.795±0.02 | 43±27 | 609±101 (9) | 862±74 (9) | 1046±0 (1) | 16±9 |
| Decoder | 13 | 66±66 | 0.727±0.109 | 56±54 | 641±216 (10) | 788±148 (8) | 975±75 (2) | 37±25 |
| Decoder | 14 | 38±37 | 0.696±0.139 | 33±34 | 679±169 (10) | 868±104 (7) | 1004±0 (1) | 46±28 |
| Decoder | 15 | 38±56 | 0.671±0.100 | 25±32 | 668±241 (9) | 809±159 (5) | 977±9 (2) | 56±28 |
| Decoder | 16 | 33±41 | 0.716±0.084 | 25±29 | 572±221 (10) | 900±122 (8) | 984±0 (1) | 78±38 |
| Decoder | 17 | 50±48 | 0.707±0.091 | 37±30 | 595±250 (10) | 797±86 (7) | 1007±34 (2) | 91±42 |
| Decoder | 18 | 30±36 | 0.732±0.049 | 26±32 | 701±135 (8) | 886±101 (6) | 1025±0 (1) | 124±41 |
| Decoder | 19 | 35±31 | 0.715±0.056 | 28±21 | 640±240 (10) | 852±155 (8) | 1031±0 (1) | 159±64 |
| Decoder | 20 | 51±51 | 0.733±0.047 | 39±38 | 585±277 (9) | 862±136 (8) | 984±49 (2) | 172±69 |

3. Mamba notably generates many repeated SMILES but sample efficiency improves, thus it is not detrimental under the assumption that the reward is *near deterministic* and oracle evaluations are cached.

4. In general, Decoder does not outperform RNN

**Taking these observations together and exactly like RNN results, increasing augmentation rounds and decreasing batch size *can* trade-off diversity for sample efficiency (inconsistently and with higher variance).**

**However, of difference, is that Mamba at lower batch sizes (particularly 16) and relatively high augmentation rounds (10) improves sample efficiency in a statistically significant way (at the 95% confidence level).**

**Further note.** We have observed that with low batch size and high augmentation rounds, Mamba can temporarily lose generative ability. Specifically, the validity of the generated batch can be 0. Sampling a new batch can recover this validity but we have observed in extremely rare cases, that validity can be 0 for over 10 successive epochs. We observed this scenario twice in over 5,000 experiments, occurring with a batch size of 8 and augmentation rounds 19 and 20. We speculate the reason is extreme mode collapse to a chemical space where syntax is sensitive. Consequently, once the Selective Memory Purge starts penalizing the reward and the Agent is brought back towards the prior, large gradient updates coupled with sensitive syntax may cause invalid SMILES. This process often recovers but in practice, with high-fidelity oracles, one would checkpoint models frequently (even every epoch), as each batch of oracle evaluation would be costly. Alternatively, as all high reward SMILES (so far) generated can be pre-emptively saved. It would be feasible to even start a new run with these SMILES seeded in the replay buffer, akin to inception in REINVENT (Olivecrona et al., 2017) (transfer learning would work too). This would kick-start the optimization and already guide the Agent to this chemical space, preventing optimization progress from completely "lost". Moreover, we also do not recommend a batch size of 8 and augmentation rounds above 10 as the performance variance becomes high. This behavior is likely also highly dependent on the objective function which affects the optimization landscape. Finally, in the rare cases this occurs, and when validity recovers, the effect is minimal as sampling is cheap compared to oracle evaluations. We write this note for full transparency into all the behavior we have observed in our grid-search.

Fig. C3 shows a 2D heatmap of the sample efficiency (Yield) and diversity (IntDiv1) trade-off, as a function of augmentation rounds for Mamba with batch size 16.

Table 10: Decoder batch size 32.

| Model | Aug. Rounds | Yield | IntDiv1 | Scaffolds | OB 1 | OB 10 | OB 100 | Repeats |
|---|---|---|---|---|---|---|---|---|
| Decoder | 0 | 4±4 | 0.710±0.023 | 4±4 | 647±232 (6) | 982±39 (2) | Failed (0) | 10±13 |
| Decoder | 2 | 45±23 | 0.813±0.021 | 43±22 | 557±174 (10) | 844±91 (10) | Failed (0) | 1±1 |
| Decoder | 3 | 66±44 | 0.801±0.033 | 63±43 | 515±146 (10) | 779±70 (9) | 918±0 (1) | 1±1 |
| Decoder | 4 | 111±88 | 0.791±0.017 | 100±80 | 476±131 (10) | 726±133 (10) | 908±81 (5) | 3±3 |
| Decoder | 5 | 94±70 | 0.791±0.043 | 81±53 | 489±155 (10) | 753±112 (9) | 897±63 (3) | 3±2 |
| Decoder | 6 | 94±66 | 0.770±0.075 | 82±60 | 476±204 (10) | 696±126 (9) | 921±52 (4) | 11±6 |
| Decoder | 7 | 117±87 | 0.730±0.084 | 105±84 | 473±270 (10) | 659±99 (8) | 936±93 (6) | 54±84 |
| Decoder | 8 | 78±69 | 0.776±0.032 | 67±52 | 519±204 (10) | 797±147 (10) | 926±94 (3) | 35±13 |
| Decoder | 9 | 59±35 | 0.767±0.032 | 51±32 | 575±76 (10) | 856±83 (10) | 968±0 (1) | 44±33 |
| Decoder | 10 | 91±75 | 0.742±0.065 | 68±52 | 492±176 (9) | 769±121 (9) | 879±66 (2) | 77±56 |
| Decoder | 11 | 70±46 | 0.739±0.059 | 57±36 | 559±128 (10) | 811±96 (10) | 974±6 (3) | 84±45 |
| Decoder | 12 | 114±58 | 0.730±0.041 | 82±45 | 559±177 (10) | 715±59 (9) | 942±48 (6) | 124±81 |
| Decoder | 13 | 93±83 | 0.741±0.064 | 77±68 | 598±114 (10) | 788±129 (9) | 874±34 (3) | 146±76 |
| Decoder | 14 | 147±112 | 0.752±0.064 | 109±84 | 486±147 (9) | 694±152 (9) | 791±37 (4) | 257±269 |
| Decoder | 15 | 140±100 | 0.718±0.085 | 111±78 | 516±256 (10) | 676±143 (9) | 916±106 (7) | 222±128 |
| Decoder | 16 | 130±142 | 0.709±0.045 | 82±66 | 552±177 (10) | 772±164 (10) | 851±173 (4) | 405±272 |
| Decoder | 17 | 130±125 | 0.720±0.075 | 95±89 | 624±209 (10) | 771±186 (10) | 841±137 (4) | 444±265 |
| Decoder | 18 | 153±165 | 0.718±0.055 | 110±130 | 565±191 (10) | 718±197 (9) | 668±81 (3) | 544±503 |
| Decoder | 19 | 149±94 | 0.686±0.055 | 104±69 | 547±215 (10) | 731±113 (9) | 897±83 (7) | 594±172 |
| Decoder | 20 | 137±135 | 0.693±0.046 | 78±56 | 555±200 (9) | 740±181 (9) | 855±145 (5) | 514±399 |

Table 11: Decoder batch size 16.

| Model | Aug. Rounds | Yield | IntDiv1 | Scaffolds | OB 1 | OB 10 | OB 100 | Repeats |
|---|---|---|---|---|---|---|---|---|
| Decoder | 0 | 2±3 | 0.55±0.1 | 2±2 | 810±93 (7) | 983±0 (1) | Failed (0) | 78±25 |
| Decoder | 2 | 66±50 | 0.796±0.037 | 59±41 | 602±158 (10) | 799±106 (9) | 921±3 (2) | 8±7 |
| Decoder | 3 | 84±66 | 0.77±0.037 | 64±44 | 536±170 (10) | 769±122 (9) | 919±44 (4) | 28±24 |
| Decoder | 4 | 71±44 | 0.74±0.102 | 62±41 | 632±118 (10) | 780±82 (9) | 977±36 (3) | 22±12 |
| Decoder | 5 | 154±93 | 0.748±0.052 | 122±70 | 439±151 (10) | 679±128 (10) | 907±92 (8) | 90±90 |
| Decoder | 6 | 116±94 | 0.748±0.039 | 86±64 | 517±165 (10) | 728±158 (10) | 904±126 (5) | 73±42 |
| Decoder | 7 | 108±85 | 0.747±0.051 | 71±50 | 510±222 (10) | 740±127 (9) | 868±48 (4) | 126±63 |
| Decoder | 8 | 108±94 | 0.708±0.109 | 72±57 | 538±164 (10) | 742±116 (9) | 887±87 (4) | 150±72 |
| Decoder | 9 | 78±83 | 0.687±0.116 | 51±55 | 614±244 (10) | 790±150 (8) | 890±62 (3) | 242±139 |
| Decoder | 10 | 120±128 | 0.691±0.042 | 74±73 | 663±170 (9) | 768±169 (8) | 805±65 (4) | 344±218 |
| Decoder | 11 | 146±134 | 0.727±0.038 | 110±100 | 609±169 (9) | 725±166 (9) | 829±132 (5) | 389±199 |
| Decoder | 12 | 119±127 | 0.704±0.047 | 76±68 | 624±185 (9) | 779±176 (9) | 828±110 (4) | 363±256 |
| Decoder | 13 | 183±177 | 0.696±0.031 | 97±80 | 484±227 (9) | 671±216 (9) | 753±144 (5) | 498±412 |
| Decoder | 14 | 146±111 | 0.673±0.055 | 88±60 | 572±240 (10) | 737±162 (9) | 850±87 (6) | 702±387 |
| Decoder | 15 | 146±100 | 0.64±0.123 | 108±79 | 623±141 (10) | 772±150 (10) | 867±70 (6) | 774±414 |
| Decoder | 16 | 209±173 | 0.688±0.043 | 155±130 | 530±124 (9) | 654±161 (9) | 813±170 (7) | 1369±777 |
| Decoder | 17 | 190±168 | 0.662±0.109 | 154±149 | 571±207 (10) | 674±179 (9) | 746±162 (5) | 1096±883 |
| Decoder | 18 | 226±138 | 0.668±0.052 | 174±115 | 550±156 (10) | 646±131 (9) | 802±118 (8) | 1540±986 |
| Decoder | 19 | 232±154 | 0.648±0.07 | 168±96 | 564±152 (10) | 681±161 (10) | 781±147 (7) | 1693±1165 |
| Decoder | 20 | 258±200 | 0.636±0.077 | 166±103 | 448±223 (9) | 589±179 (8) | 763±177 (8) | 1741±1020 |

Table 12: Decoder batch size 8.

| Model | Aug. Rounds | Yield | IntDiv1 | Scaffolds | OB 1 | OB 10 | OB 100 | Repeats |
|---|---|---|---|---|---|---|---|---|
| Decoder | 0 | 57±64 | 0.621±0.222 | 37±36 | 554±137 (9) | 766±178 (7) | 912±52 (3) | 368±164 |
| Decoder | 2 | 120±76 | 0.745±0.055 | 97±59 | 497±207 (10) | 667±110 (8) | 913±62 (7) | 39±22 |
| Decoder | 3 | 93±60 | 0.73±0.06 | 74±45 | 530±166 (10) | 759±87 (9) | 918±22 (4) | 128±82 |
| Decoder | 4 | 111±49 | 0.741±0.036 | 79±34 | 467±170 (10) | 737±101 (10) | 950±32 (7) | 173±81 |
| Decoder | 5 | 79±82 | 0.724±0.044 | 59±54 | 609±123 (8) | 805±101 (8) | 901±72 (3) | 283±179 |
| Decoder | 6 | 138±112 | 0.72±0.062 | 96±78 | 608±162 (10) | 737±138 (9) | 843±81 (5) | 400±222 |
| Decoder | 7 | 197±165 | 0.688±0.064 | 149±131 | 502±287 (10) | 684±237 (10) | 758±112 (6) | 820±1051 |
| Decoder | 8 | 219±179 | 0.68±0.063 | 132±120 | 475±201 (8) | 581±127 (7) | 763±136 (7) | 840±900 |
| Decoder | 9 | 194±144 | 0.651±0.049 | 153±118 | 496±157 (8) | 627±149 (8) | 791±109 (7) | 1059±864 |
| Decoder | 10 | 183±200 | 0.684±0.055 | 130±130 | 571±201 (9) | 654±217 (8) | 789±205 (6) | 944±597 |
| Decoder | 11 | 141±123 | 0.581±0.166 | 96±84 | 617±198 (9) | 662±142 (7) | 801±97 (5) | 1715±1380 |
| Decoder | 12 | 133±196 | 0.574±0.149 | 92±135 | 665±291 (9) | 699±268 (7) | 664±209 (3) | 1604±1130 |
| Decoder | 13 | 331±151 | 0.664±0.095 | 271±143 | 418±230 (10) | 503±88 (9) | 711±107 (9) | 2030±1408 |
| Decoder | 14 | 164±152 | 0.602±0.06 | 125±109 | 620±257 (9) | 714±194 (8) | 825±133 (6) | 2628±1665 |
| Decoder | 15 | 281±242 | 0.661±0.054 | 230±185 | 496±243 (9) | 589±251 (9) | 663±201 (7) | 2482±1515 |
| Decoder | 16 | 213±191 | 0.58±0.143 | 180±176 | 512±245 (9) | 596±223 (8) | 730±186 (6) | 3113±2436 |
| Decoder | 17 | 252±186 | 0.622±0.072 | 203±167 | 614±231 (10) | 615±169 (8) | 735±139 (7) | 3278±1894 |
| Decoder | 18 | 81±113 | 0.595±0.064 | 69±97 | 630±232 (7) | 759±209 (7) | 862±102 (3) | 2811±2415 |
| Decoder | 19 | 136±171 | 0.611±0.062 | 119±154 | 645±195 (7) | 708±180 (6) | 771±142 (4) | 2886±2066 |
| Decoder | 20 | 98±139 | 0.54±0.075 | 91±136 | 736±195 (7) | 785±160 (6) | 813±140 (3) | 3190±2113 |

Table 13: Mamba batch size 64.

| Model | Aug. Rounds | Yield | IntDiv1 | Scaffolds | OB 1 | OB 10 | OB 100 | Repeats |
|---|---|---|---|---|---|---|---|---|
| Mamba | 0 | 0±0 | — | 0±0 | 946±41 (2) | Failed (0) | Failed (0) | 0±1 |
| Mamba | 2 | 2±1 | 0.580±0.086 | 2±1 | 817±244 (10) | Failed (0) | Failed (0) | 0±0 |
| Mamba | 3 | 9±6 | 0.734±0.068 | 9±6 | 659±234 (9) | 942±34 (4) | Failed (0) | 1±1 |
| Mamba | 4 | 6±3 | 0.672±0.114 | 6±3 | 652±297 (10) | 1040±7 (2) | Failed (0) | 2±2 |
| Mamba | 5 | 9±5 | 0.697±0.113 | 9±5 | 640±210 (10) | 995±30 (5) | Failed (0) | 3±3 |
| Mamba | 6 | 17±11 | 0.770±0.041 | 17±11 | 656±119 (10) | 960±90 (9) | Failed (0) | 6±4 |
| Mamba | 7 | 19±6 | 0.769±0.027 | 18±6 | 623±152 (10) | 957±65 (9) | Failed (0) | 7±3 |
| Mamba | 8 | 29±15 | 0.786±0.035 | 27±15 | 545±176 (10) | 917±82 (10) | Failed (0) | 12±8 |
| Mamba | 9 | 21±10 | 0.755±0.075 | 20±10 | 585±192 (10) | 938±57 (9) | Failed (0) | 26±23 |
| Mamba | 10 | 34±22 | 0.785±0.028 | 28±15 | 486±176 (10) | 884±91 (10) | Failed (0) | 30±21 |
| Mamba | 11 | 18±8 | 0.757±0.044 | 17±7 | 550±203 (10) | 937±31 (8) | Failed (0) | 37±21 |
| Mamba | 12 | 22±17 | 0.727±0.051 | 20±15 | 629±234 (10) | 876±53 (6) | Failed (0) | 72±68 |
| Mamba | 13 | 33±33 | 0.739±0.090 | 29±28 | 561±222 (10) | 915±120 (10) | 1020±0 (1) | 62±28 |
| Mamba | 14 | 47±39 | 0.701±0.138 | 30±15 | 540±242 (10) | 839±94 (8) | 980±0 (1) | 127±56 |
| Mamba | 15 | 60±88 | 0.725±0.117 | 31±17 | 585±225 (10) | 866±143 (10) | 726±0 (1) | 136±112 |
| Mamba | 16 | 46±40 | 0.661±0.170 | 29±22 | 614±193 (10) | 865±104 (9) | 978±33 (2) | 199±89 |
| Mamba | 17 | 43±24 | 0.727±0.054 | 30±13 | 538±185 (10) | 866±101 (10) | Failed (0) | 174±77 |
| Mamba | 18 | 51±42 | 0.732±0.056 | 40±32 | 621±219 (10) | 838±111 (9) | 995±34 (2) | 262±99 |
| Mamba | 19 | 49±40 | 0.723±0.048 | 36±25 | 633±218 (10) | 829±123 (8) | 975±0 (1) | 241±73 |
| Mamba | 20 | 77±68 | 0.695±0.088 | 46±32 | 549±241 (9) | 771±146 (8) | 940±76 (3) | 385±180 |

Table 14: Mamba batch size 32.

| Model | Aug. Rounds | Yield | IntDiv1 | Scaffolds | OB 1 | OB 10 | OB 100 | Repeats |
|---|---|---|---|---|---|---|---|---|
| Mamba | 0 | 0±0 | — | 0±0 | 773±189 (4) | Failed (0) | Failed (0) | 4±2 |
| Mamba | 2 | 12±7 | 0.744±0.060 | 12±7 | 644±199 (10) | 933±29 (5) | Failed (0) | 3±2 |
| Mamba | 3 | 16±9 | 0.759±0.050 | 15±9 | 640±158 (10) | 912±45 (6) | Failed (0) | 8±7 |
| Mamba | 4 | 30±15 | 0.797±0.029 | 29±15 | 579±140 (10) | 879±86 (10) | Failed (0) | 11±5 |
| Mamba | 5 | 38±23 | 0.718±0.151 | 35±21 | 695±159 (10) | 833±83 (8) | Failed (0) | 24±9 |
| Mamba | 6 | 44±37 | 0.770±0.044 | 41±34 | 564±145 (10) | 861±110 (9) | 1000±3 (2) | 42±17 |
| Mamba | 7 | 52±43 | 0.750±0.047 | 46±37 | 539±174 (10) | 848±123 (10) | 996±11 (2) | 68±28 |
| Mamba | 8 | 76±51 | 0.775±0.025 | 67±45 | 515±108 (10) | 794±85 (10) | 923±30 (2) | 90±49 |
| Mamba | 9 | 64±47 | 0.755±0.083 | 53±38 | 546±143 (10) | 808±116 (10) | 959±45 (2) | 140±106 |
| Mamba | 10 | 96±76 | 0.768±0.028 | 75±54 | 553±186 (10) | 782±161 (10) | 949±84 (5) | 165±63 |
| Mamba | 11 | 87±60 | 0.732±0.045 | 62±40 | 592±218 (10) | 741±105 (8) | 936±31 (3) | 303±152 |
| Mamba | 12 | 118±60 | 0.680±0.130 | 67±21 | 500±159 (10) | 730±132 (10) | 932±61 (6) | 280±151 |
| Mamba | 13 | 92±60 | 0.742±0.082 | 74±43 | 578±226 (10) | 771±98 (9) | 940±39 (4) | 353±104 |
| Mamba | 14 | 166±75 | 0.748±0.041 | 121±54 | 458±97 (10) | 659±64 (10) | 901±78 (8) | 483±202 |
| Mamba | 15 | 139±94 | 0.755±0.033 | 106±72 | 456±141 (10) | 740±127 (10) | 847±54 (5) | 488±167 |
| Mamba | 16 | 136±75 | 0.740±0.039 | 97±54 | 571±131 (10) | 742±119 (10) | 899±50 (6) | 769±354 |
| Mamba | 17 | 186±88 | 0.696±0.058 | 138±83 | 510±103 (10) | 683±88 (10) | 871±76 (8) | 937±677 |
| Mamba | 18 | 214±87 | 0.723±0.059 | 169±81 | 540±113 (10) | 672±88 (10) | 862±84 (9) | 1027±554 |
| Mamba | 19 | 242±109 | 0.686±0.041 | 184±104 | 493±133 (10) | 661±116 (10) | 819±109 (9) | 1376±596 |
| Mamba | 20 | 187±78 | 0.706±0.038 | 152±67 | 557±101 (10) | 714±80 (10) | 892±79 (9) | 1183±413 |

Table 15: Mamba batch size 16.

| Model | Aug. Rounds | Yield | IntDiv1 | Scaffolds | OB 1 | OB 10 | OB 100 | Repeats |
|---|---|---|---|---|---|---|---|---|
| Mamba | 0 | 3±4 | 0.417±0.161 | 2±2 | 545±232 (7) | 982±0 (1) | Failed (0) | 91±32 |
| Mamba | 2 | 39±29 | 0.761±0.047 | 34±23 | 609±165 (10) | 829±117 (9) | Failed (0) | 46±31 |
| Mamba | 3 | 61±51 | 0.771±0.051 | 50±39 | 498±193 (10) | 797±118 (9) | 953±15 (3) | 71±28 |
| Mamba | 4 | 52±33 | 0.779±0.031 | 42±23 | 581±102 (10) | 817±112 (10) | 970±0 (1) | 139±59 |
| Mamba | 5 | 69±38 | 0.764±0.052 | 54±28 | 542±93 (10) | 807±76 (10) | 988±17 (3) | 178±90 |
| Mamba | 6 | 138±46 | 0.759±0.039 | 110±42 | 456±89 (10) | 693±75 (10) | 919±36 (7) | 286±137 |
| Mamba | 7 | 174±95 | 0.737±0.059 | 127±83 | 427±177 (10) | 643±102 (10) | 858±77 (10) | 395±147 |
| Mamba | 8 | 209±95 | 0.751±0.030 | 137±60 | 461±151 (10) | 617±135 (10) | 817±71 (8) | 482±214 |
| Mamba | 9 | 202±98 | 0.735±0.032 | 137±80 | 389±112 (10) | 631±102 (10) | 841±92 (8) | 518±237 |
| Mamba | 10 | 306±57 | 0.714±0.035 | 206±34 | 387±148 (10) | 555±66 (10) | 761±58 (10) | 1110±636 |
| Mamba | 11 | 306±92 | 0.716±0.039 | 237±85 | 403±136 (10) | 554±93 (10) | 761±100 (10) | 1341±596 |
| Mamba | 12 | 266±100 | 0.723±0.041 | 199±83 | 392±126 (10) | 590±100 (10) | 806±111 (10) | 1312±666 |
| Mamba | 13 | 327±108 | 0.722±0.043 | 258±101 | 428±111 (10) | 549±111 (10) | 741±116 (10) | 1508±780 |
| Mamba | 14 | 318±109 | 0.695±0.061 | 246±117 | 416±164 (10) | 535±148 (10) | 736±123 (10) | 1776±912 |
| Mamba | 15 | 284±74 | 0.691±0.052 | 219±42 | 442±67 (10) | 584±87 (10) | 785±82 (10) | 2629±939 |
| Mamba | 16 | 293±112 | 0.672±0.053 | 209±77 | 483±145 (10) | 570±136 (10) | 767±130 (10) | 2284±1011 |
| Mamba | 17 | 344±115 | 0.656±0.047 | 278±92 | 462±113 (10) | 563±98 (10) | 725±121 (10) | 3512±1227 |
| Mamba | 18 | 281±155 | 0.640±0.082 | 216±125 | 464±174 (9) | 595±155 (9) | 730±93 (8) | 2885±1344 |
| Mamba | 19 | 307±115 | 0.624±0.084 | 238±102 | 491±146 (10) | 579±133 (10) | 750±119 (10) | 3318±1347 |
| Mamba | 20 | 352±69 | 0.673±0.046 | 294±61 | 403±102 (10) | 525±81 (10) | 714±79 (10) | 3331±1454 |

Table 16: Mamba batch size 8.

| Model | Aug. Rounds | Yield | IntDiv1 | Scaffolds | OB 1 | OB 10 | OB 100 | Repeats |
|-------|-------------|-------|---------|-----------|------|-------|--------|---------|
| Mamba | 0 | 3±2 | 0.43±0.133 | 2±1 | 498±322 (8) | Failed (0) | Failed (0) | 940±234 |
| Mamba | 2 | 69±32 | 0.755±0.059 | 56±28 | 453±176 (10) | 780±78 (10) | 992±8 (2) | 214±72 |
| Mamba | 3 | 156±113 | 0.745±0.035 | 109±70 | 452±221 (10) | 659±143 (9) | 792±83 (5) | 282±120 |
| Mamba | 4 | 200±117 | 0.748±0.046 | 125±64 | 402±208 (10) | 602±150 (10) | 859±145 (9) | 425±160 |
| Mamba | 5 | 240±102 | 0.719±0.062 | 195±102 | 429±191 (10) | 596±136 (10) | 805±108 (9) | 1195±687 |
| Mamba | 6 | 298±167 | 0.706±0.052 | 212±122 | 405±190 (10) | 557±197 (10) | 736±170 (9) | 1420±632 |
| Mamba | 7 | 328±116 | 0.662±0.107 | 246±112 | 332±142 (10) | 489±131 (10) | 727±124 (10) | 1657±947 |
| Mamba | 8 | 356±142 | 0.671±0.029 | 304±119 | 380±158 (10) | 514±144 (10) | 699±167 (10) | 2340±806 |
| Mamba | 9 | 359±135 | 0.682±0.054 | 298±115 | 439±140 (10) | 536±161 (10) | 663±102 (9) | 2974±1394 |
| Mamba | 10 | 368±164 | 0.692±0.032 | 305±154 | 391±234 (10) | 485±99 (9) | 658±125 (9) | 2829±1290 |
| Mamba | 11 | 321±148 | 0.636±0.048 | 280±137 | 415±154 (10) | 561±153 (10) | 720±145 (9) | 3515±1592 |
| Mamba | 12 | 335±148 | 0.637±0.055 | 285±148 | 425±162 (10) | 564±178 (10) | 687±135 (9) | 4060±1694 |
| Mamba | 13 | 260±158 | 0.579±0.121 | 213±139 | 505±168 (10) | 602±141 (9) | 744±130 (8) | 3691±1790 |
| Mamba | 14 | 290±120 | 0.608±0.047 | 235±89 | 463±213 (10) | 583±150 (10) | 765±127 (10) | 4505±1968 |
| Mamba | 15 | 343±157 | 0.621±0.069 | 317±149 | 367±140 (10) | 534±159 (10) | 706±166 (10) | 4196±1064 |
| Mamba | 16 | 320±214 | 0.61±0.095 | 293±199 | 450±210 (10) | 560±241 (9) | 602±141 (7) | 5035±1995 |
| Mamba | 17 | 233±131 | 0.611±0.059 | 219±131 | 552±165 (10) | 665±147 (10) | 806±130 (9) | 3728±1946 |
| Mamba | 18 | 270±205 | 0.617±0.061 | 256±200 | 516±155 (10) | 628±191 (10) | 705±201 (7) | 5378±2020 |
| Mamba | 19 | 168±164 | 0.632±0.070 | 139±121 | 468±221 (8) | 604±233 (8) | 805±193 (6) | 4740±2181 |
| Mamba | 20 | 256±196 | 0.539±0.190 | 245±192 | 462±225 (9) | 531±233 (8) | 642±156 (7) | 4476±2383 |

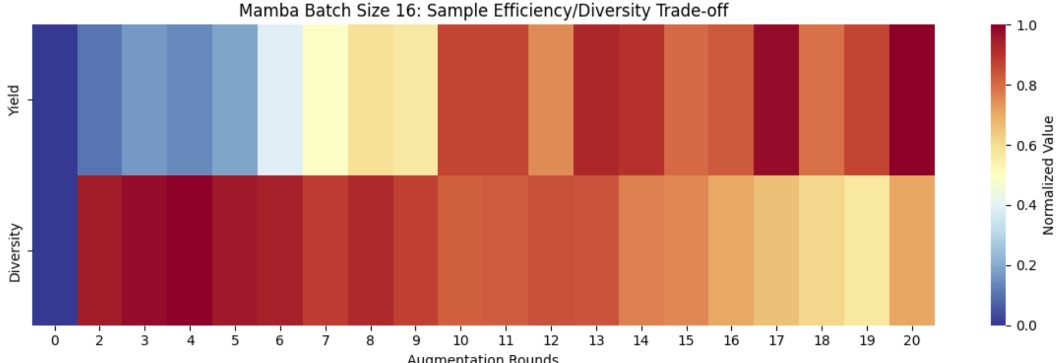

Figure C3: Mamba (batch size 16, augmentation rounds 10) Sample efficiency (Yield) and Diversity (IntDiv1) trade-off.

## C.4 ARE INCREASED AUGMENTATION ROUNDS STILL SYNERGISTIC WITH BEAM ENUMERATION?

Beam Enumeration (Guo & Schwaller, 2024b) extracts the most probable substructures for self-conditioned generation and has been shown to be synergistic with Augmented Memory (Guo & Schwaller, 2024a) such that the Yield and OB improve. In the original work, the oracle budget in the experiments was 5,000. In this work, we are interested in minimizing the oracle budget and all experiments thus far use a 1,000 oracle budget. Beam Enumeration has a *Patience* criterion which controls when substructures are extracted: only when the average reward improves for *Patience* number of successive epochs. Since we are operating at a much lower oracle budget, it is especially unclear whether Beam Enumeration can still benefit sample efficiency (we note that the explainability aspect is still applicable). In the original work, a batch size of 64 was used and a Patience of 5. Under these parameters, the earliest that Beam Enumeration can execute is 320/1000 oracle calls, which is almost 1/3 the budget already. Moreover, Beam Enumeration decreases diversity and decreasing batch size and increasing augmentation rounds also decreases diversity. *Too much* decrease in diversity may be detrimental even with oracle caching. In this subsection, we systematically study the effect of Beam Enumeration when used in conjunction with decreasing batch size and augmentation rounds in a series of hypotheses.

**Based on observations from batch size and augmentation rounds grid-searches, the following design decisions were made in this subsection**:

1. Augmentation rounds capped at 5 as diversity generally decreases more substantially past this point. Beam Enumeration itself will decrease diversity, so this is a preemptive measure against detrimental diversity-induced mode collapse.

2. Investigate batch sizes of 64 and 32. Since Beam Enumeration executes on improved reward over successive epochs, lower batch sizes would likely increase performance variance too much.

3. Focus only on RNN model as experiments will be the fastest (less repeated SMILES). If benefits are observed, move to Decoder and Mamba models. For clarity, repeated SMILES are not detrimental, as we have shown in the previous subsections but they add some wall time (this is insignificant when compared to expensive oracles).

4. Beam Enumeration can pool improbable substructures. There is a Patience Limit denoting the number epochs permitted where the entire generated batch is filtered. This limit was 100,000 in this work. This does not add that much wall time and surpassing the limit is not indicative of the experiment failing. However, we enforce this upper bound in case it occurs (seldom) to manage wall times since we are performing grid searches.

5. Use Minimum Structure Size = 15, unless otherwise stated. Enforcing larger substructure extraction was found to improve sample efficiency in the original work (Guo & Schwaller, 2024b)

### C.4.1 HYPOTHESIS 1

Beam Enumeration's Patience parameter is dependent on the mean reward of the sampled batch. With lower batch sizes, variance increases, such that executing Beam Enumeration may be *too variable*.

**Proposed solution.** Increase Beam Enumeration's default Patience (5) to mitigate lower batch size variance. We note that increasing Patience means that more of the oracle budget needs to be consumed before Beam Enumeration executes for the first time. First explore Batch sizes = [64, 32].

**Observations.** Across batch sizes = [64, 32] and all Patience = [5, 6, 7, 8, 9, 10], sample efficiency does not improve in a statistically significant manner (Tables 17 and 18). Using Beam Enumeration also leads to notably higher variance and decreased diversity.

### C.4.2 HYPOTHESIS 2

The use of "Structure" substructure is too biased when operating in an already biased environment: increasing augmentation rounds and under a low oracle budget.

**Proposed solution.** Investigate "Scaffold" substructure which is less biased.

Table 17: Beam Enumeration batch size 64 with Structure and Minimum Size 15. Filter Limit is the number of times that no SMILES contained the pool substructure in 100,000 generation epochs. Patience N/A indicates just Augmented Memory and no Beam Enumeration.

| Patience | Aug. Rounds | Yield | IntDiv1 | Scaffolds | OB 1 | OB 10 | OB 100 | Repeats | Filter Limit |
|---|---|---|---|---|---|---|---|---|---|
| N/A | 0 | 0±0 | — | 0±0 | 584±251 (5) | Failed | Failed | 1±1 | N/A |
| N/A | 2 | 15±9 | 0.775±0.073 | 15±9 | 644±173 (10) | 941±58 (8) | Failed | 0±0 | N/A |
| N/A | 3 | 33±42 | 0.788±0.043 | 32±40 | 613±96 (10) | 927±128 (9) | 993±0 (1) | 0±0 | N/A |
| N/A | 4 | 32±16 | 0.813±0.024 | 31±16 | 527±198 (10) | 880±90 (10) | Failed | 0±0 | N/A |
| N/A | 5 | 40±14 | 0.812±0.023 | 39±13 | 459±177 (10) | 862±68 (10) | Failed | 0±0 | N/A |
| 5 | 0 | 2±2 | — | 2±2 | 687±232 (7) | Failed | Failed | 17±21 | 0 |
| 5 | 2 | 29±68 | 0.688±0.044 | 22±48 | 555±185 (8) | 887±182 (4) | 866±0 (1) | 15±27 | 1 |
| 5 | 3 | 110±75 | 0.754±0.024 | 81±52 | 488±79 (10) | 711±99 (10) | 902±79 (4) | 20±21 | 0 |
| 5 | 4 | 86±82 | 0.702±0.045 | 58±53 | 504±205 (10) | 739±193 (9) | 912±76 (3) | 14±15 | 0 |
| 5 | 5 | 94±41 | 0.745±0.027 | 68±30 | 436±167 (10) | 739±88 (10) | 970±30 (4) | 15±17 | 0 |
| 6 | 0 | 2±3 | — | 2±2 | 581±205 (7) | 958±0 (1) | Failed | 25±29 | 0 |
| 6 | 2 | 20±20 | 0.619±0.168 | 16±15 | 659±226 (10) | 809±27 (4) | Failed | 9±10 | 0 |
| 6 | 3 | 82±84 | 0.73±0.039 | 52±44 | 520±84 (10) | 777±134 (10) | 863±131 | 19±26 | 0 |
| 6 | 4 | 83±91 | 0.723±0.074 | 62±62 | 508±233 (9) | 737±130 (8) | 874±93 | 19±21 | 0 |
| 6 | 5 | 84±52 | 0.693±0.049 | 54±30 | 449±169 (10) | 771±131 (10) | 973±44 | 38±56 | 0 |
| 7 | 0 | 2±2 | — | 2±2 | 599±238 (6) | Failed | Failed | 15±17 | 0 |
| 7 | 2 | 40±43 | 0.661±0.161 | 32±34 | 579±137 (10) | 836±112 (8) | 1000±28 (2) | 9±10 | 0 |
| 7 | 3 | 121±120 | 0.719±0.038 | 80±69 | 546±66 (10) | 735±131 (10) | 803±75 (3) | 27±30 | 0 |
| 7 | 4 | 69±64 | 0.701±0.098 | 45±39 | 560±249 (10) | 726±84 (7) | 941±55 (2) | 12±18 | 0 |
| 7 | 5 | 61±34 | 0.735±0.055 | 43±21 | 467±188 (10) | 796±77 (10) | 1026±4 (2) | 11±15 | 0 |
| 8 | 0 | 1±2 | — | 1±1 | 556±225 (5) | 1010±0 (1) | Failed | 24±32 | 0 |
| 8 | 2 | 80±90 | 0.697±0.074 | 51±60 | 604±153 (10) | 775±119 (8) | 882±94 (3) | 8±11 | 0 |
| 8 | 3 | 79±86 | 0.714±0.028 | 58±67 | 579±88 (10) | 769±131 (9) | 920±139 (3) | 7±6 | 0 |
| 8 | 4 | 68±85 | 0.671±0.044 | 45±55 | 537±202 (10) | 786±115 (6) | 902±49 (3) | 20±23 | 0 |
| 8 | 5 | 88±61 | 0.711±0.098 | 64±45 | 459±184 (10) | 757±118 (9) | 960±33 (4) | 15±27 | 0 |
| 9 | 0 | 1±1 | — | 1±1 | 564±226 (5) | Failed | Failed | 11±11 | 0 |
| 9 | 2 | 49±53 | 0.7±0.119 | 36±34 | 620±171 (10) | 826±115 (8) | 953±12 (2) | 2±4 | 0 |
| 9 | 3 | 87±81 | 0.739±0.034 | 53±38 | 599±92 (10) | 787±100 (10) | 935±122 (3) | 9±11 | 0 |
| 9 | 4 | 65±49 | 0.688±0.08 | 48±41 | 518±187 (10) | 798±88 (10) | 910±0 (1) | 11±17 | 0 |
| 9 | 5 | 99±84 | 0.694±0.098 | 60±51 | 459±180 (10) | 774±80 (10) | 907±93 (3) | 19±27 | 0 |
| 10 | 0 | 1±1 | — | 1±1 | 564±226 (5) | Failed | Failed | 11±11 | 0 |
| 10 | 2 | 49±53 | 0.7±0.119 | 36±34 | 620±171 (10) | 826±115 (8) | 953±12 (2) | 2±4 | 0 |
| 10 | 3 | 87±81 | 0.739±0.034 | 53±38 | 599±92 (10) | 787±100 (10) | 935±122 (3) | 9±11 | 0 |
| 10 | 4 | 65±49 | 0.688±0.08 | 48±41 | 518±187 (10) | 798±88 (10) | 910±0 (1) | 11±17 | 0 |
| 10 | 5 | 99±84 | 0.694±0.098 | 60±51 | 459±180 (10) | 774±80 (10) | 907±93 (3) | 19±27 | 0 |

Table 18: Beam Enumeration batch size 32 with Structure and Minimum Size 15. Filter Limit is the number of times that no SMILES contained the pool substructure in 100,000 generation epochs. Patience N/A indicates just Augmented Memory and no Beam Enumeration.

| Patience | Aug. Rounds | Yield | IntDiv1 | Scaffolds | OB 1 | OB 10 | OB 100 | Repeats | Filter Limit |
|---|---|---|---|---|---|---|---|---|---|
| N/A | 0 | 0±0 | — | 0±0 | 798±101 (5) | Failed | Failed | 1±1 | N/A |
| N/A | 2 | 43±25 | 0.825±0.029 | 42±24 | 608±151 (10) | 844±90 (9) | Failed | 0±0 | N/A |
| N/A | 3 | 52±34 | 0.81±0.059 | 51±32 | 522±141 (10) | 789±100 (9) | 1018±0 (2) | 0±1 | N/A |
| N/A | 4 | 87±33 | 0.82±0.018 | 83±31 | 466±120 (10) | 740±77 (10) | 987±30 (4) | 1±3 | N/A |
| N/A | 5 | 98±57 | 0.817±0.027 | 89±50 | 408±184 (10) | 714±136 (10) | 915±20 (4) | 1±2 | N/A |
| 5 | 0 | 2±4 | 0.611±0.074 | 2±3 | 776±155 (4) | 983±0 (1) | Failed | 43±30 | 0 |
| 5 | 2 | 18±27 | 0.666±0.077 | 15±19 | 705±173 (8) | 857±104 (4) | Failed | 9±9 | 0 |
| 5 | 3 | 26±20 | 0.652±0.076 | 19±11 | 618±88 (10) | 850±108 (7) | Failed | 16±18 | 0 |
| 5 | 4 | 65±64 | 0.695±0.092 | 54±53 | 604±214 (10) | 742±124 (6) | 936±55 (3) | 65±90 | 0 |
| 5 | 5 | 99±110 | 0.713±0.046 | 66±61 | 452±216 (10) | 741±173 (9) | 870±146 (4) | 64±56 | 0 |
| 6 | 0 | 2±5 | 0.655±0.051 | 2±4 | 614±213 (4) | 836±0 (1) | Failed | 39±27 | 0 |
| 6 | 2 | 36±49 | 0.691±0.096 | 32±47 | 625±188 (9) | 834±139 (7) | 943±31 (2) | 9±9 | 0 |
| 6 | 3 | 60±58 | 0.662±0.124 | 47±53 | 574±148 (10) | 811±146 (10) | 895±81 (2) | 93±220 | 0 |
| 6 | 4 | 67±52 | 0.654±0.185 | 54±43 | 592±214 (10) | 740±133 (8) | 934±50 (3) | 114±154 | 0 |
| 6 | 5 | 66±70 | 0.68±0.059 | 50±44 | 530±209 (10) | 822±141 (9) | 933±69 (3) | 65±70 | 0 |
| 7 | 0 | 1±2 | — | 1±2 | 686±161 (6) | Failed | Failed | 83±78 | 0 |
| 7 | 2 | 49±60 | 0.699±0.101 | 41±56 | 601±156 (10) | 821±152 (8) | 923±93 (2) | 18±20 | 0 |
| 7 | 3 | 47±46 | 0.67±0.107 | 37±36 | 623±198 (9) | 810±161 (8) | 994±16 (3) | 20±21 | 0 |
| 7 | 4 | 41±45 | 0.686±0.058 | 33±42 | 588±81 (9) | 838±94 (9) | 905±0 (1) | 53±43 | 0 |
| 7 | 5 | 76±76 | 0.698±0.111 | 66±74 | 531±210 (10) | 776±128 (8) | 866±69 (2) | 126±325 | 0 |
| 8 | 0 | 16±37 | — | 14±33 | 749±210 (8) | 668±194 (2) | 949±0 (1) | 109±163 | 0 |
| 8 | 2 | 33±48 | 0.691±0.049 | 24±33 | 692±144 (9) | 856±142 (6) | 974±35 (2) | 15±18 | 0 |
| 8 | 3 | 50±30 | 0.675±0.068 | 40±22 | 636±109 (10) | 803±84 (8) | Failed | 39±49 | 0 |
| 8 | 4 | 104±104 | 0.73±0.056 | 84±96 | 406±128 (10) | 696±149 (9) | 879±141 (4) | 30±36 | 0 |
| 8 | 5 | 42±30 | 0.7±0.051 | 32±18 | 506±186 (10) | 848±95 (10) | 974±0 (1) | 30±45 | 0 |
| 9 | 0 | 7±12 | — | 6±10 | 713±201 (7) | 848±1 (2) | Failed | 68±50 | 0 |
| 9 | 2 | 36±34 | 0.686±0.052 | 28±28 | 559±138 (10) | 812±96 (7) | 1015±0 (1) | 29±28 | 0 |
| 9 | 3 | 81±89 | 0.668±0.102 | 52±52 | 598±186 (10) | 732±159 (7) | 826±49 (3) | 23±19 | 0 |
| 9 | 4 | 158±103 | 0.723±0.041 | 104±63 | 432±104 (10) | 639±115 (10) | 868±106 (7) | 60±78 | 0 |
| 9 | 5 | 91±66 | 0.707±0.036 | 57±35 | 453±194 (10) | 763±131 (10) | 928±65 (4) | 40±29 | 0 |
| 10 | 0 | 2±3 | — | 2±3 | 768±107 (5) | 1003±0 (1) | Failed | 93±97 | 0 |
| 10 | 2 | 55±54 | 0.722±0.027 | 44±40 | 559±156 (10) | 807±149 (10) | 836±0 (1) | 26±39 | 0 |
| 10 | 3 | 86±46 | 0.705±0.063 | 67±36 | 478±143 (10) | 678±114 (9) | 962±33 (4) | 41±50 | 0 |
| 10 | 4 | 99±77 | 0.705±0.048 | 63±43 | 474±162 (10) | 693±91 (9) | 944±113 (4) | 58±86 | 0 |
| 10 | 5 | 110±100 | 0.715±0.039 | 80±78 | 430±164 (10) | 750±142 (10) | 881±107 (4) | 57±55 | 0 |

Table 19: Beam Enumeration batch size 64 with Scaffold and Minimum Size 15. Filter Limit is the number of times that no SMILES contained the pool substructure in 100,000 generation epochs. Patience N/A indicates just Augmented Memory and no Beam Enumeration.

| Patience | Aug. Rounds | Yield | IntDiv1 | Scaffolds | OB 1 | OB 10 | OB 100 | Repeats | Filter Limit |
|---|---|---|---|---|---|---|---|---|---|
| N/A | 0 | 0±0 | — | 0±0 | 584±251 (5) | Failed | Failed | 1±1 | N/A |
| N/A | 2 | 15±9 | 0.775±0.073 | 15±9 | 644±173 (10) | 941±58 (8) | Failed | 0±0 | N/A |
| N/A | 3 | 33±42 | 0.788±0.043 | 32±40 | 613±96 (10) | 927±128 (9) | 993±0 (1) | 0±0 | N/A |
| N/A | 4 | 32±16 | 0.813±0.024 | 31±16 | 527±198 (10) | 880±90 (10) | Failed | 0±0 | N/A |
| N/A | 5 | 40±14 | 0.812±0.023 | 39±13 | 459±177 (10) | 862±68 (10) | Failed | 0±0 | N/A |
| 5 | 0 | 5±17 | 0.726±0.0 | 5±15 | 653±275 (3) | 819±0 (1) | Failed | 48±31 | 0 |
| 5 | 2 | 14±22 | 0.616±0.182 | 13±21 | 635±226 (7) | 850±131 (3) | Failed | 36±29 | 0 |
| 5 | 3 | 21±26 | 0.675±0.116 | 18±22 | 647±198 (8) | 852±88 (5) | Failed | 19±26 | 0 |
| 5 | 4 | 20±30 | 0.6±0.122 | 18±26 | 592±262 (9) | 869±108 (4) | 1038±0 (1) | 28±19 | 0 |
| 5 | 5 | 33±27 | 0.692±0.082 | 29±25 | 506±208 (10) | 875±101 (8) | Failed | 33±37 | 0 |
| 6 | 0 | 0±1 | 0.399±0.0 | 0±0 | 433±98 (4) | Failed | Failed | 98±99 | 0 |
| 6 | 2 | 9±16 | 0.656±0.072 | 7±13 | 713±237 (8) | 864±82 (2) | Failed | 30±25 | 0 |
| 6 | 3 | 16±19 | 0.645±0.072 | 14±18 | 662±152 (8) | 905±103 (5) | Failed | 27±30 | 0 |
| 6 | 4 | 15±23 | 0.644±0.069 | 14±22 | 466±185 (8) | 884±137 (4) | Failed | 23±16 | 0 |
| 6 | 5 | 24±28 | 0.599±0.139 | 21±22 | 583±293 (10) | 849±83 (5) | 1014±0 (1) | 35±38 | 0 |
| 7 | 0 | 0±1 | — | 0±1 | 459±139 (4) | Failed | Failed | 82±47 | 0 |
| 7 | 2 | 10±10 | 0.64±0.072 | 9±10 | 666±180 (9) | 911±76 (3) | Failed | 37±59 | 0 |
| 7 | 3 | 27±31 | 0.659±0.119 | 23±23 | 648±153 (9) | 880±122 (7) | 1041±0 (1) | 11±8 | 0 |
| 7 | 4 | 20±19 | 0.634±0.125 | 19±18 | 575±249 (10) | 853±72 (5) | Failed | 46±59 | 0 |
| 7 | 5 | 14±13 | 0.676±0.096 | 12±10 | 519±267 (10) | 932±75 (6) | Failed | 24±32 | 0 |
| 8 | 0 | 0±0 | — | 0±0 | 383±53 (3) | Failed | Failed | 36±23 | 0 |
| 8 | 2 | 10±13 | 0.665±0.131 | 10±12 | 654±201 (8) | 910±85 (4) | Failed | 15±19 | 0 |
| 8 | 3 | 30±48 | 0.693±0.031 | 29±46 | 624±164 (9) | 863±129 (6) | 901±0 (1) | 24±21 | 0 |
| 8 | 4 | 29±43 | 0.667±0.095 | 23±30 | 571±268 (9) | 745±98 (4) | 981±0 (1) | 20±26 | 0 |
| 8 | 5 | 40±47 | 0.665±0.093 | 35±45 | 450±168 (10) | 879±95 (9) | 920±0 (1) | 43±74 | 0 |
| 9 | 0 | 0±0 | — | 0±0 | 500±207 (4) | Failed | Failed | 31±29 | 0 |
| 9 | 2 | 20±36 | 0.683±0.055 | 19±36 | 683±226 (9) | 825±84 (3) | 1005±0 (1) | 8±9 | 0 |
| 9 | 3 | 41±34 | 0.675±0.08 | 34±28 | 654±155 (10) | 849±134 (8) | Failed | 25±22 | 0 |
| 9 | 4 | 16±14 | 0.647±0.093 | 13±11 | 573±240 (10) | 917±39 (5) | Failed | 10±11 | 0 |
| 9 | 5 | 39±24 | 0.707±0.083 | 34±22 | 456±172 (10) | 829±67 (9) | Failed | 8±9 | 0 |
| 10 | 0 | 3±8 | — | 3±7 | 519±171 (5) | 851±0 (1) | Failed | 16±26 | 0 |
| 10 | 2 | 16±19 | 0.674±0.07 | 13±15 | 599±144 (9) | 905±95 (5) | Failed | 17±20 | 0 |
| 10 | 3 | 32±38 | 0.703±0.074 | 26±27 | 621±107 (10) | 861±129 (8) | 961±0 (1) | 5±7 | 0 |
| 10 | 4 | 18±15 | 0.682±0.087 | 16±15 | 529±202 (10) | 876±81 (7) | Failed | 5±8 | 0 |
| 10 | 5 | 37±31 | 0.711±0.057 | 30±20 | 456±172 (10) | 829±68 (8) | 996±0 (1) | 23±42 | 0 |

**Observations.** Across batch sizes = [64, 32] and all Patience = [5, 6, 7, 8, 9, 10], sample efficiency does not improve in a statistically significant manner (Tables 19 and 20). Variance decreases relative to "Structure" which is in agreement with the hypothesis that "Structure" is more biased.

C.4.3   HYPOTHESIS 3

In the original Beam Enumeration (Guo & Schwaller, 2024b) work, enforcing a Structure Minimum Size for extracted substructures improves sample efficiency across all hyperparameter combinations (and is statistically significant). The results so far suggest that this observation does not hold when optimizing under a particularly low oracle budget (1000 calls). Thus far, experiments were aimed at mitigating the Beam Enumeration bias either by tuning the Patience parameter or by changing the Substructure Type. Another method to mitigate bias is by not enforcing a Structure Minimum Size. In this scenario, Scaffold substructure should be used as Structure substructure tends to extract small functional groups (as observed in the original work).

**Proposed solution.** Investigate "Scaffold" substructure without enforcing Structure Minimum Size.

**Observations.** Across batch sizes = [64, 32] and all Patience = [5, 6, 7, 8, 9, 10], sample efficiency *sometimes* improves (Tables 21 and 22). Variance is also manageable but the performance improvements, when observed, is much less than with lower batch size and higher augmentation rounds (for instance Mamba batch size 16 and augmentation rounds 10).

**Conclusions.** Based on the grid-search results, Beam Enumeration can *sometimes* improve sample efficiency when using "Scaffold" structure and without enforcing Structure Minimum Size. However, the improvements are minor, such that it would be better to use small batch sizes with high augmentation rounds. Thus, we do not further experiment with Beam Enumeration in this work.

Table 20: Beam Enumeration batch size 32 with Scaffold and Minimum Size 15. Filter Limit is the number of times that no SMILES contained the pool substructure in 100,000 generation epochs. Patience N/A indicates just Augmented Memory and no Beam Enumeration.

| Patience | Aug. Rounds | Yield | IntDiv1 | Scaffolds | OB 1 | OB 10 | OB 100 | Repeats | Filter Limit |
|---|---|---|---|---|---|---|---|---|---|
| N/A | 0 | 0±0 | — | 0±0 | 798±101 (5) | Failed | Failed | 1±1 | N/A |
| N/A | 2 | 43±25 | 0.825±0.029 | 42±24 | 608±151 (10) | 844±90 (9) | Failed | 0±0 | N/A |
| N/A | 3 | 52±34 | 0.81±0.059 | 51±32 | 522±141 (10) | 789±100 (9) | 1018±0 (2) | 0±1 | N/A |
| N/A | 4 | 87±33 | 0.82±0.018 | 83±31 | 466±120 (10) | 740±77 (10) | 987±30 (4) | 1±3 | N/A |
| N/A | 5 | 98±57 | 0.817±0.027 | 89±50 | 408±184 (10) | 714±136 (10) | 915±20 (4) | 1±2 | N/A |
| 5 | 0 | 0±0 | — | 0±0 | 852±141 (2) | Failed | Failed | 119±78 | 0 |
| 5 | 2 | 25±38 | 0.65±0.109 | 23±35 | 698±191 (8) | 779±127 (4) | 959±0 (1) | 57±67 | 0 |
| 5 | 3 | 33±59 | 0.629±0.073 | 26±44 | 636±148 (8) | 867±133 (6) | 871±0 (1) | 88±123 | 1 |
| 5 | 4 | 57±68 | 0.666±0.032 | 44±51 | 648±163 (9) | 834±128 (7) | 952±70 (3) | 118±104 | 0 |
| 5 | 5 | 50±69 | 0.649±0.038 | 33±39 | 498±268 (9) | 855±170 (8) | 890±3 (2) | 89±46 | 0 |
| 6 | 0 | 2±6 | — | 2±6 | 788±161 (3) | 840±0 (1) | Failed | 174±112 | 0 |
| 6 | 2 | 25±59 | 0.618±0.148 | 16±36 | 672±240 (7) | 694±238 (3) | 706±0 (1) | 53±55 | 1 |
| 6 | 3 | 35±47 | 0.667±0.119 | 27±35 | 702±189 (8) | 789±93 (5) | 974±0 (2) | 52±43 | 0 |
| 6 | 4 | 46±66 | 0.653±0.068 | 39±56 | 656±127 (9) | 831±144 (6) | 945±67 (2) | 135±206 | 0 |
| 6 | 5 | 57±76 | 0.584±0.157 | 45±59 | 571±274 (8) | 668±83 (4) | 907±7 (3) | 101±113 | 0 |
| 7 | 0 | 14±27 | 0.551±0.116 | 10±17 | 663±109 (5) | 814±130 (3) | Failed | 106±58 | 0 |
| 7 | 2 | 19±41 | 0.657±0.121 | 12±24 | 660±127 (6) | 894±136 (5) | 929±0 (1) | 34±23 | 0 |
| 7 | 3 | 38±51 | 0.636±0.115 | 28±30 | 650±161 (10) | 812±131 (6) | 863±0 (1) | 45±33 | 0 |
| 7 | 4 | 36±36 | 0.652±0.109 | 26±21 | 700±151 (10) | 811±76 (7) | 981±0 (1) | 67±49 | 0 |
| 7 | 5 | 46±45 | 0.608±0.108 | 39±40 | 485±204 (9) | 810±50 (6) | 991±5 (2) | 237±244 | 0 |
| 8 | 0 | 0±0 | — | 0±0 | 794±302 (4) | Failed | Failed | 149±100 | 0 |
| 8 | 2 | 34±45 | 0.625±0.105 | 30±39 | 696±175 (9) | 777±105 (5) | 901±0 (1) | 57±46 | 0 |
| 8 | 3 | 53±77 | 0.543±0.174 | 42±61 | 652±213 (9) | 715±141 (5) | 836±6 (2) | 57±87 | 1 |
| 8 | 4 | 30±53 | 0.631±0.092 | 24±39 | 684±235 (9) | 781±165 (3) | 957±51 (2) | 54±43 | 0 |
| 8 | 5 | 90±101 | 0.632±0.124 | 70±74 | 556±248 (9) | 706±127 (6) | 879±78 (4) | 179±158 | 0 |
| 9 | 0 | 0±0 | — | 0±0 | 733±157 (3) | Failed | Failed | 175±142 | 0 |
| 9 | 2 | 20±37 | 0.61±0.124 | 15±25 | 643±237 (8) | 849±152 (4) | 967±0 (1) | 61±69 | 0 |
| 9 | 3 | 28±25 | 0.639±0.09 | 23±20 | 661±121 (10) | 819±78 (6) | Failed | 53±60 | 0 |
| 9 | 4 | 67±63 | 0.66±0.105 | 55±56 | 605±203 (9) | 783±126 (8) | 906±58 (2) | 92±65 | 0 |
| 9 | 5 | 55±73 | 0.618±0.13 | 36±41 | 513±225 (9) | 779±149 (6) | 877±74 (2) | 150±206 | 0 |
| 10 | 0 | 2±5 | — | 1±3 | 835±154 (4) | 890±0 (1) | Failed | 93±68 | 0 |
| 10 | 2 | 5±4 | — | 4±3 | 680±196 (8) | 960±0 (1) | Failed | 58±52 | 0 |
| 10 | 3 | 32±48 | 0.636±0.143 | 31±47 | 572±171 (10) | 880±130 (7) | 900±0 (1) | 30±36 | 0 |
| 10 | 4 | 44±32 | 0.693±0.059 | 34±26 | 503±195 (10) | 811±126 (9) | 965±0 (1) | 107±125 | 0 |
| 10 | 5 | 51±55 | 0.581±0.206 | 36±37 | 584±317 (9) | 712±88 (5) | 949±34 (2) | 156±239 | 1 |

Table 21: Beam Enumeration batch size 64 with Scaffold and no Minimum Size enforced. Filter Limit is the number of times that no SMILES contained the pool substructure in 100,000 generation epochs. Patience N/A indicates just Augmented Memory and no Beam Enumeration.

| Patience | Aug. Rounds | Yield | IntDiv1 | Scaffolds | OB 1 | OB 10 | OB 100 | Repeats | Filter Limit |
|---|---|---|---|---|---|---|---|---|---|
| N/A | 0 | 0±0 | — | 0±0 | 584±251 (5) | Failed | Failed | 1±1 | 0 |
| N/A | 2 | 15±9 | 0.775±0.073 | 15±9 | 644±173 (10) | 941±58 (8) | Failed | 0±0 | 0 |
| N/A | 3 | 33±42 | 0.788±0.043 | 32±40 | 613±96 (10) | 927±128 (9) | 993±0 (1) | 0±0 | 0 |
| N/A | 4 | 32±16 | 0.813±0.024 | 31±16 | 527±198 (10) | 880±90 (10) | Failed | 0±0 | 0 |
| N/A | 5 | 40±14 | 0.812±0.023 | 39±13 | 459±177 (10) | 862±68 (10) | Failed | 0±0 | 0 |
| 5 | 0 | 0±0 | — | 0±0 | 307±0 (1) | Failed | Failed | 0±0 | 0 |
| 5 | 2 | 15±12 | 0.744±0.068 | 14±11 | 678±227 (10) | 930±70 (5) | Failed | 0±0 | 0 |
| 5 | 3 | 38±14 | 0.791±0.026 | 37±14 | 552±70 (10) | 824±44 (9) | Failed | 0±0 | 0 |
| 5 | 4 | 43±45 | 0.791±0.021 | 42±43 | 516±230 (10) | 839±132 (9) | 918±0 (1) | 0±0 | 0 |
| 5 | 5 | 55±33 | 0.77±0.073 | 50±30 | 467±197 (10) | 811±81 (9) | 961±0 (1) | 0±1 | 0 |
| 6 | 0 | 0±0 | — | 0±0 | 594±268 (5) | Failed | Failed | 0±0 | 0 |
| 6 | 2 | 28±23 | 0.752±0.053 | 26±21 | 671±190 (10) | 880±72 (6) | Failed | 0±0 | 0 |
| 6 | 3 | 44±28 | 0.782±0.032 | 42±24 | 584±120 (10) | 832±64 (9) | 1006±0 (1) | 0±0 | 0 |
| 6 | 4 | 41±37 | 0.778±0.028 | 39±36 | 571±241 (10) | 874±118 (9) | 959±0 (1) | 0±0 | 0 |
| 6 | 5 | 54±21 | 0.794±0.025 | 49±17 | 453±169 (10) | 827±72 (10) | Failed | 0±0 | 0 |
| 7 | 0 | 0±0 | — | 0±0 | 567±234 (5) | Failed | Failed | 0±1 | 0 |
| 7 | 2 | 27±13 | 0.778±0.072 | 27±13 | 603±148 (10) | 880±80 (9) | Failed | 0±0 | 0 |
| 7 | 3 | 47±33 | 0.797±0.027 | 44±30 | 586±73 (10) | 859±113 (10) | 1035±1 (2) | 0±0 | 0 |
| 7 | 4 | 48±23 | 0.799±0.017 | 45±20 | 498±176 (10) | 828±87 (10) | Failed | 0±0 | 0 |
| 7 | 5 | 51±23 | 0.793±0.023 | 48±21 | 463±190 (10) | 854±72 (10) | Failed | 0±0 | 0 |
| 8 | 0 | 0±0 | — | 0±0 | 383±53 (3) | Failed | Failed | 0±0 | 0 |
| 8 | 2 | 20±12 | 0.755±0.072 | 20±12 | 637±153 (10) | 929±62 (8) | Failed | 0±0 | 0 |
| 8 | 3 | 39±32 | 0.793±0.021 | 38±31 | 593±85 (10) | 882±111 (10) | 962±0 (1) | 0±0 | 0 |
| 8 | 4 | 47±30 | 0.793±0.024 | 45±29 | 544±208 (10) | 873±75 (10) | 1013±0 (1) | 0±0 | 0 |
| 8 | 5 | 69±28 | 0.803±0.019 | 64±22 | 446±162 (10) | 789±73 (10) | 991±0 (1) | 0±0 | 0 |
| 9 | 0 | 0±0 | — | 0±0 | 656±281 (6) | Failed | Failed | 0±0 | 0 |
| 9 | 2 | 16±10 | 0.761±0.041 | 16±10 | 640±166 (10) | 946±48 (6) | Failed | 0±0 | 0 |
| 9 | 3 | 52±60 | 0.798±0.021 | 49±55 | 619±106 (10) | 847±107 (10) | 847±0 (1) | 0±0 | 0 |
| 9 | 4 | 50±25 | 0.802±0.01 | 48±22 | 505±177 (10) | 846±79 (10) | 1004±0 (1) | 0±0 | 0 |
| 9 | 5 | 54±26 | 0.792±0.024 | 50±24 | 450±165 (10) | 809±55 (9) | Failed | 0±0 | 0 |
| 10 | 0 | 0±0 | — | 0±0 | 636±260 (6) | Failed | Failed | 0±0 | 0 |
| 10 | 2 | 21±17 | 0.739±0.091 | 21±17 | 643±178 (10) | 920±78 (8) | Failed | 0±0 | 0 |
| 10 | 3 | 46±48 | 0.791±0.024 | 43±43 | 613±99 (10) | 853±115 (9) | 899±0 (1) | 0±0 | 0 |
| 10 | 4 | 44±35 | 0.783±0.041 | 42±33 | 541±222 (10) | 858±89 (9) | 990±0 (1) | 0±0 | 0 |
| 10 | 5 | 48±18 | 0.792±0.024 | 45±15 | 456±173 (10) | 853±50 (10) | Failed | 0±0 | 0 |

Table 22: Beam Enumeration batch size 32 with Scaffold and no Minimum Size enforced. Filter Limit is the number of times that no SMILES contained the pool substructure in 100,000 generation epochs. Patience N/A indicates just Augmented Memory and no Beam Enumeration.

| Patience | Aug. Rounds | Yield | IntDiv1 | Scaffolds | OB 1 | OB 10 | OB 100 | Repeats | Filter Limit |
|---|---|---|---|---|---|---|---|---|---|
| N/A | 0 | 0±0 | — | 0±0 | 798±101 (5) | Failed | Failed | 1±1 | 0 |
| N/A | 2 | 43±25 | 0.825±0.029 | 42±24 | 608±151 (10) | 844±90 (9) | Failed | 0±0 | 0 |
| N/A | 3 | 52±34 | 0.81±0.059 | 51±32 | 522±141 (10) | 789±100 (9) | 1018±0 (2) | 0±1 | 0 |
| N/A | 4 | 87±33 | 0.82±0.018 | 83±31 | 466±120 (10) | 740±77 (10) | 987±30 (4) | 1±3 | 0 |
| N/A | 5 | 98±57 | 0.817±0.027 | 89±50 | 408±184 (10) | 714±136 (10) | 915±20 (4) | 1±2 | 0 |
| 5 | 0 | 0±1 | — | 0±1 | 783±134 (3) | Failed | Failed | 0±1 | 0 |
| 5 | 2 | 38±28 | 0.796±0.03 | 35±25 | 504±111 (9) | 828±115 (9) | Failed | 1±1 | 0 |
| 5 | 3 | 63±44 | 0.762±0.073 | 57±38 | 593±170 (10) | 763±82 (8) | 988±29 (3) | 1±2 | 0 |
| 5 | 4 | 87±57 | 0.779±0.038 | 72±43 | 540±145 (10) | 764±139 (10) | 958±48 (5) | 2±4 | 0 |
| 5 | 5 | 106±61 | 0.784±0.031 | 84±41 | 467±187 (10) | 718±109 (10) | 960±41 (6) | 1±2 | 0 |
| 6 | 0 | 1±3 | — | 1±3 | 837±135 (3) | 998±0 (1) | Failed | 2±2 | 0 |
| 6 | 2 | 40±33 | 0.761±0.078 | 36±29 | 609±149 (9) | 811±64 (7) | 1014±0 (1) | 1±2 | 0 |
| 6 | 3 | 49±23 | 0.796±0.03 | 46±21 | 585±104 (10) | 839±101 (10) | Failed | 1±2 | 0 |
| 6 | 4 | 57±41 | 0.783±0.031 | 53±37 | 557±187 (10) | 771±82 (8) | 987±10 (3) | 1±2 | 0 |
| 6 | 5 | 106±85 | 0.776±0.05 | 85±55 | 508±241 (10) | 718±151 (9) | 927±94 (5) | 3±6 | 0 |
| 7 | 0 | 0±0 | — | 0±0 | 741±222 (5) | Failed | Failed | 1±1 | 0 |
| 7 | 2 | 43±27 | 0.79±0.037 | 41±26 | 631±182 (10) | 799±77 (8) | Failed | 0±0 | 0 |
| 7 | 3 | 84±67 | 0.79±0.021 | 73±56 | 578±188 (10) | 781±117 (9) | 937±42 (4) | 0±1 | 0 |
| 7 | 4 | 74±43 | 0.785±0.041 | 69±37 | 574±149 (10) | 789±111 (10) | 948±39 (2) | 1±3 | 0 |
| 7 | 5 | 121±52 | 0.786±0.033 | 105±39 | 422±155 (10) | 673±90 (10) | 898±52 (5) | 4±9 | 0 |
| 8 | 0 | 3±5 | — | 3±5 | 683±213 (5) | 882±0 (1) | Failed | 2±3 | 0 |
| 8 | 2 | 44±39 | 0.713±0.166 | 40±30 | 629±177 (10) | 778±97 (7) | 995±0 (1) | 1±4 | 0 |
| 8 | 3 | 69±43 | 0.794±0.039 | 65±40 | 530±183 (10) | 778±104 (9) | 975±8 (3) | 0±2 | 0 |
| 8 | 4 | 75±39 | 0.795±0.033 | 66±30 | 547±142 (10) | 770±118 (10) | 981±29 (3) | 1±1 | 0 |
| 8 | 5 | 103±55 | 0.761±0.091 | 90±49 | 488±221 (10) | 693±142 (9) | 961±39 (7) | 4±5 | 0 |
| 9 | 0 | 2±4 | — | 2±4 | 805±127 (4) | 915±0 (1) | Failed | 1±1 | 0 |
| 9 | 2 | 41±23 | 0.79±0.022 | 40±22 | 572±132 (10) | 839±95 (10) | Failed | 0±0 | 0 |
| 9 | 3 | 59±34 | 0.81±0.021 | 54±31 | 520±110 (9) | 778±68 (9) | 993±0 (1) | 0±1 | 0 |
| 9 | 4 | 101±60 | 0.799±0.025 | 89±45 | 515±142 (10) | 725±104 (10) | 944±91 (4) | 1±1 | 0 |
| 9 | 5 | 128±61 | 0.792±0.022 | 102±41 | 425±179 (10) | 684±93 (10) | 919±51 (6) | 2±2 | 0 |
| 10 | 0 | 0±1 | — | 0±1 | 822±160 (4) | Failed | Failed | 1±1 | 0 |
| 10 | 2 | 53±45 | 0.795±0.025 | 49±44 | 515±129 (9) | 793±106 (9) | 973±30 (2) | 2±5 | 0 |
| 10 | 3 | 86±63 | 0.759±0.119 | 73±46 | 553±179 (10) | 720±62 (8) | 956±69 (4) | 0±1 | 0 |
| 10 | 4 | 89±35 | 0.794±0.034 | 77±26 | 464±132 (10) | 743±51 (10) | 984±27 (4) | 3±5 | 0 |
| 10 | 5 | 123±58 | 0.795±0.031 | 105±44 | 434±177 (10) | 704±102 (10) | 949±59 (8) | 2±2 | 0 |

### C.5 Hallucinated Memory: Is it beneficial to allocate a portion of the oracle budget to hallucination?

In this section, we investigate coupling GraphGA (Jensen, 2019) to Saturn. GraphGA in itself a sample-efficient generative algorithm (Gao et al., 2022) and was recently used in the GEAM model proposed by Lee et al. (Lee et al., 2024) which achieves impressive MPO performance. Previously work (Liu et al., 2021) found that coupling a GA in RL can encourage diverse sampling. In the previous sections, we have identified Mamba with batch size 16 and 10 augmentation rounds as the best hyperparameters so far. The improved sample efficiency comes at a trade-off in diversity. The objective in the experiments to follow is to investigate whether allocating a portion of the oracle budget to GraphGA generation (which we call "hallucinating") is beneficial in recovering diversity while maintaining sample efficiency.

Before presenting the grid-search results, we describe the GraphGA integration further. GraphGA is only activated when the replay buffer is full (100 SMILES). Once full, at every epoch thereafter, the replay buffer itself is treated as the parent population to generate new SMILES. These new SMILES are then concatenated with the sampled batch (16 SMILES) and used to update the Agent. Importantly, these hallucinated SMILES are also deposited into the replay buffer (if they possess higher reward). Finally, 100 SMILES are hallucinated and either 5 or 10 are selected. The selection criteria are **Random** or **Tanimoto Distance**. Random selects at random while Tanimoto Distance selects via maximum fingerprint *dissimilarity* to the replay buffer. Our rationale is that dissimilar new SMILES will help encourage diversity since Augmented Memory heavily biases towards the replay buffer SMILES.

**The grid-search investigated the following hyperparameter settings**:

1. Fix Mamba with batch size 16
2. Augmentation Rounds = [5,20]
3. GA with Random and Tanimoto Distance selection criterion
4. Select 5 or 10 hallucinations at every epoch

The reason we increased the augmentation rounds back to 20 in our grid-search is because if indeed the GA recovers diversity, then the "augmentation tolerability" of Saturn would probably be increased. Higher augmentation rounds lead to more repeated SMILES precisely due to overfitting. If new high reward SMILES *refresh* the replay buffer, Saturn may be more tolerable to higher augmentation rounds to potentially further improve sample efficiency. The results of the grid-search are presented in Tables 23 and 24.

**Observations.** The results show that coupling a GA to the replay buffer does not improve sample efficiency. However, we make several interesting observations. Firstly, the number of repeated SMILES *notably* drops and IntDiv1 (Polykovskiy et al., 2020) recovers. This is in agreement with our hypothesis and previous work (Liu et al., 2021) that coupling a GA to RL can recover diversity. Secondly, hallucinating SMILES does indeed lead to some replacement of the replay buffer, and hence, these SMILES are necessarily are high reward. Thirdly, rarely are the hallucinated SMILES the best in the buffer. Finally, we note that hallucinated SMILES are generated off-policy and Agent updates may be more meaningful with importance sampling (Schlegel et al., 2019), which we did not explore this this work.

### C.6 Saturn: Final Hyperparameters

The most sample-efficient hyperparameter settings, on average, are: **Mamba with batch size 16 and 10 augmentation rounds**. The results in the immediate previous section shows that the GA can recover diversity, which can be a useful setting that can easily be activated on and off depending on the oracle setting.

## D Mechanism of Augmented Memory and Mamba

In this subsection, we show additional results supporting our statement on Augmented Memory's (Guo & Schwaller, 2024a) mechanism: Augmented Memory squeezes the likelihood of generating

Table 23: Mamba batch size 16 with GraphGA (Jensen, 2019) applied on the replay buffer. The hallucinated SMILES were selected at *Random*. **Hall. Yield** is the yield from GraphGA. **Buf. Replace** is the number of times a hallucinated SMILES replaced another SMILES in the buffer. This means that it was better than the top-100 SMILES generated in the run so far. **Buf. Best** is the number of times the hallucinated SMILES was better than the top-1 in the buffer.

| GA Random | Aug. Rounds | Hall. Yield | Total Yield | Buffer Replace | Buffer Best | IntDiv1 | Scaffolds | OB 1 | OB 10 | OB 100 | Sampled Repeats | Hall. Repeats |
|---|---|---|---|---|---|---|---|---|---|---|---|---|
| 5 | 5 | 9±7 | 54±43 | 91±13 | 2±1 | 0.756±0.043 | 45±33 | 538±212 (10) | 812±114 (9) | 989±27 (3) | 58±39 | 5±3 |
| 5 | 6 | 21±10 | 88±56 | 92±11 | 3±1 | 0.773±0.046 | 68±41 | 457±122 (10) | 729±103 (10) | 936±83 (3) | 57±29 | 6±3 |
| 5 | 7 | 11±9 | 57±42 | 90±17 | 3±2 | 0.73±0.063 | 49±37 | 619±125 (10) | 795±116 (9) | 988±13 (3) | 122±50 | 6±3 |
| 5 | 8 | 14±11 | 63±42 | 95±15 | 3±2 | 0.758±0.044 | 49±25 | 574±166 (10) | 793±96 (10) | 916±0 (1) | 177±80 | 6±3 |
| 5 | 9 | 20±15 | 106±75 | 92±14 | 2±1 | 0.767±0.03 | 86±55 | 531±128 (10) | 733±121 (10) | 833±57 (3) | 207±101 | 9±5 |
| 5 | 10 | 21±11 | 113±61 | 93±19 | 2±1 | 0.742±0.04 | 83±38 | 496±158 (10) | 690±118 (10) | 910±59 (5) | 257±143 | 7±3 |
| 5 | 11 | 15±11 | 102±69 | 89±13 | 3±2 | 0.739±0.031 | 69±43 | 552±141 (10) | 730±116 (10) | 887±62 (4) | 308±116 | 7±3 |
| 5 | 12 | 29±17 | 139±83 | 101±13 | 3±1 | 0.781±0.025 | 101±55 | 488±104 (10) | 666±92 (10) | 856±76 (5) | 339±153 | 9±4 |
| 5 | 13 | 25±14 | 144±97 | 97±15 | 3±1 | 0.727±0.048 | 94±50 | 463±209 (10) | 658±155 (10) | 843±99 (6) | 511±226 | 10±4 |
| 5 | 14 | 36±22 | 176±82 | 102±18 | 3±2 | 0.742±0.038 | 133±56 | 475±121 (10) | 640±110 (10) | 863±82 (8) | 691±333 | 13±7 |
| 5 | 15 | 42±17 | 208±65 | 104±18 | 4±2 | 0.746±0.06 | 167±58 | 401±115 (10) | 595±89 (10) | 844±91 (10) | 693±319 | 13±8 |
| 5 | 16 | 34±9 | 187±77 | 100±20 | 5±2 | 0.744±0.055 | 150±59 | 421±119 (10) | 624±106 (10) | 829±83 (8) | 789±465 | 10±5 |
| 5 | 17 | 33±25 | 181±95 | 99±14 | 3±1 | 0.750±0.042 | 127±64 | 469±142 (10) | 664±132 (10) | 838±86 (8) | 830±417 | 10±6 |
| 5 | 18 | 35±18 | 164±57 | 102±24 | 4±2 | 0.727±0.038 | 133±54 | 459±105 (10) | 637±76 (10) | 872±66 (8) | 881±389 | 16±16 |
| 5 | 19 | 30±16 | 190±76 | 103±16 | 3±1 | 0.744±0.046 | 145±51 | 467±123 (10) | 630±113 (10) | 822±59 (8) | 1072±465 | 12±9 |
| 5 | 20 | 44±18 | 247±83 | 96±10 | 3±1 | 0.748±0.034 | 185±60 | 380±144 (10) | 566±115 (10) | 761±59 (9) | 1310±512 | 14±6 |
| 10 | 5 | 12±10 | 44±44 | 141±13 | 3±1 | 0.77±0.066 | 35±29 | 478±206 (10) | 802±133 (9) | 888±0 (1) | 24±14 | 8±5 |
| 10 | 6 | 16±13 | 44±34 | 139±7 | 4±2 | 0.784±0.023 | 37±29 | 534±139 (10) | 812±87 (9) | 936±0 (1) | 38±19 | 8±4 |
| 10 | 7 | 14±9 | 43±27 | 139±23 | 4±2 | 0.739±0.109 | 37±23 | 594±117 (10) | 800±54 (9) | Failed | 61±34 | 9±4 |
| 10 | 8 | 20±16 | 55±41 | 148±13 | 4±2 | 0.771±0.026 | 46±30 | 520±114 (10) | 805±129 (10) | 924±0 (1) | 71±30 | 9±4 |
| 10 | 9 | 22±18 | 70±51 | 143±19 | 4±2 | 0.753±0.04 | 57±42 | 520±174 (10) | 788±149 (10) | 952±44 (3) | 113±58 | 11±7 |
| 10 | 10 | 17±16 | 65±63 | 148±19 | 4±2 | 0.714±0.104 | 48±37 | 539±183 (10) | 758±141 (9) | 773±0 (1) | 138±69 | 11±6 |
| 10 | 11 | 18±11 | 57±47 | 140±21 | 5±1 | 0.761±0.031 | 42±29 | 605±139 (10) | 789±104 (9) | 931±38 (2) | 192±90 | 10±7 |
| 10 | 12 | 37±37 | 88±79 | 165±26 | 4±1 | 0.734±0.092 | 70±59 | 591±142 (10) | 716±119 (9) | 882±110 (3) | 222±106 | 17±14 |
| 10 | 13 | 29±25 | 84±84 | 150±22 | 4±1 | 0.727±0.078 | 61±51 | 502±195 (10) | 737±169 (9) | 842±52 (3) | 260±134 | 13±7 |
| 10 | 14 | 29±16 | 97±64 | 149±14 | 5±2 | 0.756±0.046 | 72±44 | 456±217 (10) | 733±164 (10) | 908±9 (5) | 271±116 | 9±6 |
| 10 | 15 | 37±24 | 102±64 | 161±13 | 4±1 | 0.759±0.03 | 85±48 | 480±184 (10) | 688±162 (10) | 913±77 (5) | 336±182 | 19±10 |
| 10 | 16 | 40±22 | 110±60 | 157±18 | 5±3 | 0.754±0.028 | 91±50 | 432±200 (10) | 691±149 (10) | 913±55 (6) | 361±185 | 15±10 |
| 10 | 17 | 34±22 | 103±62 | 156±28 | 5±2 | 0.75±0.048 | 80±47 | 529±154 (10) | 704±117 (9) | 916±45 (6) | 467±214 | 15±8 |
| 10 | 18 | 25±15 | 91±52 | 148±22 | 5±1 | 0.745±0.03 | 64±31 | 562±102 (10) | 750±88 (10) | 927±42 (4) | 572±322 | 17±10 |
| 10 | 19 | 25±14 | 88±46 | 145±17 | 6±2 | 0.750±0.036 | 71±39 | 563±127 (10) | 751±114 (10) | 948±33 (5) | 603±236 | 16±9 |
| 10 | 20 | 38±24 | 136±80 | 148±19 | 6±1 | 0.748±0.059 | 95±48 | 444±150 (10) | 626±117 (9) | 867±90 (6) | 781±360 | 13±5 |

Table 24: Mamba batch size 16 with GraphGA (Jensen, 2019) applied on the replay buffer. The hallucinated SMILES were selected by highest *Tanimoto Distance*. **Hall. Yield** is the yield from GraphGA. **Buf. Replace** is the number of times a hallucinated SMILES replaced another SMILES in the buffer. This means that it was better than the top-100 SMILES generated in the run so far. **Buf. Best** is the number of times the hallucinated SMILES was better than the top-1 in the buffer.

| GA Random | Aug. Rounds | Hall. Yield | Total Yield | Buffer Replace | Buffer Best | IntDiv1 | Scaffolds | OB 1 | OB 10 | OB 100 | Sampled Repeats | Hall. Repeats |
|---|---|---|---|---|---|---|---|---|---|---|---|---|
| 5 | 5 | 12±11 | 68±60 | 84±16 | 2±1 | 0.770±0.050 | 57±46 | 532±244 (10) | 752±125 (8) | 913±51 (3) | 50±35 | 17±7 |
| 5 | 6 | 8±8 | 61±73 | 83±13 | 1±1 | 0.763±0.041 | 51±57 | 602±171 (10) | 834±151 (10) | 890±110 (2) | 62±36 | 17±11 |
| 5 | 7 | 15±8 | 68±46 | 90±10 | 4±2 | 0.776±0.035 | 60±38 | 610±62 (10) | 797±86 (10) | 855±0 (1) | 122±59 | 17±8 |
| 5 | 8 | 11±8 | 89±61 | 77±13 | 2±1 | 0.765±0.031 | 72±45 | 473±120 (10) | 753±116 (10) | 888±42 (3) | 156±84 | 14±8 |
| 5 | 9 | 22±17 | 123±86 | 88±8 | 2±1 | 0.757±0.049 | 97±66 | 471±187 (10) | 712±164 (10) | 872±96 (5) | 309±150 | 16±7 |
| 5 | 10 | 18±15 | 97±79 | 87±14 | 2±1 | 0.758±0.045 | 78±57 | 544±183 (10) | 748±158 (10) | 901±107 (4) | 317±133 | 16±9 |
| 5 | 11 | 18±14 | 92±60 | 84±15 | 2±2 | 0.785±0.031 | 78±49 | 560±130 (10) | 749±97 (10) | 846±42 (2) | 314±126 | 20±9 |
| 5 | 12 | 26±17 | 146±101 | 90±10 | 2±1 | 0.772±0.043 | 109±70 | 491±165 (10) | 684±184 (10) | 838±124 (6) | 418±220 | 22±15 |
| 5 | 13 | 21±15 | 114±77 | 90±19 | 2±1 | 0.74±0.053 | 97±62 | 494±200 (10) | 706±134 (9) | 912±71 (6) | 494±218 | 19±13 |
| 5 | 14 | 28±24 | 158±95 | 91±21 | 2±1 | 0.756±0.042 | 131±82 | 505±152 (10) | 681±152 (10) | 846±85 (7) | 682±355 | 27±20 |
| 5 | 15 | 39±20 | 189±98 | 97±8 | 3±1 | 0.752±0.074 | 151±76 | 415±159 (10) | 600±176 (10) | 818±103 (8) | 698±382 | 28±14 |
| 5 | 16 | 45±30 | 189±110 | 100±29 | 2±2 | 0.788±0.042 | 152±91 | 456±171 (10) | 630±168 (10) | 784±98 (7) | 771±329 | 33±16 |
| 5 | 17 | 29±22 | 166±89 | 95±13 | 3±1 | 0.760±0.053 | 124±58 | 506±145 (10) | 652±130 (10) | 874±102 (8) | 733±343 | 26±15 |
| 5 | 18 | 17±12 | 114±75 | 88±16 | 3±2 | 0.686±0.104 | 87±50 | 549±154 (10) | 668±86 (8) | 913±65 (6) | 911±412 | 30±20 |
| 5 | 19 | 16±14 | 117±86 | 73±22 | 2±2 | 0.708±0.101 | 94±70 | 559±169 (10) | 706±153 (9) | 862±117 (5) | 1287±520 | 24±23 |
| 5 | 20 | 32±16 | 183±72 | 85±17 | 3±2 | 0.752±0.072 | 151±60 | 417±161 (10) | 628±111 (10) | 878±102 (10) | 1241±508 | 22±13 |
| 10 | 5 | 13±13 | 39±39 | 127±17 | 3±2 | 0.768±0.065 | 35±34 | 551±214 (9) | 765±155 (7) | 942±0 (1) | 34±15 | 19±8 |
| 10 | 6 | 11±10 | 43±34 | 128±17 | 2±1 | 0.76±0.064 | 41±32 | 556±156 (10) | 777±99 (7) | Failed | 34±20 | 16±8 |
| 10 | 7 | 13±8 | 41±28 | 138±12 | 3±2 | 0.767±0.066 | 38±27 | 550±140 (10) | 835±106 (9) | 997±0 (1) | 62±43 | 19±9 |
| 10 | 8 | 12±9 | 41±26 | 138±13 | 2±2 | 0.751±0.093 | 36±22 | 575±156 (10) | 786±123 (9) | Failed | 75±41 | 21±9 |
| 10 | 9 | 18±12 | 56±35 | 129±20 | 3±2 | 0.764±0.072 | 48±30 | 527±156 (10) | 732±79 (8) | 991±0 (1) | 117±78 | 19±9 |
| 10 | 10 | 10±12 | 42±46 | 133±14 | 3±2 | 0.775±0.055 | 32±31 | 660±225 (10) | 797±127 (7) | 870±0 (1) | 158±80 | 15±7 |
| 10 | 11 | 10±8 | 39±39 | 124±18 | 3±1 | 0.713±0.109 | 32±30 | 626±173 (10) | 828±124 (7) | 964±0 (1) | 181±93 | 30±23 |
| 10 | 12 | 16±19 | 63±64 | 139±18 | 3±1 | 0.733±0.123 | 53±56 | 534±207 (10) | 731±113 (8) | 897±107 (2) | 236±106 | 29±23 |
| 10 | 13 | 20±19 | 67±63 | 140±21 | 3±2 | 0.732±0.117 | 50±41 | 542±228 (9) | 746±139 (8) | 902±38 (3) | 300±150 | 30±19 |
| 10 | 14 | 15±13 | 61±50 | 128±21 | 2±1 | 0.714±0.114 | 49±41 | 589±175 (10) | 770±102 (8) | 924±22 (2) | 365±210 | 26±15 |
| 10 | 15 | 28±25 | 80±71 | 144±22 | 5±1 | 0.762±0.033 | 68±58 | 599±160 (10) | 741±129 (8) | 925±100 (4) | 366±228 | 32±19 |
| 10 | 16 | 30±28 | 89±77 | 152±28 | 5±2 | 0.765±0.07 | 74±63 | 563±186 (10) | 719±167 (9) | 832±34 (3) | 376±188 | 35±24 |
| 10 | 17 | 30±25 | 101±80 | 147±16 | 3±1 | 0.787±0.028 | 77±58 | 532±182 (9) | 719±173 (9) | 880±45 (5) | 503±237 | 42±25 |
| 10 | 18 | 16±13 | 54±39 | 137±33 | 3±2 | 0.721±0.071 | 43±31 | 543±152 (10) | 811±112 (9) | 926±0 (1) | 609±309 | 48±59 |
| 10 | 19 | 21±12 | 83±54 | 129±15 | 3±2 | 0.761±0.034 | 64±41 | 495±135 (9) | 738±121 (9) | 920±40 (4) | 620±259 | 30±17 |
| 10 | 20 | 16±17 | 54±44 | 133±24 | 2±1 | 0.761±0.044 | 46±34 | 524±206 (9) | 796±86 (8) | 925±0 (1) | 747±416 | 32±17 |

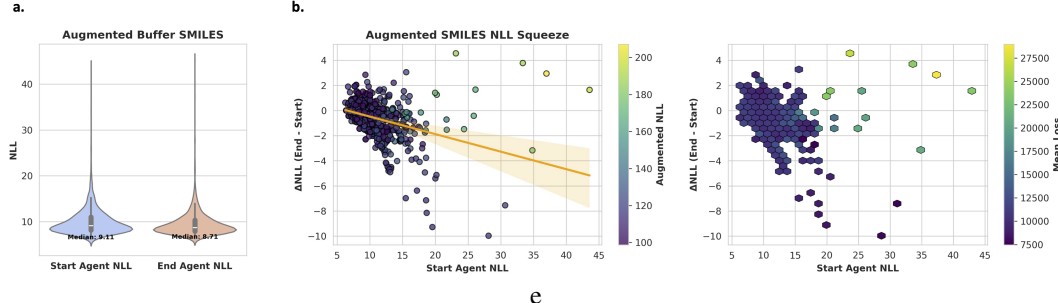

e

Figure D4: Mamba (batch size 16, augmentation rounds 10) after running for 500 oracle calls of the illustrative example and isolating the effect of Augmented Memory. **a.** Augmented Memory makes the likelihood of generating SMILES in the Buffer more likely. **b.** Augmented forms of the Buffer SMILES become more likely, but still regularized by the prior.

the Buffer *molecules* such that it becomes probable to generate *some* SMILES representation of them. In the main text, the experiment to show likelihood squeezing was as follows: starting from the pre-trained Mamba model, generate molecules until the Buffer is full and then save the Agent state before and after Augmented Memory. Every augmented Buffer SMILES was also saved. This experiment isolates the effect of Augmented Memory on a *clean* pre-trained model.

The first set of additional results we show is the same experiment but we first allow the Agent 500 oracle calls of optimization on the test experiment. Our intention is to show that later in the run, Augmented Memory still makes generating the Buffer *molecules* more likely (Fig. D4). There are cases when a large loss magnitude does not make the sequence more likely to be generated. This could occur for instance when the likelihood under the prior is extremely low (large NLL) where the intended behavior is actually to regress the Agent back towards the prior. In these cases, the large loss could make the update less stable for the parameter updates.

Next, the main text results showed that Mamba (batch size 16, augmentation rounds 10) exhibits "hop-and-locally-explore" behavior but what about RNN (batch size 16, augmentation rounds 10)? We show that the RNN model also begins to exhibit this behavior but to a lesser extent (Fig. D5), in agreement with the enhanced likelihood convergence observed for Mamba (Appendix C.1).

We now focus on Mamba (batch size 16, augmentation rounds 10) and present additional results to qualitatively and quantitatively demonstrate "hop-and-locally-explore" behavior. Firstly, we supplement the main text Fig. 2e. The figure shows the intra- and inter-chunk similarities across chunks of generated molecules. Specifically, the test experiment was run with an oracle budget of 3,000 and this generated set is chunked. To provide a more granular inspection into the generative behavior, we chunk this set into 30 chunks (each 100 SMILES) instead of 10 chunks (each 300 SMILES) in the main text. Mamba (batch size 16, augmentation rounds) exhibits notably higher intra-chunk similarity and even inter-chunk similarity at this more granular chunking level (Fig. D6a). We further supplement these quantitative results with a qualitative inspection. Looking at **unique** molecules generated at adjacent epochs, common substructures are shared (Fig. D6b highlights), displaying a "neighborhood-like" exploration.

### D.1 IS "HOP-AND-LOCALLY-EXPLORE" *Always* GOOD?

The results in the main text and this section so far provide evidence that Mamba with batch size 16 and 10 augmentation rounds exhibits local exploration behavior. We hypothesize that sample efficiency improves because "similar molecules, on average, exhibit similar properties". But is this always true? In the test experiment, it is straightforward to see that this indeed holds true. Cross-referencing Fig. D6b, small changes to the molecular graphs should still display high polar surface area which is the objective. However, oracles we care about are physics-based simulations. In the main text results and later in the Appendix for Part 2 and Part 3 additional results, we show that this behavior is beneficial for sample efficiency. The physics-based oracles used in this work are AutoDock Vina (Trott & Olson, 2010) and QuickVina 2 (Alhossary et al., 2015) which run molecular docking. The question we pose is: are these oracles *too permissive*? Such that the optimization landscape is smooth

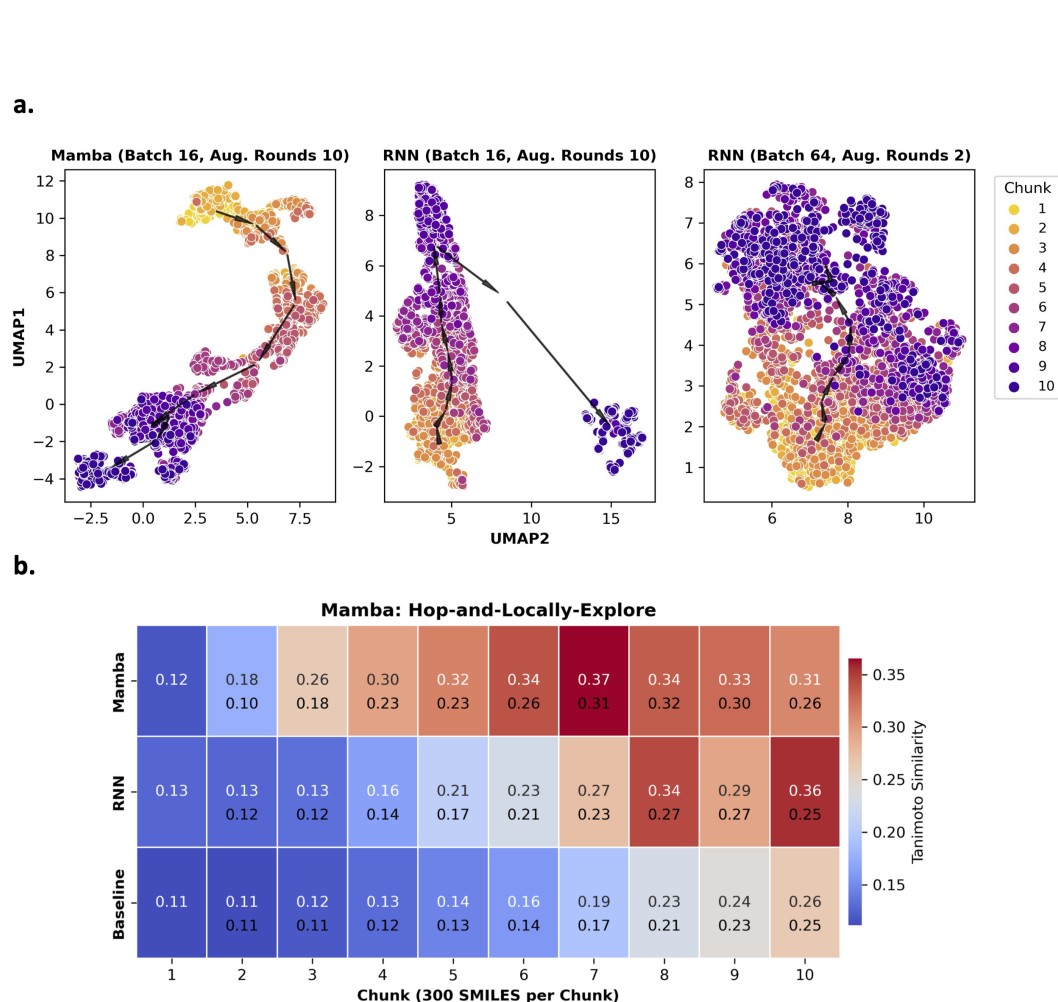

Figure D5: Mamba and RNN (both batch size 16, augmentation rounds 10) and baseline Augmented Memory (batch size 64, augmentation rounds 2). **a.** 3,000 oracle budget test experiment chunked into 300 SMILES. UMAP embedding of the Agent chemical space traversal (arrows are the centroid of each chunk). **b.** Mamba exhibits a "hop-and-locally-explore" behavior where the intra-chunk Tanimoto similarity (top values) are higher than RNN. The bottom value is the inter-chunk similarity.

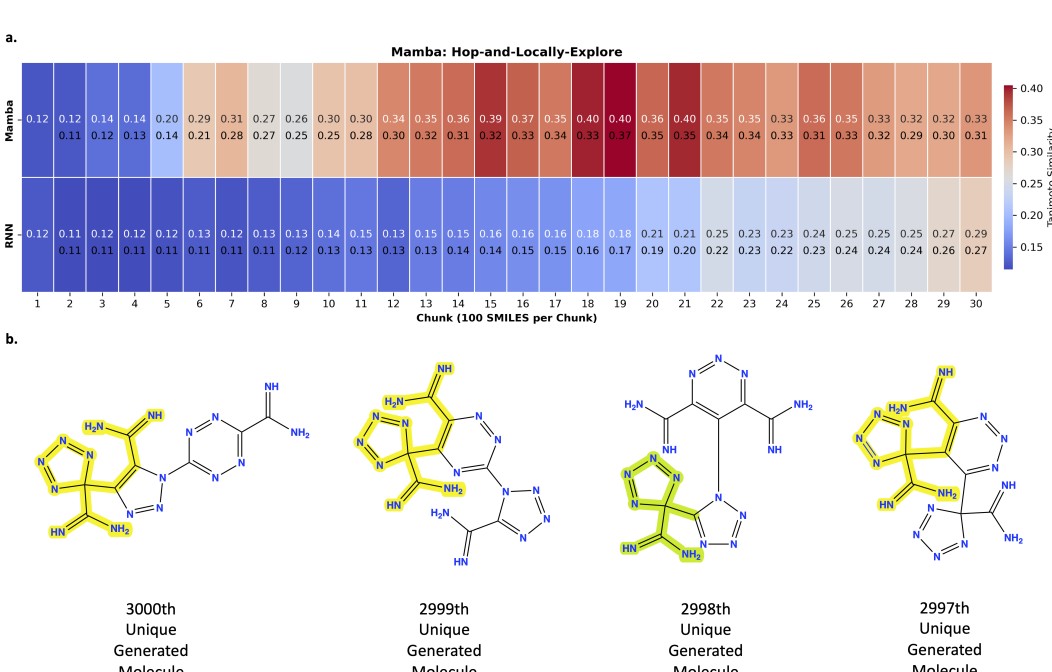

Figure D6: Mamba (batch size 16, augmentation rounds 10) and baseline Augmented Memory (batch size 64, augmentation rounds 2) which is labelled as **RNN**. **a.** 3,000 oracle budget test experiment **chunked into 100 SMILES**. Mamba exhibits a "hop-and-locally-explore" behavior where the intra-chunk Tanimoto similarity (top values) are higher than RNN. The bottom value is the inter-chunk similarity. **b.** Qualitative examples of unique molecules generated at adjacent epochs. Many substructures are shared and the model generates in the local neighborhood. Yellow highlights are exact substructures shared while green indicates a portion.

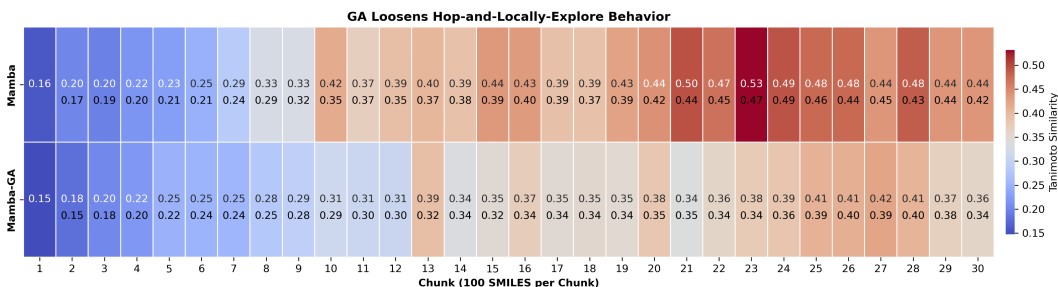

Figure D7: Mamba (batch size 16, augmentation rounds 10) with and without GA (Jensen, 2019) activated. The experiment is the Part 3 MPO objective (docking against parp1).

(Aldeghi et al., 2022). As we push towards higher-fidelity oracles such as QM/MM and free energy simulations (Moore et al., 2023; Crivelli-Decker et al., 2024), it is expected that they will be more stringent and demand more specificity. This means that the current hypothesis of "similar molecules, on average, exhibit similar properties" may be loosened. Whether this turns out to be detrimental or not in high-fidelity oracle settings remains to be empirically tested which we leave for future work. By characterizing the behavior of Saturn and understanding what *exactly* Augmented Memory is doing, it is possible to adapt the current model accordingly. For example, decreasing augmentation rounds relaxes the "hop-and-locally-explore" behavior, which *could* be advantageous for high-fidelity oracles.

### D.2 GENETIC ALGORITHM LOOSENS "HOP-AND-LOCALLY-EXPLORE BEHAVIOR"

In our investigations to applying a GA on the replay buffer, we show that while sample efficiency does not improve, diversity recovers. To quantitatively show why, we plot the chunk similarity for an experiment from Part 3 on the parp1 target with and without the GA activated (Fig. D7). The Mamba model in both cases uses batch size 16 and 10 augmentation rounds. With the GA activated, the intra-chunk similarities decrease, thus loosening the locally exploration behavior and is the reason why diversity recovers.

## E PART 2: TRANSFERABILITY OF SAMPLE EFFICIENCY TO PHYSICS-BASED ORACLES

This section contains information on the Autodock VinaTrott & Olson (2010) docking protocol and additional results. All results are averaged across 10 seeds (0-9 inclusive).

### E.1 DOCKING PROTOCOL

All protein receptor structures were pre-processed from the raw PDB.

**The following were removed**:

1. Duplicate protein chains and duplicate ligands.
2. Co-factors.
3. Ions.
4. All waters.

Next, Schrödinger's Protein Preparation Wizard (Madhavi Sastry et al., 2013; sch) with default parameters was used to pre-process the structure. PROPKA hydrogen-bond network optimization was performed at pH 7.4 and energy minimization with OPLS3e force-field (Roos et al., 2019). Below are details on the docking grids generated from the pre-processed PDBs.

**DRD2 - Dopamine Type 2 Receptor.** The PDB ID is 6CM4Wang et al. (2018) and the docking grid was centered at (x, y, z) = (9.93, 5.85, -9.58).

Table 25: Docking MPO with 1,000 oracle budget. Baseline is vanilla Augmented Memory (Guo & Schwaller, 2024a). All metrics are computed at the 0.7 reward threshold. IntDiv1 is the internal diversity, scaffolds is the number of unique Bemis-Murcko scaffolds, OB is Oracle Burden (oracle calls required to generate $N$ unique molecules). The number in parentheses in the OB statistics represent how many runs out of 10 were successful. The mean and standard deviation across 10 seeds (0-9 inclusive) is reported. Saturn-RNN is RNN with batch size 16 and augmentation rounds 10.

| Model | Yield (↑) | IntDiv1 (↑) | Scaffolds (↑) | OB 1 (↓) | OB 10 (↓) | OB 100 (↓) |
|---|---|---|---|---|---|---|
| DRD2 | | | | | | |
| Baseline | $630 \pm 45$ | $0.858 \pm 0.006$ | $585 \pm 43$ | $57 \pm 2(10)$ | $57 \pm 2(10)$ | $279 \pm 32(10)$ |
| Saturn-RNN | $818 \pm 22$ | $0.821 \pm 0.011$ | $671 \pm 56$ | $14 \pm 1(10)$ | $31 \pm 6(10)$ | $219 \pm 16(10)$ |
| Saturn | $850 \pm 23$ | $0.784 \pm 0.015$ | $677 \pm 51$ | $14 \pm 1(10)$ | $35 \pm 7(10)$ | $199 \pm 20(10)$ |
| Saturn-GA | $804 \pm 26$ | $0.817 \pm 0.022$ | $685 \pm 56$ | $14 \pm 1(10)$ | $35 \pm 7(10)$ | $199 \pm 19(10)$ |
| MK2 Kinase | | | | | | |
| Baseline | $431 \pm 32$ | $0.863 \pm 0.005$ | $406 \pm 26$ | $57 \pm 2(10)$ | $74 \pm 26(10)$ | $396 \pm 37(10)$ |
| Saturn-RNN | $704 \pm 25$ | $0.833 \pm 0.013$ | $525 \pm 32$ | $14 \pm 1(10)$ | $43 \pm 9(10)$ | $282 \pm 19(10)$ |
| Saturn | $702 \pm 43$ | $0.811 \pm 0.022$ | $519 \pm 69$ | $17 \pm 6(10)$ | $52 \pm 12(10)$ | $282 \pm 31(10)$ |
| Saturn-GA | $636 \pm 29$ | $0.827 \pm 0.019$ | $506 \pm 68$ | $17 \pm 6(10)$ | $52 \pm 12(10)$ | $291 \pm 31(10)$ |
| AChE | | | | | | |
| Baseline | $801 \pm 27$ | $0.867 \pm 0.006$ | $759 \pm 30$ | $57 \pm 2(10)$ | $57 \pm 2(10)$ | $201 \pm 29(10)$ |
| Saturn-RNN | $909 \pm 21$ | $0.842 \pm 0.006$ | $772 \pm 73$ | $14 \pm 1(10)$ | $25 \pm 6(10)$ | $163 \pm 19(10)$ |
| Saturn | $906 \pm 15$ | $0.816 \pm 0.014$ | $742 \pm 76$ | $14 \pm 1(10)$ | $27 \pm 4(10)$ | $158 \pm 13(10)$ |
| Saturn-GA | $874 \pm 21$ | $0.841 \pm 0.008$ | $732 \pm 48$ | $14 \pm 1(10)$ | $27 \pm 4(10)$ | $158 \pm 14(10)$ |

**MK2 - MK2 Kinase.** The PDB ID is 3KC3Argiriadi et al. (2010) and the docking grid for the extracted monomer was centered at (x, y, z) = (-61.62, 30.31, -21.9).

**AChE - Acetylcholinesterase.** The PDB ID is 1EVEKryger et al. (1999) and the docking grid was centered at (x, y, z) = (2.78, 64.38, 67.97).

**Docking.** The search box for all grids was 15Å x 15Å x 15Å and docking was executed through DockStream (Guo et al., 2021). All generated molecules were first embedded using the RDKit Universal Force Field (UFF) (Rappé et al., 1992) with the maximum convergence set to 600 iterations. Docking was parallelized over 16 CPU cores (since the generative model's batch size was 16). The cores were Intel(R) Xeon(R) Platinum 8360Y processors.

### E.2  ADDITIONAL RESULTS

In the main text, results were shown at the 0.8 reward threshold. In this section, we also show results for Saturn-RNN (batch size 16, augmentation rounds 10) and for the 0.7 reward threshold (Tables 25 and 26). At the 0.7 reward threshold, Saturn-RNN's performance is almost identical to Saturn. However, at the 0.8 reward threshold, Saturn (using Mamba) is more performant. We highlight that although at times, the difference may be small, it can be highly practically relevant when using expensive oracles, e.g., 50 docking calls may be inconsequential but 50 molecular dynamics simulations can be costly. Both Saturn-RNN and Saturn outperform baseline Augmented Memory. Finally, adding a GA on top of Saturn recovers diversity but sample efficiency decreases.

### E.3  COMPUTE TIME

Due to insufficient GPU resources, we ran all experiments in this section on CPU. Averaged across all targets and across all 10 replicates, the wall time were as follows: 172 minutes (approximately 3 hours) for Augmented Memory, 246 minutes (approximately 4 hours) for Saturn-RNN, 1,426 minutes (approximately 24 hours) for Saturn, and 1,111 minutes (approximately 18.5 hours) for Saturn-GA. There is such a large discrepancy in run time due to repeated SMILES (which do not impose additional oracle calls) that still require backpropagation. Moreover, the runs with Mamba take so much longer because the GPU implementation is highly optimized. When run on GPU, the difference in wall time between Saturn-RNN and Saturn (Mamba) are not significant.

Table 26: Docking MPO with 1,000 oracle budget. Baseline is vanilla Augmented Memory (Guo & Schwaller, 2024a). All metrics are computed at the 0.8 reward threshold. IntDiv1 is the internal diversity, scaffolds is the number of unique Bemis-Murcko scaffolds, OB is Oracle Burden (oracle calls required to generate $N$ unique molecules). The number in parentheses in the OB statistics represent how many runs out of 10 were successful. The mean and standard deviation across 10 seeds (0-9 inclusive) is reported. Saturn-RNN is RNN with batch size 16 and augmentation rounds 10.

| Model | Yield ($\uparrow$) | IntDiv1 ($\uparrow$) | Scaffolds ($\uparrow$) | OB 1 ($\downarrow$) | OB 10 ($\downarrow$) | OB 100 ($\downarrow$) |
|---|---|---|---|---|---|---|
| DRD2 | | | | | | |
| Baseline | $22 \pm 7$ | $0.774 \pm 0.019$ | $22 \pm 7$ | $143 \pm 75(10)$ | $733 \pm 120(10)$ | Failed |
| Saturn-RNN | $185 \pm 40$ | $0.745 \pm 0.022$ | $148 \pm 47$ | $128 \pm 94(10)$ | $440 \pm 72(10)$ | $854 \pm 63(10)$ |
| Saturn | $369 \pm 62$ | $0.671 \pm 0.050$ | $310 \pm 70$ | $93 \pm 53(10)$ | $391 \pm 56(10)$ | $663 \pm 55(10)$ |
| Saturn-GA | $209 \pm 55$ | $0.745 \pm 0.041$ | $189 \pm 57$ | $96 \pm 56(10)$ | $403 \pm 75(10)$ | $806 \pm 84(10)$ |
| MK2 Kinase | | | | | | |
| Baseline | $0.2 \pm 0.4$ | — | $0.2 \pm 0.4$ | $836 \pm 186(2)$ | Failed | Failed |
| Saturn-RNN | $2.5 \pm 3.4$ | $0.414 \pm 0.213$ | $2.5 \pm 3.4$ | $642 \pm 91(6)$ | $999 \pm 0(1)$ | Failed |
| Saturn | $14.9 \pm 14.1$ | $0.454 \pm 0.212$ | $14.1 \pm 13.2$ | $677 \pm 186(9)$ | $861 \pm 108(6)$ | Failed |
| Saturn-GA | $6.1 \pm 6.5$ | $0.415 \pm 0.202$ | $5.5 \pm 5.5$ | $678 \pm 140(9)$ | $911 \pm 11(2)$ | Failed |
| AChE | | | | | | |
| Baseline | $173 \pm 19$ | $0.843 \pm 0.009$ | $170 \pm 18$ | $57 \pm 2(10)$ | $189 \pm 52(10)$ | $776 \pm 58(10)$ |
| Saturn-RNN | $419 \pm 38$ | $0.804 \pm 0.019$ | $338 \pm 55$ | $21 \pm 11(10)$ | $165 \pm 60(10)$ | $531 \pm 36(10)$ |
| Saturn | $480 \pm 79$ | $0.757 \pm 0.020$ | $400 \pm 96$ | $32 \pm 24(10)$ | $185 \pm 82(10)$ | $508 \pm 80(10)$ |
| Saturn-GA | $343 \pm 57$ | $0.809 \pm 0.013$ | $287 \pm 50$ | $32 \pm 25(10)$ | $187 \pm 80(10)$ | $565 \pm 80(10)$ |

# F  PART 3: PART 3: BENCHMARKING SATURN

In this section, we detail how Saturn was pre-trained for benchmarking, the procedure we followed to reproduce GEAM (Lee et al., 2024), and additional results. We ensured exact reproducibility by using GEAM's official code: `https://anonymous.4open.science/r/GEAM-45EF`. For running Saturn with GEAM's objective function, all the oracle code was taken, without modification, from the same repository.

## F.1  SATURN ZINC 250K PRE-TRAINING

GEAM pre-trained on ZINC 250k (Sterling & Irwin, 2015) and provide the dataset in their repository. We used this dataset as is for Saturn pre-training (Mamba model).

**The pre-training parameters were**:

1. Training steps = 50 (each training step entails a full pass through the dataset)

2. Seed = 0

3. Batch size = 512

4. Learning rate = 0.0001

5. Train with SMILES randomization (Bjerrum, 2017) (all SMILES in each batch was randomized)

**Mamba model**:

1. Vocabulary size = 66 (including the 2 added tokens for <START> and <END>)

2. 5,272,832 parameters

3. Used checkpoint from epoch 50 (NLL = 28.10, Validity (10k) = 95.2%)

All Saturn experiments were run on a single workstation equipped with an NVIDIA RTX A6000 GPU and AMD Ryzen 9 5900X 12-Core CPU. The total run time for Saturn across all targets was 41.5 hours (total of 50 runs: 5 targets, 10 seeds each).

## F.2 REPRODUCING GEAM'S RESULTS

We followed the instructions directly in GEAM's README: `https://anonymous.4open.science/r/GEAM-45EF/README.md`. We trained the FGIB with seed 0. Everything else was run with their default parameters. In the original work, 3 replicates were run but the seeds were not specified. In our comparisons, we run GEAM across 10 seeds (0-9 inclusive) using an NVIDIA V100 GPU with a Xeon-Gold processor (2.1 GHz and 20 cores) CPU. The reason why a different GPU was used in GEAM experiments compared to Saturn is due to CUDA compatibility in GEAM's code. For GEAM, the wall times were:

1. **parp1**: 3.02±0.19 hours
2. **fa7**: 3.38±0.04 hours
3. **5ht1b**: 3.17±0.08 hours
4. **braf**: 3.02±0.19 hours
5. **jak2**: 3.28±0.04 hours

Except for **parp1**, the wall times are the mean and standard deviation across 10 seeds. For **parp1**, the wall times are across 7 seeds (3-9 inclusive). Seeds 0-2 inclusive were run on CPU due to insufficient GPU resources. CPU runs take much longer so we only report GPU times.

## F.3 GEAM'S MPO OBJECTIVE

GEAM optimized for the following objective:

$$R(x) = \widehat{DS}(x) \times QED(x) \times \widehat{SA}(x) \in [0, 1] \qquad (22)$$

$\widehat{DS}$ is the normalized QuickVina 2 (Alhossary et al., 2015) docking score (Eq. 23), QED (Bickerton et al., 2012) is the quantitative estimate of drug-likeness, and $\widehat{SA}$ is the normalized synthetic accessibility score (Ertl & Schuffenhauer, 2009) (Eq. 24).

$$\widehat{DS} = -\frac{DS}{20} \qquad (23)$$

$$\widehat{SA} = \frac{10 - SA}{9} \qquad (24)$$

## F.4 SATURN-TANIMOTO

In GEAM (Lee et al., 2024), the "Novel" in **Novel Hit Ratio** enforces molecules to possess < 0.4 Tanimoto similarity to ZINC 250k (Sterling & Irwin, 2015). GEAM achieves this by use of their fragment assembly *and* genetic algorithm which directly uses GraphGA (Jensen, 2019). The crossover and mutation operations promote diversityLiu et al. (2021). Otherwise, generative models are pre-trained to model the training data distribution. This means that generated molecules would not necessarily be particularly dissimilar to the training data, especially if the training data actually possesses "good" molecules already. By virtue of pre-training on a selected dataset, we implicitly assume that the pre-training dataset is "good" for our task, otherwise, we probably should not pre-train on this data. This is the rationale on why ChEMBL (Gaulton et al., 2012) and ZINC 250k (Sterling & Irwin, 2015) are popular pre-training datasets: they contain bio-active molecules. To satisfy GEAM's "Novel" criterion, we take the base Saturn model and first teach it to generate molecules that are dissimilar to the ZINC 250k dataset which was used for pre-training. The objective function is then defined as minimizing the max Tanimoto similarity to any molecule in ZINC 250k. This experiment was run with an oracle budget of 1,500 and took about 10 minutes. The resulting **Saturn-Tanimoto** model generates molecules with low Tanimoto similarity to ZINC 250k. Starting from this model, we run GEAM's case study and the results from this are reported in the main text and here in the Appendix. We finally note that this criterion is somewhat arbitrary and we do it so we can exactly match GEAM's experiments.

## F.5 QUANTITATIVE SUPPLEMENTARY RESULTS

In this section, we present supplementary benchmarking results and show additional results for Saturn-GA.

Table 27: Hit Ratio (%). Results are from Lee et al. (Lee et al., 2023) except GEAM, datasets, and Saturn which we ran across 10 seeds (0-9 inclusive). The mean and standard deviation are reported. Best results (statistically significant at the 95% confidence level) are bolded.

| Method | Target Protein | | | | |
| --- | --- | --- | --- | --- | --- |
| | parp1 | fa7 | 5ht1b | braf | jak2 |
| **Datasets** | | | | | |
| ZINC 250k (Sterling & Irwin, 2015) | $3.993 \pm 0.355$ | $1.097 \pm 0.192$ | $24.260 \pm 0.622$ | $1.020 \pm 0.193$ | $6.183 \pm 0.344$ |
| ChEMBL 33 (Gaulton et al., 2012) | $6.077 \pm 0.453$ | $1.830 \pm 0.240$ | $24.163 \pm 0.715$ | $2.073 \pm 0.181$ | $9.013 \pm 0.562$ |
| **Generative Models** | | | | | |
| REINVENT (Olivecrona et al., 2017) | $4.693 \pm 1.776$ | $1.967 \pm 0.661$ | $26.047 \pm 2.497$ | $2.207 \pm 0.800$ | $5.667 \pm 1.067$ |
| JT-VAE (Jin et al., 2018) | $3.200 \pm 0.348$ | $0.933 \pm 0.152$ | $18.044 \pm 0.747$ | $0.644 \pm 0.157$ | $5.856 \pm 0.204$ |
| GraphAF (Shi et al., 2020) | $0.822 \pm 0.113$ | $0.011 \pm 0.016$ | $6.978 \pm 0.952$ | $1.422 \pm 0.556$ | $1.233 \pm 0.284$ |
| MORLD (Jeon & Kim, 2020) | $0.047 \pm 0.050$ | $0.007 \pm 0.013$ | $0.893 \pm 0.758$ | $0.047 \pm 0.040$ | $0.227 \pm 0.118$ |
| HierVAE (Jin et al., 2020a) | $1.180 \pm 0.182$ | $0.033 \pm 0.030$ | $0.740 \pm 0.371$ | $0.367 \pm 0.187$ | $0.487 \pm 0.183$ |
| GraphDF (Luo et al., 2021) | $0.044 \pm 0.031$ | $0.000 \pm 0.000$ | $0.000 \pm 0.000$ | $0.011 \pm 0.016$ | $0.011 \pm 0.016$ |
| FREED (Yang et al., 2021) | $4.860 \pm 1.415$ | $1.487 \pm 0.242$ | $14.227 \pm 5.116$ | $2.707 \pm 0.721$ | $6.067 \pm 0.790$ |
| FREED-QS (Yang et al., 2021) | $5.960 \pm 0.902$ | $1.687 \pm 0.177$ | $23.140 \pm 2.422$ | $3.880 \pm 0.623$ | $7.653 \pm 1.373$ |
| LIMO (Eckmann et al., 2022) | $0.456 \pm 0.057$ | $0.044 \pm 0.016$ | $1.200 \pm 0.178$ | $0.278 \pm 0.134$ | $0.711 \pm 0.329$ |
| GDSS (Jo et al., 2022) | $2.367 \pm 0.316$ | $0.467 \pm 0.112$ | $6.267 \pm 0.287$ | $0.300 \pm 0.198$ | $1.367 \pm 0.258$ |
| MOOD (Lee et al., 2023) | $7.260 \pm 0.764$ | $0.787 \pm 0.128$ | $21.427 \pm 0.502$ | $5.913 \pm 0.311$ | $10.367 \pm 0.616$ |
| Augmented Memory (Guo & Schwaller, 2024a) | $16.966 \pm 3.224$ | $2.637 \pm 0.860$ | $52.016 \pm 2.302$ | $8.307 \pm 1.714$ | $21.548 \pm 4.938$ |
| GEAM (Lee et al., 2024) | $\mathbf{45.158 \pm 2.408}$ | $\mathbf{20.552 \pm 2.357}$ | $47.664 \pm 1.198$ | $\mathbf{30.444 \pm 1.610}$ | $46.129 \pm 2.073$ |
| **Ours** | | | | | |
| Saturn | $\mathbf{57.981 \pm 18.537}$ | $14.527 \pm 9.961$ | $\mathbf{68.185 \pm 3.400}$ | $38.999 \pm 10.114$ | $\mathbf{60.827 \pm 11.502}$ |
| Saturn-GA | $55.597 \pm 5.617$ | $16.711 \pm 6.761$ | $63.112 \pm 4.316$ | $34.284 \pm 10.345$ | $58.625 \pm 6.982$ |
| *Saturn-Tanimoto* | $77.674 \pm 7.127$ | $23.119 \pm 6.852$ | $78.433 \pm 1.029$ | $30.258 \pm 12.315$ | $83.012 \pm 6.678$ |

Table 28: Strict Hit Ratio (%) (QED > 0.7 and SA < 3) additional results. GEAM and Saturn results are across 10 seeds (0-9 inclusive). OB is Oracle Burden (oracle calls required to generate $N$ unique molecules). The number in parentheses in the OB statistics represent how many runs out of 10 were successful. The mean and standard deviation are reported. Best results (statistically significant at the 95% confidence level) are bolded.

| Method | Target Protein | | | | |
| --- | --- | --- | --- | --- | --- |
| | parp1 | fa7 | 5ht1b | braf | jak2 |
| **GEAM (Lee et al., 2024) - Presented in Main Text** | | | | | |
| Strict Hit Ratio (↑) | $6.510 \pm 1.087$ | $2.106 \pm 0.958$ | $8.719 \pm 0.903$ | $3.685 \pm 0.524$ | $7.944 \pm 1.157$ |
| IntDiv1 (↑) | $0.766 \pm 0.017$ | $0.709 \pm 0.043$ | $0.799 \pm 0.017$ | $0.751 \pm 0.023$ | $0.763 \pm 0.021$ |
| #Circles (↑) | $14 \pm 3$ | $7 \pm 2$ | $25 \pm 3$ | $11 \pm 2$ | $18 \pm 2$ |
| OB (1) (↓) | $250 \pm 157(10)$ | $433 \pm 209(10)$ | $114 \pm 112(10)$ | $355 \pm 96(10)$ | $230 \pm 117(10)$ |
| OB (10) (↓) | $743 \pm 52(10)$ | $1446 \pm 404(10)$ | $531 \pm 38(10)$ | $892 \pm 144(10)$ | $537 \pm 70(10)$ |
| OB (100) (↓) | $2106 \pm 202(10)$ | $2927 \pm 0(1)$ | $1527 \pm 110(10)$ | $2674 \pm 163(6)$ | $1606 \pm 218(10)$ |
| **Saturn (ours) - Presented in Main Text** | | | | | |
| Strict Hit Ratio | $55.102 \pm 18.027$ | $13.887 \pm 9.723$ | $64.730 \pm 3.717$ | $37.250 \pm 9.615$ | $55.903 \pm 13.613$ |
| IntDiv1 (↑) | $0.596 \pm 0.049$ | $0.592 \pm 0.066$ | $0.685 \pm 0.021$ | $0.597 \pm 0.042$ | $0.638 \pm 0.034$ |
| #Circles (↑) | $5 \pm 0$ | $3 \pm 1$ | $17 \pm 3$ | $4 \pm 0$ | $7 \pm 1$ |
| OB (1) (↓) | $139 \pm 96(10)$ | $352 \pm 206(10)$ | $21 \pm 7(10)$ | $291 \pm 143(10)$ | $88 \pm 56(10)$ |
| OB (10) (↓) | $518 \pm 92(10)$ | $924 \pm 247(10)$ | $105 \pm 23(10)$ | $581 \pm 123(10)$ | $348 \pm 96(10)$ |
| OB (100) (↓) | $956 \pm 259(10)$ | $1776 \pm 551(10)$ | $441 \pm 44(10)$ | $1057 \pm 187(10)$ | $785 \pm 191(10)$ |
| **Saturn-GA (ours) - Newly presented here** | | | | | |
| Strict Hit Ratio | $47.146 \pm 4.952$ | $13.187 \pm 6.340$ | $53.055 \pm 3.764$ | $28.377 \pm 9.703$ | $49.528 \pm 5.463$ |
| IntDiv1 (↑) | $0.659 \pm 0.023$ | $0.636 \pm 0.039$ | $0.724 \pm 0.022$ | $0.625 \pm 0.047$ | $0.676 \pm 0.041$ |
| #Circles (↑) | $8 \pm 2$ | $4 \pm 1$ | $22 \pm 4$ | $6 \pm 1$ | $12 \pm 2$ |
| OB (1) (↓) | $121 \pm 71(10)$ | $350 \pm 203(10)$ | $20 \pm 6(10)$ | $242 \pm 194(10)$ | $91 \pm 43(10)$ |
| OB (10) (↓) | $467 \pm 114(10)$ | $912 \pm 168(10)$ | $110 \pm 36(10)$ | $582 \pm 177(10)$ | $375 \pm 120(10)$ |
| OB (100) (↓) | $937 \pm 136(10)$ | $1852 \pm 349(10)$ | $499 \pm 85(10)$ | $1266 \pm 486(10)$ | $861 \pm 123(10)$ |

**Hit Ratio (%).** Table 27 shows the Hit Ratio (%) results. Random sampling of 3,000 molecules from common datasets (ZINC 250k (Sterling & Irwin, 2015) and ChEMBL 33 (Gaulton et al., 2012)) are included as baselines. The results show that only GEAM (Lee et al., 2024) and Saturn outperform these baselines with both methods performing similarly overall. With the exception of a few targets where performance differs (significant at the 95% confidence level), Saturn notably exhibits higher variance which is expected given the small batch size (16). One way to mitigate high variance is to use a larger batch size, as this makes the approximation for the expected reward less noisy. Next, we show that the Saturn-Tanimoto Agent displays notably high Hit Ratios but do not present this in the main results as the purpose of the Tanimoto Agent is to generate hits that have less than 0.4 Tanimoto similarity to the ZINC 250k (Sterling & Irwin, 2015) training dataset. It is difficult to predict *a priori* a favorable chemical space to move the Agent. However, this result is interesting as it

suggests that this simple additional pre-training which took minutes via curriculum learning (CL), makes the Agent more suited for the docking tasks. Finally, we show that using the GA (Saturn-GA) is a straightforward solution to recover diversity. From Part 1 and Part 2 experiments, activating the GA comes at the expense of some sample efficiency but interestingly, this is not the case here (Table 28). Moreover, Saturn-GA also decreases variance in this case study (Table 27). Based on these results, it would actually be beneficial to activate the GA in this case, but it is difficult to know *a priori* the best configuration, thus we report the out-of-the-box hyperparameters (without GA) in the main text based on tuning on the test experiment in Part 1.

Table 29: Strict Novel Hit Ratio (%) (QED > 0.7 and SA < 3). GEAM and Saturn results are across 10 seeds (0-9 inclusive). OB is Oracle Burden (oracle calls required to generate *N* unique molecules). The number in parentheses in the OB statistics represent how many runs out of 10 were successful. The mean and standard deviation are reported. Best results (statistically significant at the 95% confidence level) are bolded.

| Method | Target Protein | | | | |
|---|---|---|---|---|---|
| | parp1 | fa7 | 5ht1b | braf | jak2 |
| **GEAM** (Lee et al., 2024) | | | | | |
| Strict Hit Ratio (↑) | $4.018 \pm 0.849$ | $1.676 \pm 0.836$ | $5.338 \pm 0.789$ | $2.621 \pm 0.464$ | $5.930 \pm 1.151$ |
| **IntDiv1** (↑) | $0.768 \pm 0.019$ | $0.710 \pm 0.047$ | $0.793 \pm 0.019$ | $0.753 \pm 0.026$ | $0.763 \pm 0.026$ |
| **#Circles** (↑) | $13 \pm 2$ | $5 \pm 2$ | $21 \pm 3$ | $11 \pm 2$ | $16 \pm 3$ |
| OB (1) (↓) | $319 \pm 175(10)$ | $502 \pm 209(10)$ | $253 \pm 159(10)$ | $419 \pm 102(10)$ | $242 \pm 124(10)$ |
| OB (10) (↓) | $857 \pm 86(10)$ | $1625 \pm 380(10)$ | $689 \pm 77(10)$ | $1047 \pm 136(10)$ | $616 \pm 83(10)$ |
| OB (100) (↓) | $2633 \pm 202(9)$ | Failed | $2221 \pm 224(10)$ | $2942 \pm 0(1)$ | $2005 \pm 268(10)$ |
| **Saturn-Tanimoto (ours)** | | | | | |
| **Strict Novel Hit Rate** | $47.405 \pm 8.593$ | $17.130 \pm 5.538$ | $50.445 \pm 6.334$ | $18.228 \pm 9.438$ | $45.185 \pm 13.321$ |
| IntDiv1 (↑) | $0.595 \pm 0.029$ | $0.600 \pm 0.030$ | $0.559 \pm 0.032$ | $0.520 \pm 0.040$ | $0.567 \pm 0.041$ |
| #Circles (↑) | $2 \pm 0$ | $2 \pm 0$ | $2 \pm 0$ | $1 \pm 0$ | $1 \pm 0$ |
| **OB (1)** (↓) | $26 \pm 17(10)$ | $98 \pm 53(10)$ | $15 \pm 0(10)$ | $164 \pm 137(10)$ | $18 \pm 7(10)$ |
| **OB (10)** (↓) | $177 \pm 38(10)$ | $320 \pm 69(10)$ | $31 \pm 5(10)$ | $388 \pm 156(10)$ | $70 \pm 13(10)$ |
| **OB (100)** (↓) | $562 \pm 94(10)$ | $1051 \pm 251(10)$ | $223 \pm 50(10)$ | $1041 \pm 585(9)$ | $402 \pm 196(10)$ |
| **Saturn-Tanimoto-GA (ours)** | | | | | |
| Strict Novel Hit Rate | $29.801 \pm 11.603$ | $11.895 \pm 5.197$ | $40.261 \pm 8.168$ | $17.845 \pm 7.943$ | $37.498 \pm 11.200$ |
| IntDiv1 (↑) | $0.621 \pm 0.041$ | $0.596 \pm 0.030$ | $0.613 \pm 0.042$ | $0.640 \pm 0.040$ | $0.606 \pm 0.034$ |
| #Circles (↑) | $3 \pm 1$ | $2 \pm 1$ | $3 \pm 1$ | $3 \pm 1$ | $3 \pm 1$ |
| OB (1) (↓) | $36 \pm 38(10)$ | $216 \pm 232(10)$ | $15 \pm 0(10)$ | $181 \pm 122(10)$ | $17 \pm 5(10)$ |
| OB (10) (↓) | $205 \pm 65(10)$ | $556 \pm 275(10)$ | $27 \pm 5(10)$ | $472 \pm 135(10)$ | $96 \pm 13(10)$ |
| OB (100) (↓) | $703 \pm 113(10)$ | $1490 \pm 460(9)$ | $272 \pm 39(10)$ | $1367 \pm 561(10)$ | $480 \pm 84(10)$ |

**Novel Hit Ratio (%).** Table 29 shows the Novel Hit Ratio (%) results with all additional metrics, mirroring the main text table. Similar to the main text results, Mamba-Tanimoto Agent generates significantly more molecules passing the strict filter and also much faster (fewer oracle calls). However, the diversity notably drops (much more than the Mamba Agent without Tanimoto distance training presented in the main text). However, diversity is particularly low. We first not that when moving to high-fidelity oracles where satisfying the objective function equates to higher true positive hit rates, low diversity need not be detrimental. We additionally run an experiment with the GA activated and we see diversity recovers, but is still notably lower than GEAM. Moreover, the sample efficiency drops notably here compared to without GA, but is still much more performant than GEAM in finding hits faster. Finally, to recover more diversity, one could make the Diversity Filter (Blaschke et al., 2020b) more stringent. In this work, a bucket size of 10 was used (allow 10 of the same scaffold to be generated before truncating the reward to 0). Decreasing the bucket size to 5 or even lower, may recover more diversity.

F.6 SATURN: ARCHITECTURE SCALING.

In the main text Part 1, we investigated *why* Mamba (5.2M) outperforms LSTM (Hochreiter & Schmidhuber, 1997) RNN (5.8M) and Decoder transformer (Vaswani et al., 2017; Radford et al., 2019) (6.3M). Augmented Memory (Guo & Schwaller, 2024a) squeezes the likelihood of generating augmented forms of *any* replay buffer *molecules*. Increased capacity to match this distribution directly leads to the "hop-and-locally-explore" behavior which improves sample efficiency. We note that our observations are for optimization landscapes that are not *too rough* (Guo et al., 2022; Aldeghi et al., 2022). It is difficult to know *a priori* the roughness of optimization and also whether the benefits of "hop-and-locally-explore" behavior is beneficial in higher-fidelity oracle settings. We leave this for future work.

Based on these observations, we investigate scaling benefits for the LSTM RNN and Decoder transformer models. Increasing model size can lead to lower loss convergence, which in this case, means modeling the conditional token distribution of the SMILES (Weininger, 1988). One may argue that this is simply a hyperparameter tuning which we missed. However, the purpose of this work is in the goal-directed learning setting where we want to *tune* the model's distribution towards desirable molecules. If desirable molecules are already in the training data, minimal optimization is required. Moreover, it is difficult to know *a priori* whether matching the training distribution *very closely* is strictly advantageous for an arbitrary MPO objective, unless we have an enormous amount of data, by the law of large numbers. Therefore, all pre-trained models (priors) in this work were trained until loss flattens out and Validity (fraction of valid SMILES generated) is high.

In this section, we scale up the LSTM RNN and Decoder transformer models to around 25M to make the *distribution learning capability* approach Mamba (5.2M). We use the training loss for this, where similar loss convergence is taken as the proxy. We first present the exact model parameter counts, hyperparameters, and training details.

**LSTM RNN 24.7M**:

1. Seed = 0
2. Parameters = 24,741,442
3. Vocabulary Size = 66
4. Embedding Dimension = 256
5. Hidden Dimension = 512
6. Number of Layers = 12
7. Dropout = 0.0
8. Layer Normalization = False
9. Train Epochs = 300
10. Batch Size = 512
11. Learning Rate = 0.0001
12. Final NLL Loss at Epoch 300 = 29.318

**Decoder 25.3M**:

1. Seed = 0
2. Parameters = 25,306,178
3. Vocabulary Size = 66
4. Embedding Dimension = 256
5. Hidden Dimension = 1024
6. Number of Layers = 32
7. Number of Heads = 16
8. Dropout = 0.0
9. Train Epochs = 100
10. Batch Size = 512
11. Learning Rate = 0.0001
12. Final NLL Loss at Epoch 100 = 26.963

In addition, we scale up Mamba to 16M and 21M and also present the exact model parameter counts, hyperparameters, and training details. For these two models, we intentionally train until the loss is at similar values (NLL = 26) which suggests both models have learned the training distribution to a similar extent. Optimization then starts from a similar distribution.

**Mamba 15.8M**:

1. Seed = 0

2. Parameters = 15,785,728

3. Vocabulary Size = 66

4. Embedding Dimension = 256

5. **Number of Layers = 36**

6. Use RMSNorm = True

7. Residual in fp32 = True

8. Fused AddNorm = True

9. Train Epochs = 100

10. Batch Size = 512

11. Learning Rate = 0.0001

12. Final NLL Loss at Epoch 92 = 26.003

**Mamba 21.0M**:

1. Seed = 0

2. Parameters = 21,041,920

3. Vocabulary Size = 66

4. Embedding Dimension = 256

5. **Number of Layers = 48**

6. Use RMSNorm = True

7. Residual in fp32 = True

8. Fused AddNorm = True

9. Train Epochs = 100

10. Batch Size = 512

11. Learning Rate = 0.0001

12. Final NLL Loss at Epoch 75 = 25.993

Table 30: Architecture scaling experiments: Hit Ratio (%) metrics. GEAM (Lee et al., 2024) and Saturn results are across 10 seeds (0-9 inclusive). The mean and standard deviation are reported.

| Method | Target Protein | | | | |
|---|---|---|---|---|---|
| | parp1 | fa7 | 5ht1b | braf | jak2 |
| **Datasets** | | | | | |
| ZINC 250k (Sterling & Irwin, 2015) | $3.993 \pm 0.355$ | $1.097 \pm 0.192$ | $24.26 \pm 0.622$ | $1.020 \pm 0.193$ | $6.183 \pm 0.344$ |
| ChEMBL 33 (Gaulton et al., 2012) | $6.077 \pm 0.453$ | $1.830 \pm 0.240$ | $24.163 \pm 0.715$ | $2.073 \pm 0.181$ | $9.013 \pm 0.562$ |
| **Generative Models** | | | | | |
| Augmented Memory (Guo & Schwaller, 2024a) | $16.983 \pm 3.221$ | $2.641 \pm 0.868$ | $52.046 \pm 2.327$ | $8.354 \pm 1.727$ | $21.604 \pm 4.958$ |
| GEAM (Lee et al., 2024) | $49.597 \pm 3.078$ | $21.988 \pm 2.968$ | $51.765 \pm 1.463$ | $33.086 \pm 1.673$ | $51.228 \pm 3.132$ |
| **Ours** | | | | | |
| Saturn-Mamba 5.2M | $57.981 \pm 18.537$ | $14.527 \pm 9.961$ | $68.185 \pm 3.400$ | $38.999 \pm 10.114$ | $60.827 \pm 11.502$ |
| Saturn-Mamba 15.8M | $56.088 \pm 9.899$ | $18.804 \pm 13.980$ | $68.322 \pm 3.885$ | $38.699 \pm 19.841$ | $61.320 \pm 18.673$ |
| Saturn-Mamba 21.0M | $56.299 \pm 16.583$ | $23.764 \pm 19.280$ | $65.015 \pm 6.060$ | $32.018 \pm 12.584$ | $59.175 \pm 20.689$ |
| Saturn-Decoder 25.3M | $61.732 \pm 16.032$ | $21.058 \pm 13.940$ | $68.340 \pm 5.094$ | $37.399 \pm 12.632$ | $65.470 \pm 12.628$ |
| Saturn-RNN 24.7M | $52.914 \pm 9.955$ | $13.254 \pm 7.276$ | $63.799 \pm 3.249$ | $33.805 \pm 8.694$ | $54.165 \pm 7.445$ |

**Hit Ratios (%).** Table 30 shows the Hit Ratios of compared models. Saturn outperforms baseline Augmented Memory and GEAM. In terms of architecture scaling, we show decoder transformer and RNN approach Mamba performance but are still less performant. Scaling up Mamba does not necessarily lead to better results, as there is notably even higher variance.

**Sample Efficiency Metrics** Table 31 presents the Strict Hit Ratios for compared models. While GEAM outperforms baseline Augmented Memory for the Hit Ratio, the results here show that the optimization capability of baseline Augmented Memory exceeds that of GEAM. Saturn outperforms both Augmented Memory and GEAM to generate more hits and also finds them faster (lower

Table 31: Architecture scaling experiments: Strict Hit Ratio (%) (QED > 0.7 and SA < 3). GEAM and Saturn results are across 10 seeds (0-9 inclusive). OB is Oracle Burden (oracle calls required to generate $N$ unique molecules). The number in parentheses in the OB statistics represent how many runs out of 10 were successful. The mean and standard deviation are reported.

| Method | Target Protein | | | | |
|---|---|---|---|---|---|
| | parp1 | fa7 | 5ht1b | braf | jak2 |
| **GEAM** (Lee et al., 2024) | | | | | |
| Strict Hit Ratio (↑) | $6.510 \pm 1.087$ | $2.106 \pm 0.958$ | $8.719 \pm 0.903$ | $3.685 \pm 0.524$ | $7.944 \pm 1.157$ |
| IntDiv1 (↑) | $0.766 \pm 0.017$ | $0.709 \pm 0.043$ | $0.799 \pm 0.017$ | $0.751 \pm 0.023$ | $0.763 \pm 0.021$ |
| #Circles (↑) | $14 \pm 3$ | $7 \pm 2$ | $25 \pm 3$ | $11 \pm 2$ | $18 \pm 2$ |
| OB (1) (↓) | $250 \pm 157(10)$ | $433 \pm 209(10)$ | $114 \pm 112(10)$ | $355 \pm 96(10)$ | $230 \pm 117(10)$ |
| OB (10) (↓) | $743 \pm 52(10)$ | $1446 \pm 404(10)$ | $531 \pm 38(10)$ | $892 \pm 144(10)$ | $537 \pm 70(10)$ |
| OB (100) (↓) | $2106 \pm 202(10)$ | $2927 \pm 0(1)$ | $1527 \pm 110(10)$ | $2674 \pm 163(6)$ | $1606 \pm 218(10)$ |
| **Augmented Memory** (Guo & Schwaller, 2024a) | | | | | |
| Strict Hit Ratio | $13.486 \pm 3.033$ | $1.757 \pm 0.805$ | $43.824 \pm 2.124$ | $6.920 \pm 1.734$ | $17.884 \pm 4.636$ |
| IntDiv1 (↑) | $0.748 \pm 0.019$ | $0.718 \pm 0.047$ | $0.779 \pm 0.007$ | $0.685 \pm 0.022$ | $0.772 \pm 0.013$ |
| #Circles (↑) | $20 \pm 5$ | $9 \pm 2$ | $54 \pm 6$ | $8 \pm 1$ | $27 \pm 3$ |
| OB (1) (↓) | $173 \pm 149(10)$ | $503 \pm 313$ | $61 \pm 1(10)$ | $329 \pm 152$ | $80 \pm 28(10)$ |
| OB (10) (↓) | $686 \pm 214(10)$ | $1776 \pm 257(10)$ | $117 \pm 51(10)$ | $1173 \pm 375(10)$ | $420 \pm 54(10)$ |
| OB (100) (↓) | $1836 \pm 174(10)$ | $2867 \pm 0(1)$ | $657 \pm 80(10)$ | $2396 \pm 139(9)$ | $1499 \pm 109(10)$ |
| **Ours** | | | | | |
| **Saturn-Mamba 5.2M** | | | | | |
| Strict Hit Ratio | $55.102 \pm 18.027$ | $13.887 \pm 9.723$ | $64.730 \pm 3.717$ | $37.250 \pm 9.615$ | $55.903 \pm 13.613$ |
| IntDiv1 (↑) | $0.596 \pm 0.049$ | $0.592 \pm 0.066$ | $0.685 \pm 0.021$ | $0.597 \pm 0.042$ | $0.638 \pm 0.034$ |
| #Circles (↑) | $5 \pm 0$ | $3 \pm 1$ | $17 \pm 3$ | $4 \pm 0$ | $7 \pm 1$ |
| OB (1) (↓) | $139 \pm 96(10)$ | $352 \pm 206(10)$ | $21 \pm 7(10)$ | $291 \pm 143(10)$ | $88 \pm 56(10)$ |
| OB (10) (↓) | $518 \pm 92(10)$ | $924 \pm 247(10)$ | $105 \pm 23(10)$ | $581 \pm 123(10)$ | $348 \pm 96(10)$ |
| OB (100) (↓) | $956 \pm 259(10)$ | $1776 \pm 551(10)$ | $441 \pm 44(10)$ | $1057 \pm 187(10)$ | $785 \pm 191(10)$ |
| **Saturn-Mamba 15.8M** | | | | | |
| Strict Hit Ratio | $52.093 \pm 12.503$ | $18.064 \pm 13.932$ | $63.740 \pm 5.623$ | $37.350 \pm 19.173$ | $59.372 \pm 18.465$ |
| IntDiv1 (↑) | $0.587 \pm 0.033$ | $0.587 \pm 0.068$ | $0.662 \pm 0.042$ | $0.568 \pm 0.064$ | $0.633 \pm 0.035$ |
| #Circles (↑) | $6 \pm 2$ | $3 \pm 1$ | $18 \pm 3$ | $4 \pm 1$ | $9 \pm 2$ |
| OB (1) (↓) | $157 \pm 112(10)$ | $223 \pm 167(10)$ | $25 \pm 10(10)$ | $204 \pm 115(10)$ | $54 \pm 43(10)$ |
| OB (10) (↓) | $406 \pm 111(10)$ | $691 \pm 151(10)$ | $108 \pm 31(10)$ | $634 \pm 180(10)$ | $266 \pm 50(10)$ |
| OB (100) (↓) | $905 \pm 204(10)$ | $1491 \pm 389(8)$ | $421 \pm 61(10)$ | $1220 \pm 410(10)$ | $786 \pm 254(10)$ |
| **Saturn-Mamba 21.0M** | | | | | |
| Strict Hit Ratio | $54.297 \pm 16.480$ | $23.021 \pm 19.064$ | $61.307 \pm 5.991$ | $30.972 \pm 12.605$ | $57.013 \pm 20.601$ |
| IntDiv1 (↑) | $0.590 \pm 0.041$ | $0.535 \pm 0.056$ | $0.655 \pm 0.042$ | $0.560 \pm 0.060$ | $0.605 \pm 0.046$ |
| #Circles (↑) | $6 \pm 1$ | $4 \pm 1$ | $17 \pm 3$ | $4 \pm 1$ | $8 \pm 1$ |
| OB (1) (↓) | $167 \pm 73(10)$ | $316 \pm 236(10)$ | $28 \pm 13(10)$ | $235 \pm 138(10)$ | $68 \pm 78(10)$ |
| OB (10) (↓) | $425 \pm 91(10)$ | $710 \pm 314(10)$ | $115 \pm 44(10)$ | $556 \pm 147(10)$ | $335 \pm 118(10)$ |
| OB (100) (↓) | $831 \pm 147(10)$ | $1446 \pm 629(9)$ | $432 \pm 69(10)$ | $1134 \pm 282(10)$ | $798 \pm 340(10)$ |
| **Saturn-Decoder 25.3M** | | | | | |
| Strict Hit Ratio | $59.560 \pm 15.480$ | $20.195 \pm 13.394$ | $65.202 \pm 5.847$ | $35.857 \pm 12.228$ | $62.874 \pm 11.810$ |
| IntDiv1 (↑) | $0.615 \pm 0.034$ | $0.575 \pm 0.078$ | $0.658 \pm 0.031$ | $0.614 \pm 0.045$ | $0.590 \pm 0.062$ |
| #Circles (↑) | $6 \pm 1$ | $3 \pm 1$ | $13 \pm 3$ | $4 \pm 1$ | $6 \pm 1$ |
| OB (1) (↓) | $98 \pm 81(10)$ | $242 \pm 160(10)$ | $18 \pm 5(10)$ | $248 \pm 81(10)$ | $52 \pm 37(10)$ |
| OB (10) (↓) | $375 \pm 131(10)$ | $797 \pm 227(10)$ | $92 \pm 29(10)$ | $515 \pm 98(10)$ | $320 \pm 63(10)$ |
| OB (100) (↓) | $769 \pm 165(10)$ | $1698 \pm 507(10)$ | $378 \pm 43(10)$ | $1101 \pm 216(10)$ | $722 \pm 140(10)$ |
| **Saturn-RNN 24.7M** | | | | | |
| Strict Hit Ratio | $50.586 \pm 9.574$ | $12.731 \pm 7.211$ | $60.331 \pm 3.294$ | $32.380 \pm 8.503$ | $51.819 \pm 7.247$ |
| IntDiv1 (↑) | $0.654 \pm 0.023$ | $0.642 \pm 0.042$ | $0.719 \pm 0.018$ | $0.636 \pm 0.030$ | $0.693 \pm 0.027$ |
| #Circles (↑) | $8 \pm 2$ | $4 \pm 1$ | $25 \pm 5$ | $7 \pm 1$ | $12 \pm 2$ |
| OB (1) (↓) | $126 \pm 99(10)$ | $384 \pm 289(10)$ | $27 \pm 19(10)$ | $186 \pm 170(10)$ | $50 \pm 52(10)$ |
| OB (10) (↓) | $465 \pm 71(10)$ | $1243 \pm 273(10)$ | $111 \pm 41(10)$ | $714 \pm 214(10)$ | $305 \pm 100(10)$ |
| OB (100) (↓) | $1045 \pm 148(10)$ | $2150 \pm 311(10)$ | $487 \pm 61(10)$ | $1404 \pm 269(10)$ | $935 \pm 130(10)$ |

OB). Next, we investigate architecture scaling again, but this time, under the strict filter. Decoder transformer (25.3M) approaches Mamba (5.2M) performance and outperforms it in many tasks (Fig. 31), trading off even more diversity. Variance is also higher. However, we believe this is an interesting observation as Augmented Memory's mechanism is squeezing the likelihood of augmented sequences. By simply scaling up the architecture and enabling the model to converge to this distribution, sample efficiency improves. This directly draws parallel to NLP LLMs where scaling improves downstream performance on many tasks, when trained on next token prediction (Wei et al., 2022). Finally, while scaling up the architecture to the parameter counts we have investigated adds negligible generation time, Mamba (5.2M) is *parameter-efficient* in its synergistic behavior with Augmented Memory.

## F.7    QUALITATIVE SUPPLEMENTARY RESULTS

In this section, we show random generated molecules from Saturn that pass the Strict Filter (Fig. F8). All molecules possess QuickVina 2 (Alhossary et al., 2015) docking scores better than the median of known actives (Lee et al., 2023) while possessing QED (Bickerton et al., 2012) > 0.7 and SA score (Ertl & Schuffenhauer, 2009) < 3. We further highlight two points: firstly, there may be some

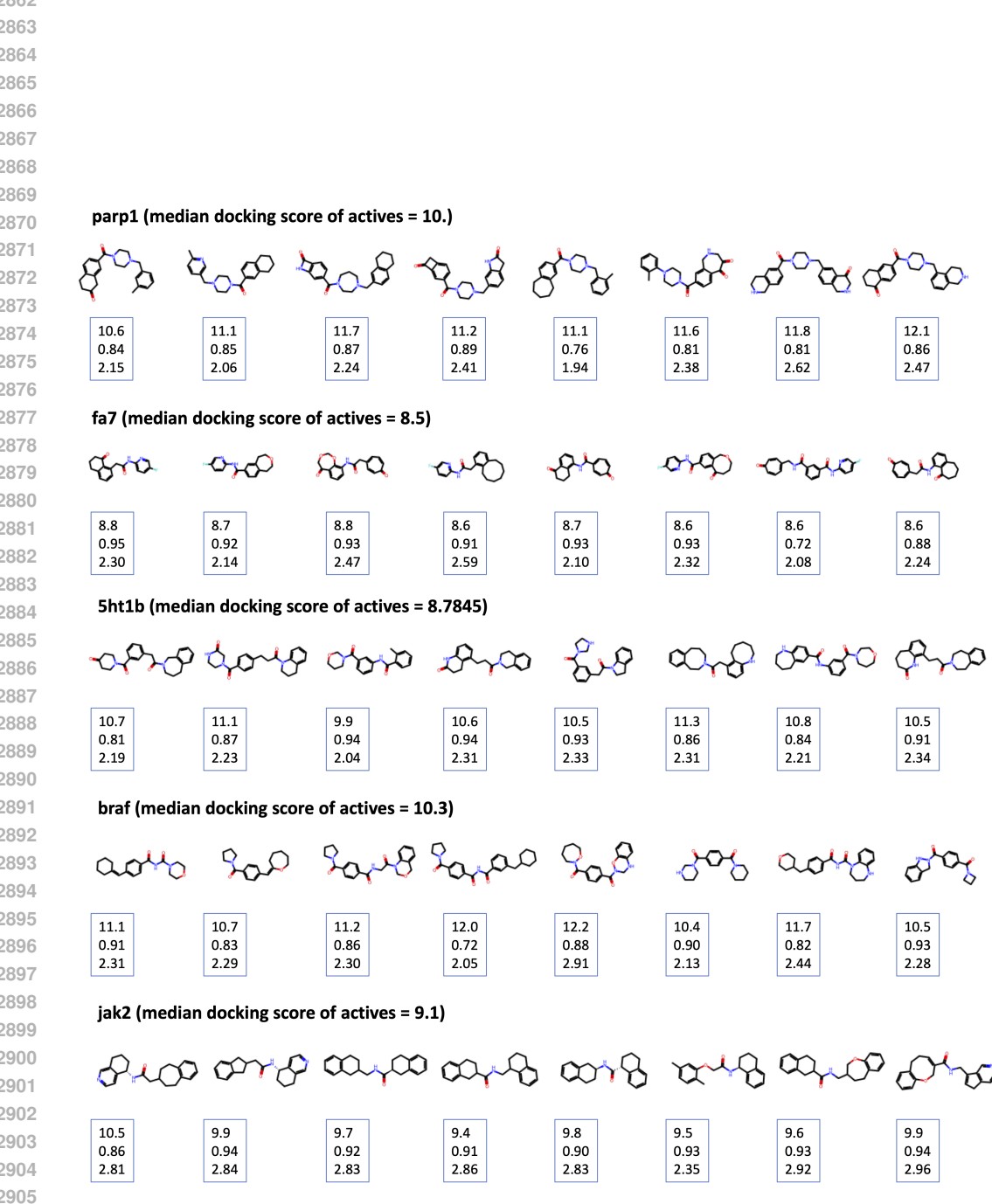

Figure F8: Example Saturn generated molecules passing the Strict Filter for all 5 targets: parp1, fa7, 5ht1b, braf, and jak2. The scores are annotated from top to bottom, QuickVina 2 (Alhossary et al., 2015) docking score, QED (Bickerton et al., 2012), and SA score (Ertl & Schuffenhauer, 2009).

particularly large rings that are undesirable from a chemistry perspective, even though QED and SA score permits them. Saturn is an optimization engine and if specific chemistry is desired, including it into the MPO objective will steer the Agent away from this chemical space. In this work, a concrete example of this is in the main text Part 3 experiments where the Saturn pre-trained model was additionally pre-trained via curriculum learning (Guo et al., 2022) to generate molecules dissimilar to the ZINC 250k (Sterling & Irwin, 2015) training data to satisfy the *Novel* metric defined Lee et al (Lee et al., 2023; 2024). This example shows the flexibility of Saturn. Secondly, as stereochemistry was not purged from the vocabulary, Saturn can generate stereoisomers.

## G    POTENTIAL CHALLENGES WHEN PUSHING TOWARDS HIGH-FIDELITY ORACLES

Throughout the main text and Appendix, we have made an effort to demonstrate Saturn's broad applicability. However, it remains to be seen whether performance will carry over to high-fidelity oracles with rougher optimization landscapes (Aldeghi et al., 2022), where the "hop-and-locally-explore" behavior may be disadvantageous. However, as we have identified *why* this behavior manifests, we can tailor the sampling behavior for the optimization landscape, if required. For example, activating the genetic algorithm and lowering augmentation rounds loosens the local sampling behavior, as shown in Appendix D.2.

