**Saturn ICLR 2025 Rebuttal**

**Preliminary results on direct optimization of Density Functional Theory (DFT) oracles**

DFT details:
- B3LYP functional
- D3 dispersion correction
- Def2-TZVP basis set

Experimental details:
- Saturn was used *out-of-the-box* (Mamba, batch size 16, 10 augmentation rounds)
- 300 oracle budget (*given the limited rebuttal time, we set a very constrained budget*)
- Goal is to minimize the HOMO-LUMO gap
- *Every generated molecule underwent full DFT geometry optimization*
- *Every generated molecule underwent DFT on a node with 72 CPU cores*

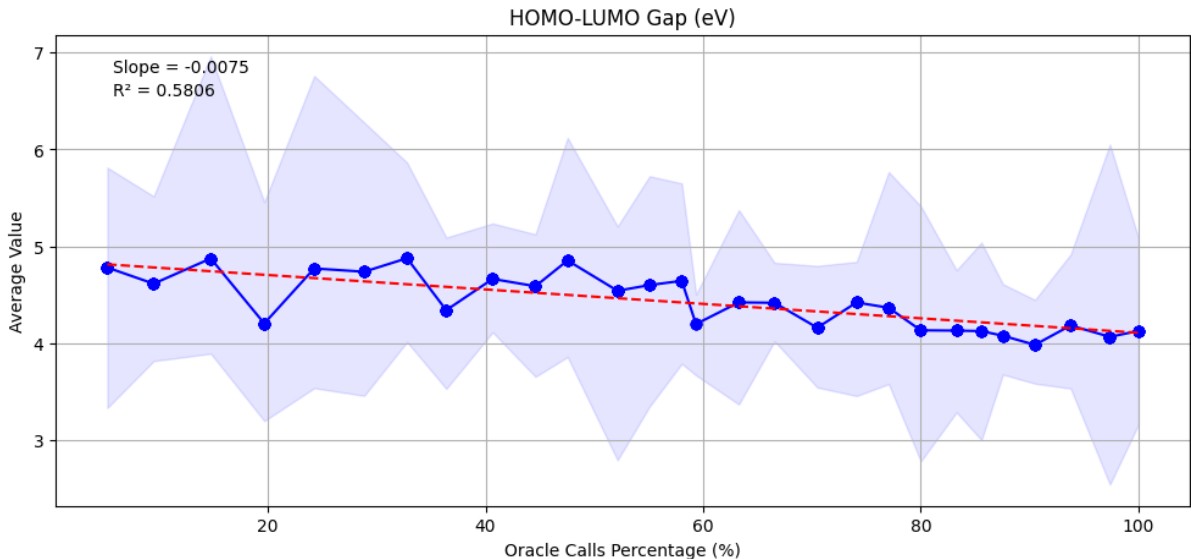

**Key points:**
1. Blue dots are the average (in the batch) HOMO-LUMO value
2. Shaded region is the min-max at every batch
3. Red line is a linear trend line showing the gradual decrease, as desired

*This is to show that Saturn's performance on comparatively lower-fidelity oracles can transfer to high-fidelity oracles. We acknowledge the results are preliminary.*

**Saturn/GEAM Ligand Efficiency and Molecular Weight**

**Definitions**
**Ligand efficiency**: docking score / number of heavy atoms. Higher ligand efficiency is desirable as it can suggest more meaningful interactions being formed between the ligand and protein.

**What we want to convey**
- GEAM generates larger molecules with lower ligand efficiency. This makes the docking scores better as it is well known that larger molecules can get better docking scores simply by forming more hydrophobic interactions. This can be problematic because unspecific binding leads to off-target toxicity.

**Analysis**
- All generated molecules across all 5 protein targets and across 10 seeds have their ligand efficiency and molecular weight computed. The results show the mean and standard deviation.

| Target | Hit Ratio (GEAM) | Hit Ratio (Saturn) | Ligand Efficiency (GEAM) | Ligand Efficiency (Saturn) | Molecular Weight (GEAM) | Molecular Weight (Saturn) |
|---|---|---|---|---|---|---|
| parp1 | 45.158±2.408 | 57.981±18.537 | 0.374±0.005 | 0.405±0.011 | 416.81±4.94 | 371.21±7.85 |
| fa7 | 20.552±2.357 | 68.185±3.4 | 0.302±0.006 | 0.408±0.013 | 417.98±7.73 | 353.17±13.71 |
| 5ht1b | 47.664±1.198 | 38.999±10.114 | 0.38±0.006 | 0.406±0.017 | 394.92±3.63 | 345.21±11.49 |
| braf | 30.444±1.61 | 14.527±9.961 | 0.384±0.004 | 0.343±0.011 | 404.93±3.15 | 365.96±17.41 |
| jak2 | 46.129±2.073 | 60.827±11.502 | 0.363±0.005 | 0.389±0.017 | 395.08±3.59 | 347.08±12.79 |

**Key points:**
1. For almost every target, Saturn generates molecules with higher ligand efficiency and lower molecular weight
2. For **braf**, GEAM generates molecules with better ligand efficiency