# OpenReview forum: "Saturn: Sample-efficient Generative Molecular Design using Memory Manipulation"
_ICLR.cc/2025/Conference — Submitted to ICLR 2025_

### Official Review · Reviewer_f1gd · 2024-11-02

**Soundness:** 3
**Presentation:** 3
**Contribution:** 2
**Rating:** 3
**Confidence:** 3

**Summary:**

This paper propose a molecular generation framework for drug discovery by combining language model and Mamba neural network. The author experimentally show that a better generative performance can be achieved by incorporating Mamba network into the language model.

**Strengths:**

The empirical evaluation is comprehensive. The author also provide open-source implementation, which enhance the credibility of the paper.

**Weaknesses:**

1. The presentation of the paper should be significantly improved. The description of the proposed framework is hard to understand. The author should at least provide a description of how the molecular generation problem is formulated as an optimization problem using clear math notation in Section 3. Many terminologies, such as "local sampling", "augmentation round", lacks clear description. The author should also include a clear description of Mamba network as most of the reader may not be familiar with it.
2. It is unclear what the novelty of this paper is. The novelty of this paper seems to be limited to the application of Mamba network to molecular generation.
3. Many model choices/claims of the proposed framework lack justification. For example, the author assume "oracle evaluations are near deterministic" without any justification. The author also need to discuss why Mamba can be potentially better compared to transformer/RNN.
4. The experiment part of this paper seems to be simply Methods X Metrics. It is unclear what the author want to show other than the empirical superiority of Mamba compared to RNN/transformer. The author claim more than 5000 experiments by taking different random seeds into account, which is quite misleading. The author also need to compare the computational consumption of Mamba compared to RNN/transformer.
5. The math notation is messy. For example, in Eq. (1): what is $x$? what is $\theta$?

**Questions:**

1. Could the author provide a more principled description of "augmented memory"? The explanation in Section 3 is hard to understand.
2. The Saturn perform really bad compared to GEAM and the author improve Saturn by finetune the model with dissimilar data. Do the authors do the same for GEAM?

---

> ### Author Response · Authors · 2024-11-15
> **Author Response (1/1)**
>
> We thank the reviewer for their feedback and the detailed questions around areas of insufficient clarity. For questions that we address in the General Response, we will explicitly note this.
>
> ## **Q1: Math notation and terminology lacks clarity**
>
> We apologize for the insufficient clarity. We have added explanations in the main text around “local sampling” and “augmentation rounds” which we have highlighted in the updated manuscript. Regarding the math notation for the optimization problem, Saturn operates in a reinforcement learning framework where the optimization goal is to maximize the expected reward (equation 2 in the main text). In Appendix B.4, we show that maximizing the reward is equivalent to minimizing equation 4 in the main text. We could not include this full section in the main text due to space constraints.
>
> ## **Q2: Mamba network details**
>
> We will include this information in the Appendix in the next version of the manuscript (next few days). We wanted to first release initial responses.
>
> ## **Q3: What is the novelty of this work?**
>
> Please see **General Response 4.**
>
> ## **Q4: Why are oracle calls deterministic? Why is Mamba better than RNN/decoder transformer?**
>
> Physico-chemical property oracles are inherently deterministic. For example, in the QED oracle which includes molecular weight and number of hydrogen bond donors, this never changes. These are intrinsic molecular properties. Where there can be stochasticity is in simulation oracles, such as docking, as used in the work. In docking, a search over conformation space is done which can have stochasticity. We minimized this by fixing the seed and we have added a passage to explicitly say this in the updated manuscript. For a discussion on Mamba compared to RNN and decoder transformer, please see **General Response 4 Sub-response 2.**
>
> ## **Q5: What are we trying to convey in the paper? Computational consumption of Mamba vs. RNN/decoder transformer**
>
> For the findings in this paper, please see General Response 4. There is essentially no difference in the computational consumption of these models because we operate in such small model sizes. The only difference is pre-training, for which Mamba and Decoder transformer can take 1-2 hours longer but this is done once. For sampling, it is true that RNN samples faster than Mamba and Decoder transformer but the difference is milliseconds so practically negligible. This is true in our setting because we are modeling small molecule SMILES sequences (< 80 tokens typically). These are extremely short sequences when put in the context of LLM modeling.
>
> ## **Q6: Messy math notation**
>
> In the preceding text to equation 1, we define x as a SMILES sequence and theta as the parameters of the Agent (the generative model).
>
>
> ## **Q7: Better description of Augmented Memory**
> Please see **General Response 4 Sub-response 1.** We have added this extra clarity in the Appendix.
>
> ## **Q8: Saturn performs worse than GEAM and then uses dissimilar data fine-tuning but does not do the same for GEAM**
>
> Please see **General Response 2.**
>
> \
> \
> We are thankful for the reviewer’s time and apologize for any insufficient clarity. We hope to have answered their questions. We kindly ask the reviewer to consider raising their evaluation of our work.
>
> Please let us know if you have any questions!

---

> > ### Comment · Reviewer_f1gd · 2024-11-20
> >
> > I would like to thank the author for detailed response.
> >
> > After reading the rebuttal, I find that many of my initial concerns remain unresolved:
> >
> > The methodology section remains unclear:
> > 1. Mamba is still not described in sufficient detail.
> > 2. Technical concepts such as Augmented Memory, Genetic Algorithm, and Oracle Caching lack proper definitions and explanations.
> >
> > This manuscript feels more like an experimental report than a complete technical paper. A self-contained approach is essential—your paper should not depend on readers having to consult external references to understand the methods. I recommend revisiting your baseline method, GEAM, which provides a clear and structured description.
> >
> > Due to the lack of clarity in the method description, it remains challenging to identify the technical contributions of the paper.
> >
> > Thus, I decide to keep my initial score.

---

> > > ### Author Response · Authors · 2024-11-20
> > > **Response to Reviewer f1gd**
> > >
> > > Thank you for the reply and we appreciate the comments to make our work more self-contained.
> > >
> > > Regarding Mamba details, we are in the process of adding information about the architecture in the Appendix. Augmented Memory is described in more detail in the newly expanded **Appendix C.2.** More information on the actual optimization algorithm is presented in **Appendix B.4**. The genetic algorithm is GraphGA [1] which is described in **Appendix B.3**. We note that GEAM also uses GraphGA. Oracle caching is described in **Appendix B.2**.
> > >
> > > We could not include all this information in the main text due to space limits. We appreciate that the reviewer thinks our work is more an experimental report but we hope with the above Appendix sections, that the technical information is conveyed. Finally, we believe strong empirical results should still be viewed as a valuable contribution. In the end, all papers in the field are trying to make generative design more practical and aligned with real world use. We believe Saturn provides interesting insights into sample efficiency as detailed in **General Response 4**.
> > >
> > > Thank you for your time and we are eager to continue discussion.
> > >
> > > [1] https://pubs.rsc.org/en/content/articlelanding/2019/sc/c8sc05372c

---

> > > > ### Author Response · Authors · 2024-11-22
> > > > **Mamba details**
> > > >
> > > > Thank you to the reviewer for stating a description of Mamba is necessary. We agree and apologize for this missing information. In the updated manuscript, we included a new Appendix section contrasting key operation differences between how embedded sequences (SMILES in our case) are passed in transformer and Mamba.

---

> > > > > ### Comment · Reviewer_f1gd · 2024-11-25
> > > > >
> > > > > Thank you for the detailed responses!
> > > > >
> > > > > While several of my initial concerns have been addressed, a few critical points still remain:
> > > > >
> > > > > 1. Some concepts, such as “local sampling,” are still difficult to understand. These terms do not seem to be fundamental machine learning concepts that can be left unexplained, and I recommend providing clearer definitions.
> > > > >
> > > > > 2. As a technical paper, the presentation remains challenging to follow. I suggest restructuring the methodology section by moving essential components, such as the Mamba network/RL formulation, into the main text to improve clarity. Additionally, consider emphasizing the primary technical differences from previous work and relocating details like sensitivity analysis and hyperparameter tuning to the appendix.
> > > > >
> > > > > 3. Due to these presentation challenges, it is difficult to assess the technical contribution of the paper. The application of the Mamba network to molecular generation seems to be the central contribution, while Augmented Memory appears to be more of a data augmentation technique than a novel contribution.
> > > > >
> > > > > 4. Finally, I agree with Reviewer XhFy’s view that ICLR may not be the best venue for this work. While strong empirical results are valuable, the combination of existing methods applied to a specific application may limit its appeal to the ICLR audience. A different venue might better suit this work and reach a more appropriate audience.
> > > > >
> > > > > Based on the points noted above, I have increased my score for Soundness and Presentation to 3, but I am keeping my initial overall rating at 3.

---

> > > > > > ### Author Response · Authors · 2024-11-26
> > > > > > **Response to Reviewer f1gd**
> > > > > >
> > > > > > Thank you for replying and for your additional comments! Your feedback helps us with how the paper is presented. We have uploading an updated manuscript draft following your suggestions to move some Mamba details to the main text. We then state how the Mamba model is connected to the RL framework. Next, we added a new section to be explicit on the differences in this work compared to previous work. **All changes are highlighted in the PDF.**
> > > > > >
> > > > > > **1.** In the new added section, we add a further explanation of what we mean by "local sampling". We hope this makes the intended message more clear.
> > > > > >
> > > > > > **2. and 3.** We have restructured the Methods section to incorporate this feedback. Thank you for this.
> > > > > >
> > > > > > **4.** We appreciate your view and also reviewer XhFy's view. We only want to convey that in our benchmark experiment against 22 models, many of these were published at NeurIPS, ICLR, and ICML in recent years and they are all tackling the same optimization tasks. In this work, we have shown *how* sample efficiency can be improved which we believe is generally useful and interesting to the field. The same concepts around "local sampling" can inform future method development. In fact, recent work [1] developed a model specifically for this "local sampling" behavior and this model is used for real world drug discovery.
> > > > > >
> > > > > > \
> > > > > > \
> > > > > > We hope our response at the very least improved the presentation of the paper for the reviewer and we remain very eager to continue discussion.
> > > > > >
> > > > > > \
> > > > > > \
> > > > > > [1] Mol2Mol local sampling: https://www.nature.com/articles/s41467-024-51672-4

---

> > > > > > > ### Comment · Reviewer_f1gd · 2024-11-26
> > > > > > >
> > > > > > > Thank you for your response.
> > > > > > >
> > > > > > > While some aspects are clearer, I still have a few remaining questions and suggestions:
> > > > > > >
> > > > > > > 1. Although I still don’t fully understand the concept of local sampling, I will defer to the other reviewers if they find it sufficiently clear.
> > > > > > >
> > > > > > > 2. I appreciate the restructuring of the method section, as it is now more readable. However, after reviewing it again, I have a few questions:
> > > > > > >
> > > > > > > * Eq(1) if $\bar{A}, \bar{B}, \bar{C}, \bar{D}$ are learnable matrices, what do $\bar{B}(x), \bar{C}(x)$ mean?
> > > > > > > * Based on Eq(1) it seems that Mamba is not significant to RNN. Can the author explain the difference?
> > > > > > > *  Eq (1) what's the difference between $x$, $s$ and $a$? Why don't use the same notation? Is $P$ the same as $\pi_{\theta}$?
> > > > > > > *  Eq (2) and (3) same question: what's the difference between $R(x)$ and $R(a, s)$?
> > > > > > > *  Eq (3) Are the authors defining $R$ or $\pi_{\text{Augmented}}$ here? If it’s $\pi_{\text{Augmented}}$, how is $R$ defined?
> > > > > > >
> > > > > > > 3. I appreciate the authors for comprehensive benchmark results, but I also want to note that:
> > > > > > > * The cited works [1][2] are well-motivated, clearly presented, and have substantial technical contributions. However, the technical contribution of this paper, beyond the use of the Mamba network, remains unclear. Reviewer XhFy and Reviewer u4yG have also raised this concern. The presence of papers on similar tasks in ICML/NeurIPS/ICLR does not necessarily justify this paper’s suitability for ICLR.
> > > > > > > * Additionally, many of the 22 benchmark results appear to be adapted from [1] and [2].
> > > > > > >
> > > > > > > 4. Some other suggestions:
> > > > > > > * I recommend replacing some of the tables with line charts to better illustrate the trends in hyperparameter performance.
> > > > > > >
> > > > > > > [1] Lee, Seul, Jaehyeong Jo, and Sung Ju Hwang. "Exploring chemical space with score-based out-of-distribution generation." International Conference on Machine Learning. PMLR, 2023.
> > > > > > >
> > > > > > > [2] Lee, Seul, et al. "Drug discovery with dynamic goal-aware fragments." arXiv preprint arXiv:2310.00841 (2023).

---

> ### Author Response · Authors · 2024-11-27
> **Response to Reviewer f1gd**
>
> Thank you for replying. Our comment on similar tasks at ICML/NeurIPS/ICLR was only meant to convey that newly developed methods are tackling the same tasks. We agree the cited works have technical contributions and we did not comment otherwise.
>
> In the works cited by the reviewer, they are by the same authors. [1] **(MOOD)** proposed the benchmark tasks and [2] **(GEAM)** further develops a method that largely outperforms their previous work. The authors themselves state using these tasks to validate their approach and these are the same tasks we compare to. Since our approach achieves strong performance and can outperform these methods, we believe there are valuable insights our work offers, particularly given the fact that GEAM’s results, when first published, were **many** times better than the 2nd best approach (which was their previous work, MOOD) - GEAM is empirically extremely strong and we have cited this as the reason we compared to these tasks. We studied the factors that contribute to Saturn’s sample efficiency and we believe this constitutes a notion of novelty.
>
> However, we respect the reviewer’s perspective on novelty and regardless of whether the reviewer increases their rating or not, we will focus our effort on conveying our findings and continuing discussion, as it is helpful for us.
>
> ### 1. Local sampling
>
> It is important for us to be able to convey this. We will attempt to explain the concept from another angle. We forget about molecules for now and consider the general task of modelling sequences. Consider a pre-trained model, trained by maximum likelihood estimation. Sampling from this model samples from the distribution of the dataset. Now suppose we have a small set of *other* sequences which we want to use for transfer-learning/fine-tuning. The process of fine-tuning shifts the distribution to this set of *other* sequences. Successful fine-tuning means that generating these *other* sequences is now more likely. We now draw the connection back to molecules. Consider now that these *other* sequences are the SMILES sequences in the replay buffer. The process of Augmented Memory can be thought of as transfer learning using the replay buffer sequences. Similar to the general case, successful fine-tuning means it is now more likely to generate these specific sequences. We now add SMILES augmentation: each sequence in the replay buffer is augmented $N$ times resulting in now a larger set of sequences for fine-tuning. Similar to the general case again, fine-tuning on this larger set makes the model more likely to generate these sequences. The caveat here is that augmented sequences map to the same *molecular graph*. Given a molecular graph with N augmented SMILES forms, generating any of these SMILES maps back to the same molecule. It follows that if some tokens are swapped/added/inserted during the generation of any of these augmented SMILES forms, it will map to a molecule which differs by a few atoms because each token loosely maps to an atom. This is also qualitatively shown in Figure D6 where a notable portion of molecules sampled at similar model states is quite similar. Local sampling refers to sampling similar molecules to the replay buffer molecules. This is akin to sampling in the "chemical space" of the replay buffer molecules.
>
> As a final note, why do we apply augmentation? Why not just learn from the exact sequence form without augmentation? Without augmentation, detrimental mode collapse occurs where the model generates the exact same sequences. When this occurs, the generative ability is lost.

---

> ### Author Response · Authors · 2024-11-27
> **Response to Reviewer f1gd (continued)**
>
> ### 2. Equations
>
> We realized that we missed adding a label to the Mamba equations in the main text, sorry for this.
>
> * **Eq(1) if $\bar{A}$, $\bar{B}$, $\bar{C}$, $\bar{D}$ are learnable matrices, what do $\bar{B}(x)$ and $\bar{C}(x)$
>  mean?**
>
> Mamba makes $\bar{B}(x)$ and $\bar{C}(x)$ input ($x$)-dependent  so the model can “select” what context to propagate forward. We adopted this notation from [1] which we have cited in the main text and manuscript as we believe it makes it explicit that this is done.
>
> * **Based on Eq(1) it seems that Mamba is not significant to RNN. Can the author explain the difference?**
>
> Mamba has similarities to RNNs and the authors of Mamba explicitly make this comparison. Under specific initialization, algorithm 2 in the Mamba paper [2] has the form of a gated RNN. In addition to the more general formulation of selective propagation, the authors introduced a “hardware-aware” algorithm for efficient training. We adapted their optimized code from the official repository [3]
>
> * **Eq (1) what's the difference between $x$, $s$ and $a$. Why don't use the same notation? Is $P$ the same as $\pi_\theta$?**
>
> $x$ is the SMILES sequence, $s$ is the state (all intermediate token sequences), and $a$ is the action (the token selected at a given time-step). State and Action are used in RL and we introduced this at the beginning of the section. $P$ is not the same as $\pi_\theta$. $P$ is the probability of generating the SMILES sequence, $x$.  $\pi_\theta$ denotes the network that generates the SMILES. We write this in the following sentence in the text:
>
> "where $\pi_\theta$ is the Mamba backbone and referred to as the Agent to match RL terminology and $a_t$ and $s_t$ are the token selected and token sequence so far, at time-step t, respectively."
>
> * **Eq (2) and (3) same question: what's the difference between $R(x)$ and $R(a, s)$?**
>
> $R(x)$ is the return associated with a SMILES sequence, i.e., the scalar property value. $R(a, s)$ is the state-transition reward in RL, i.e., “what is the reward associated with taking the action in the given state?” In our setting, the action is the token selected and the state is the token sequence so far. In Appendix B.5, we mention that all intermediate rewards are 0 because there is only a reward for the final SMILES sequence.
>
> * **Eq (3) Are the authors defining $R$ or $\pi_{Augmented}(x)$ here? If it’s $\pi_{Augmented}(x)$, how is $R$ defined?**
>
> We are defining $\pi_{Augmented}(x)$. $R(x)$ is any arbitrary reward function that takes as input the SMILES, $x$, and returns the scalar reward. For example, in the experiments comparing with GEAM, $R$ is the docking, QED, SA score function:
>
> $R(x) = \widehat{DS}(x) \times QED(x) \times \widehat{SA}(x) \in [0, 1]$
>
> ### 3. Line plots
>
> Thank you for this suggestion, we are thinking of how to use this to convey the information better without adding too many plots.
>
>
>
> \
> \
> [1] https://jameschen.io/jekyll/update/2024/02/12/mamba.html
>
> [2] https://arxiv.org/abs/2312.00752
>
> [3] https://github.com/state-spaces/mamba

---

> ### Comment · Reviewer_f1gd · 2024-12-02
>
> Thanks for the clarification. Some comments:
>
> 1. To the best of my knowledge, if $B(x)$ and $C(x)$ are input dependent, then they should not be called *learnable matrices*. Please check carefully the implementation of Mamba and terminologies.
>
> 2. If Mamba is similar to RNN, what is the main motivation for this research? The author should check the Mamba network in more detail and provide some justification for why Mamba could potentially lead to better empirical performance on molecular generation task.
>
> 3.  x is the SMILES sequence,  s is the state (all intermediate token sequences), and  a is the action (the token selected at a given time-step). Then shouldn't it be $x = s_t$ and $x = \{a_1, a_2, \dots, a_t\}$?
>
>
> Overall, while the author achieve strong empirical results and make some progress during rebuttal, my initial concerns about clarity and technical novelty still remains. I believe a major revision is need before this paper gets published in a venue with most of the audience from machine learning community, and I decide to keep my initial rating.

---

> > ### Author Response · Authors · 2024-12-02
> >
> > 1. We based our implementation on the official Mamba codebase. In the original paper [1], the authors stated making the matrices input-dependent in Section 3.2 and shown also in Algorithm 2. In the codebase (which we import from in our code), the sequence input is passed to a Linear projection layer. In turn, $\bar{B}(x)$ and $\bar{C}(x)$ are derived from this output so we also call them learnable as their values depend on the input sequence and the Linear layer.
> >
> > 2. Under *specific* initialization, the model has the form of a gated RNN [1]. The selection mechanism of Mamba was one of the original motivations of the Mamba paper to give a notion of *context* during sequence processing (like attention in transformers). Empirically across many domains, Mamba has achieved similar/better performance than transformers. Our motivation for using Mamba was to investigate its behaviour in the RL molecule paradigm and we found its distribution learning capabilities to be advantageous in the settings we tested.
> >
> > In our above response about "local sampling" (again, we are sorry for the insufficient clarity which we hope we better conveyed now) which leads to high sample efficiency, it is very dependent on the ability to model the token sequences of all the augmented SMILES forms. A model with better capability to model sequences benefits this mechanism. This is what we show in the main text results.
> >
> > 3. We define the generation as a Markov process so the probability of the next action only depends on the previous time-step's state. The generation process is then the token selected given the state so far and the expression in equation 1 takes the product of the probabilities of these steps from time-step 1 to T.
> >
> > Thank you to the reviewer for discussing with us during this period as it was very helpful. We hope we were able to provide better clarity on certain aspects.
> >
> >
> > \
> > \
> > [1] https://arxiv.org/abs/2312.00752
> >
> > [2] https://github.com/state-spaces/mamba/blob/main/mamba_ssm/modules/block.py

---

### Official Review · Reviewer_u4yG · 2024-11-02

**Soundness:** 3
**Presentation:** 3
**Contribution:** 3
**Rating:** 5
**Confidence:** 3

**Summary:**

The paper proposes a novel framework that introduces the Mamba architecture and enhances sample efficiency through augmented memory and experience replay mechanisms, demonstrating superior performance on multi-parameter optimization tasks relevant to drug discovery compared to the baseline models.

**Strengths:**

This paper presents the first application of the Mamba architecture in generative molecular design. It uses augmented memory and experience replay to significantly enhance sample efficiency, enabling faster optimization of high-fidelity oracles. The authors conducted sufficient experiments to demonstrate the model's generative capabilities.

**Weaknesses:**

1. While Saturn excels in identifying high-reward molecules, there is a trade-off in diversity, which may affect its capability for exploratory molecular design.
2. Although this paper used Mamba for the first time in the molecular generation task, the innovation of the model is limited.

**Questions:**

1. While the authors have demonstrated the efficiency of the Mamba through the comparison of different models under the Oracle Burden (OB) condition, it would be great to evaluate and report the actual generation time for each model. Could the authors provide any insights or results regarding the runtime performance of the different models?
2. The authors should clarify why they specifically chose to maintain only the top 100 molecules in replay buffer. Additionally, does the replay buff lead to the potential issue of storing very similar molecules in the buffer, which could subsequently reduce the overall diversity of the generated samples?
3. In part 3, the authors only compare the metrics related to Hit Ratio. Is there a problem that the generated molecules may be similar? Some metrics related to the diversity of generated molecules should be added for comparison.

---

> ### Author Response · Authors · 2024-11-15
> **Author Response (1/1)**
>
> We thank the reviewer for their feedback and their interest in the prospect of optimizing MD oracles. This is also exciting for us and achieving this is a goal for us! We will answer each question individually. For questions that we address in the General Response, we will explicitly note this.
>
> ## **Q1: Trade-off in diversity may limit capability**
> Please see **General Response 3.**
>
> ## **Q2: Innovation of the model is limited**
> Please see **General Response 4** regarding the contribution of this work.
>
> ## **Q3: Report actual generation time.**
>
> The benchmarking experiments are in Part 3 of the paper. The strongest baseline we compare to is GEAM [1] and the benchmark experiment itself is based on GEAM’s case study. The optimization task is docking against 5 targets and we ran each target across 10 seeds (0-9 inclusive). The total wall time for Saturn across all 50 experiments (5 targets, 10 seeds each) which were run sequentially, was ~41.5 hours on an NVIDIA A6000 GPU. ***This information is in section F.1. in the Appendix.*** Therefore, each experiment was, on average < 1 hour.
>
> For GEAM, we now add the wall times below (***we also newly add this information to section F.2 in the Appendix***):
>
> **braf:** 3.28 ± 0.04 hours
>
> **jak2:** 3.26 ± 0.05 hours
>
> **fa7:** 3.38 ± 0.04 hours
>
> **5ht1b:** 3.17 ± 0.08 hours
>
> **parp1:** 3.02 ± 0.19 hours
>
> The mean and standard deviation are reported across 10 seeds, except for parp1 which is across 7 seeds. The remaining 3 seeds were run on CPU due to insufficient GPU resources. CPU times are much longer so we did not include these, as that would be unfair. For more transparency, In GEAM, we used NVIDIA V100 GPUs due to cuda compatibility with their codebase. NVIDIA V100 GPUs are slower than A6000 GPUs. According to this benchmark: https://www.aime.info/blog/en/deep-learning-gpu-benchmarks/, A6000 was about 50% faster. It is hard to say exactly the difference in speed for this task, though. ***However, regardless, actual generation time for Saturn should still be faster than GEAM.***
>
> ## **Q4: Why did we store the top-100 molecules in the buffer? Is lower diversity a problem?**
>
> In the original Augmented Memory [2] work, the authors found minimal difference from decreasing and increasing the buffer size. Decreased diversity during generation is ***by design*** because it can improve sample efficiency (we refer to **General Response 4 Sub-response 1**). We emphasize that we have done extensive ablations in Appendix C so if more diversity were desired, one can increase batch size and decrease augmentation rounds. The consequence of decreased diversity, however, is that Saturn may generate repeat molecules, but we introduced oracle caching such that only “new” generated molecules need to be sent to the oracle.
>
> ## **Q5: Report diversity metrics. Is low diversity a problem?**
>
> For a discussion on diversity, we refer to **General Response 3**. We want to emphasize that we are very transparent that Saturn trades off diversity for sample efficiency. Throughout the paper, we have always reported diversity metrics, specifically IntDiv1 (internal fingerprint similarity) and #Circles (packing number). For a qualitative analysis of the generated molecules, we also show examples in Fig. F7 in the Appendix.
>
> We are grateful for the reviewer’s time and hope we have answered their questions. We are especially thankful for the comments around generation time as we believe this is a very practical question. We kindly ask the reviewer to consider raising their evaluation of our work.
>
> Please let us know if you have any questions!
>
> \
> \
> [1] GEAM: https://arxiv.org/abs/2310.00841
>
> [2] Augmented Memory: https://pubs.acs.org/doi/10.1021/jacsau.4c00066

---

> > ### Author Response · Authors · 2024-11-28
> > **Eager to continue discussion**
> >
> > We want to thank the reviewer again for their time and their question on reporting run-time. We had missed reporting this for GEAM, which we have since added. This is also a very important metric for us as it is a very practical problem.
> >
> > As the rebuttal period is nearing an end, we hope that our perspective on diversity (**General Response 3**) and run-time answer the reviewer's questions and concerns. We only want to note that in the new General Response, we have highlighted that GEAM requires supervised pre-training which pre-computed all 250k molecules in ZINC 250k for their pre-training. For each task, there is also a separately trained model, so in the end, 1.25 million docking calculations (250k molecules docking against 5 protein targets) were performed for GEAM pre-training. We hope that this difference, together with the faster generation time of Saturn, is convincing to the reviewer that Saturn is a meaningful contribution towards practical sample efficiency. We are always eager to continue discussion.

---

### Official Review · Reviewer_XhFy · 2024-11-03

**Soundness:** 4
**Presentation:** 4
**Contribution:** 1
**Rating:** 5
**Confidence:** 4

**Summary:**

Motivated by the need to optimize expensive oracle functions, the authors propose Saturn, a framework for sample-efficient molecular optimization. Saturn consists of the Mamba architecture for generating SMILES, and RL with Augmented Memory for optimization. Extensive experiments show promising results on optimizing docking score compared to baselines in a sample-efficient manner.

**Strengths:**

* The motivation of the work is very clear and convincing. Inaccurate in silico oracles is one of the major limitations of generative models in practice, and so being able to utilize highly accurate but costly oracles in generative models would be very useful.
* Clear and well-written paper, including methods and results.
* Very extensive results. Comparison with ~20 state-of-the-art molecular generative models on the docking task, where Saturn appears to be significantly superior to all except GEAM.
* Meticulous ablations of each model component and architecture hyperparameters. The appendix is also very in-depth.

**Weaknesses:**

* Very limited technical novelty. Saturn seems to be the application of the previously developed Augmented Memory approach, except swapping the backbone for the Mamba architecture. Thus, while the paper contains strong empirical results, it is probably not the right fit for ICLR.
* The paper is motivated by the need to optimize the scores from high fidelity oracles, but no experiments are conducted for this setting. The results on docking are promising, but it is not clear that the superiority of Saturn will transfer to the more complex case of optimizing high fidelity oracles. For a paper that is mostly empirical, it seems important to have results on the motivating example instead of a toy example.
* The most competitive baseline, GEAM, seems to reach about similar performance to Saturn for most tasks. While the authors argue that GEAM achieves more diversity at the cost of less sample efficiency, it appears the authors did not explore tweaking the hyperparameters of GEAM. Is it possible some hyperparameters of GEAM might encourage less diversity and increase sample efficiency?

**Questions:**

* The authors state that they add a Tanimoto dissimilarity objective to Saturn to make Saturn-Tanimoto, but I don’t see a similar GEAM-Tanimoto. Wouldn’t it be possible to add the same optimization to GEAM? If so, would it be more fair to compare Saturn-Tanimoto with GEAM-Tanimoto?
* What causes the improved sample efficiency of Saturn? It’s not clear that the Mamba architecture would be particularly sample efficient, so it seems to be the SMILES data augmentation method. However, I don’t see this explained in depth anywhere in the main text, particularly how the augmentation is actually done.
* It is not clear to me in the methods which contributions are yours and which come from previous works. Could this be clarified?

---

> ### Author Response · Authors · 2024-11-15
> **Author Response (1/2)**
>
> We thank the reviewer for their feedback and their interest in the prospect of optimizing MD oracles. This is also exciting for us and achieving this is a goal for us! We will answer each question individually. For questions that we address in the General Response, we will explicitly note this.
>
> ## **Q1: Limited technical novelty**
>
> Please see **General Response 4.** We believe the insights into underlying mechanisms of molecular optimization and its effect on the generated molecules are valuable to the field. Many of the models we compared to were published in recent years at NeurIPS, ICLR, and ICML. **GEAM [1] (the strongest baseline) was published at ICML 2024.** In the general response, we hope to have conveyed the contributions of Saturn. We appreciate that the reviewer believes the work is mostly empirical. We only want to comment that in the end, developing generative models should be based on their utility in real world tasks. The most important empirical result is that Saturn can optimize multi-parameter optimization tasks under minimal oracle calls (**we refer to the Strict Hit Ratio results, where GEAM performs considerably worse**), which is a very important practical problem. We hope this, together with showing direct DFT optimization, is convincing to the reviewer that strong empirical results are a valuable contribution as we work towards making generative models more capable for real life applications.
>
> ## **Q2: No high-fidelity oracle experiments**
>
> Please see **General Response 1** for an example of direct high-fidelity oracle optimization. We emphasize that moving to DFT/MD oracles is an enormous jump (going from CPU/GPU minutes to ***hours***). We also do not think docking is a toy example as it is used in every single structure-based drug discovery generative campaign that has achieved experimental validation [2]. In fact, in commercial drug discovery, a generated molecule has just (***as of this week***), achieved good results in Phase IIa clinical trials. The design of this molecule used docking [3]. Moreover, certain parameters in docking can be relatively computationally expensive, such as allowing for some flexible residues and performing more exhaustive pose searching. A concrete example is gnina [4] docking. The developers recently achieved very good results on the CACHE challenge (given a target, propose molecules that the CACHE organization synthesizes and tests). ***We give these specific examples to reinforce that docking is not a toy example, and they are actually used in real life.***
>
> ## **Q3: Saturn has similar performance to GEAM and why hyperparameter tuning of GEAM was not performed**
>
> We refer to **General Response 2** regarding the performance comparison between Saturn and GEAM and that Saturn optimizes the multi-parameter optimization objective to a much greater degree (Strict Hit Ratio in the main text). ***Importantly, GEAM computed the oracle values for the entire ZINC 250k dataset to pre-train. This imposes an up-front cost of 250k oracle calls.***
>
> Regarding hyperparameter tuning of GEAM [1], in the GEAM paper, **their main results** are the docking tasks against **parp1**, **fa7**, **5ht1b**, **braf**, **jak2**. While not explicitly stated in their paper, it is reasonable to believe the authors already tuned their model on this task. In our Part 3 experiments, we mimicked GEAM’s experiments **exactly** (the pre-training data and oracle code was directly taken from their codebase). ***We want to strictly emphasize that we did not tune Saturn on this task.*** We only investigated Saturn’s hyperparameters in the Part 1 toy task and then fixed them for all docking experiments in the paper (Part 2 and Part 3 results). We explicitly note this at the end of Part 1 and we copy the text here:
>
> ***“From here on, this model configuration will be referred to as Saturn and hyperparameters are fixed such that all performance metrics in the following sections are out-of-the-box”***
>
> We now want to comment on hyperparameter tuning. Given an optimization objective, generative models are directly applied to generate suitable molecules. In commercial drug discovery, models cannot be tuned for every single optimization task. This is even more problematic when the oracle is computationally expensive and makes it impractical to tune hyperparameters for every objective. Assuming GEAM did not tune their hyperparameters on their task, tuning them would likely improve performance but the same could be said for Saturn.

---

> ### Author Response · Authors · 2024-11-15
> **Author Response (2/2)**
>
> ## **Q4: Saturn-Tanimoto/GEAM-Tanimoto**
>
> We refer to **General Response 2.**
>
> ## **Q5: Why is Saturn sample-efficient? How is data augmentation performed?**
>
> Regarding why Saturn is sample-efficient, we refer to **General Response 4.**
>
> We apologize for the insufficient clarity around data augmentation. In Saturn, data augmentation is SMILES augmentation. SMILES are string-based representations of molecules and Saturn, using a Mamba backbone, learns a token distribution of these strings. Given a molecular graph, where the nodes are atoms and edges are bonds, a corresponding SMILES string can be obtained by designating a “start node (atom)” and performing a depth-first search (DFS). The node traversal order yields a SMILES representation. If the starting node were changed, a different SMILES representation results. This is SMILES augmentation. From an implementation perspective, we use RDKit to shuffle atom numbers of the Mol object to return different SMILES representations of the same molecular graph. ***We have included this excerpt in the newly uploaded manuscript highlighted in Appendix C.2.***
>
>
> ## **Q6: Which contributions are ours?**
> Please see **General Response 4.** From the methods section, the Mamba backbone, oracle caching, and genetic algorithm are new. However, we believe that the important contribution from our work is understanding optimization dynamics in generative design and how to predictably tune Saturn to yield different sampling behavior/control the types of chemistry being generated.
>
>
> We are grateful for the reviewer’s time and hope we have answered their questions. We hope to have conveyed that Saturn is a valuable step towards high-fidelity oracle optimization and kindly ask the reviewer to consider raising their evaluation of our work.
>
> Please let us know if you have any questions!
>
> \
> \
> [1] GEAM: https://arxiv.org/abs/2310.00841
>
> [2] Review Paper: https://www.nature.com/articles/s42256-024-00843-5
>
> [3] TNIK Phase 2: https://www.nature.com/articles/s41587-024-02143-0
>
> [4] gnina: https://jcheminf.biomedcentral.com/articles/10.1186/s13321-021-00522-2

---

> > ### Comment · Reviewer_XhFy · 2024-11-26
> >
> > Thank you for the response, especially the additional results on DFT oracle optimization. I think the results are strong, and empirically valuable. Unfortunately, many of my same concerns remain.
> >
> > Q1: I read General Response 4, and agree with the authors that Saturn has strong empirical results. Nonetheless, I still think there is limited technical novelty. I will leave it up to the AC to decide if the empirical results are sufficient.
> >
> > Q2: The DFT results are strong, but I don't think they can be used to say Saturn is likely to also perform well in optimizing high-fidelity binding affinity oracles. This is because it is relatively easy to minimize HOMO-LUMO by adding chemical groups (e.g. electron-rich groups), while optimizing binding affinity is more complex. Just because it is expensive to compute does not necessarily mean it is a good analog to binding affinity optimization. I do acknowledge the authors' point that docking is not a toy example, but nonetheless I think if the main motivation of the paper is to go beyond docking then it shouldn't only use docking as an oracle.
> >
> > Q3: An up-front cost of 250k oracle calls is indeed highly significant, and a strong reason to prefer Saturn over GEAM. However, do the authors have any results to say that pretraining GEAM with less data (e.g. only 1k oracle calls) performs much worse? Additionally, it would be appreciated if the authors could discuss if large datasets of pre-computed costly oracle data (e.g. binding free energy calculations) are applicable for GEAM pre-training. If so, that might mean the pre-training of GEAM is not as big of a problem, since I believe such datasets exist.
> >
> > Q4, Q5, Q6: Thank you for the responses, they address my questions.
> >
> > Overall, I lean towards keeping my initial score for now.

---

> ### Author Response · Authors · 2024-11-26
> **Response to Reviewer XhFy**
>
> Firstly, thank you for reading our General Response, we know it is long. We would like to follow-up with additional explanations and motivations.
>
> ### Q1: Limited technical novelty
>
> We appreciate the reviewer's perspective and only want to highlight the **General Response 4** again and provide additional explanation on why we believe elucidating the mechanism of Augmented Memory and understanding the sampling behavior of Saturn is novel. We begin back in 2018 (early days for generative *molecular* design), when [1] investigated different reinforcement learning algorithms for molecule optimization. Unexpectedly (also according to the original authors' words in the paper), Hill-Climbing (fine-tune the model with the top-k molecules at every generation epoch) was the most performant optimization algorithm. In the original REINVENT in 2017 [2], the idea of a replay buffer for their reinforcement learning algorithm was already implemented but not commented on. In the follow-up publication of REINVENT 2.0 in 2020 [3], the replay buffer was explicitly commented on and is actually a "Hill-Climbing Replay Buffer" because it stores the top-N molecules generated so far in the entire run (this is also what Saturn uses). In 2022, [4] benchmarked many molecule optimization methods and found that Hill-Climbing methods were amongst the most performant and REINVENT, the highest performing model at the time, uses it (in the replay buffer). One general observation was that more sample-efficient methods have lower diversity. We will come back to this soon. In the same year, [5] proposed a modified REINVENT that integrates Hill-Climbing also to the optimization directly. Now, this model uses Hill-Climbing by taking the top-k molecules at every generation epoch **and** continues to store the top-N molecules generated so far. In this work, the same observation was made: trading diversity for sample efficiency. At ICLR 2024 [6], the authors again made the same observation. If trading off diversity leads to sample efficiency, this implies that generating more similar molecules is beneficial. ***But similar to what?*** In Saturn, through the ablations on batch size and augmentation rounds, we show **explicitly** that these parameters directly control local sampling and that specifically sampling near the chemical space of the replay buffer molecules (***which are the top-N molecules generated so far***), improves sample efficiency. In Fig. D6 in the Appendix, we qualitatively show what this entails for the generated molecules - they differ by small atom changes. Perhaps this is obvious in hindsight as it is generally hypothesized that "similar molecules exhibit similar properties", so just generating similar molecules to "good" molecules will be beneficial, but we do not think it is obvious that such *extreme* local sampling (using Mamba) is actually tolerable and is performant for **all** the docking case studies. In the end, by performing so many ablations on Saturn, we can predictably control the sampling behavior and actually know *a priori* what to expect for the molecules in the generated set. We also do not think that this aggressive local sampling will be beneficial for ***all*** optimization tasks but we know how to control this behavior. We think this is a novel contribution and we shared some thoughts on how this can inform future research in **General Response 4 Sub-response 3.**
>
> \
> \
> [1] Hill-Climbing: https://openreview.net/forum?id=Bk0xiI1Dz
>
> [2] REINVENT 1.0: https://jcheminf.biomedcentral.com/articles/10.1186/s13321-017-0235-x
>
> [3] REINVENT 2.0: https://pubs.acs.org/doi/10.1021/acs.jcim.0c00915
>
> [4] PMO: https://arxiv.org/abs/2206.12411
>
> [5] Augmented Hill-Climbing: https://jcheminf.biomedcentral.com/articles/10.1186/s13321-022-00646-z
>
> [6] Hill-Climbing ICLR 2024: https://openreview.net/forum?id=nqlymMx42E

---

> > ### Author Response · Authors · 2024-11-26
> > **Response to Reviewer XhFy (continued)**
> >
> > ### Q2: DFT is not binding affinity
> >
> > We agree with the reviewer that the DFT results do not guarantee Saturn can also optimize high-fidelity binding affinity oracles. We want to note we were very prudent in writing **General Response 1**. We only said that Saturn ***can*** transfer to high-fidelity oracles and that it is a ***valuable step*** towards this prospect. We also agree that HOMO-LUMO gap is not the most challenging property and we chose this because we can qualitatively verify the generated molecules by the changing presence of electron groups during optimization, under this very constrained rebuttal period. However, we wish to say that DFT is still a high-fidelity oracle and we have optimized the ***full geometry*** of every single generated molecule. The optimization actually can extract many other properties in the ORCA output that is not HOMO-LUMO gap. We note that very recent work [1] actually uses HOMO-LUMO to inform an experimentally validated generative design.
> >
> > Another important factor we want to reinforce is that **Saturn can use a batch size of 16.** As briefly discussed in **General Response 1**, we want to batch compute (all at once) the oracle values of the generated molecules and not sequentially, otherwise this would be time prohibitive. In order to even run the DFT full optimization, we had to parallelize every single molecule on 72 CPU cores. **16 molecules = 16 nodes with 72 CPU cores each.** This can quickly become a practical resource limitation problem with larger batch sizes and binding affinity oracles that may need GPUs. Finally, to convincingly demonstrate high-fidelity binding affinity oracle optimization, we are undergoing careful validation to first show that the simulation is more accurate than a carefully designed docking protocol. For example, we refer to [2] to show that docking *can* be decently correlated with binding affinity. [3] also uses GPU-docking that can control the flexibility of residues (much higher oracle cost) and has been experimentally validated. Sample efficiency will also benefit such a setting (much more expensive docking).
> >
> > We appreciate the reviewer's comment about showing high-fidelity binding affinity optimization but to properly do this and also compare with other methods, it requires an **enormous** amount of GPU resources. It is an **enormous** jump from docking to MD, much larger than cheap physico-chemical properties like QED to docking.  We believe showing that Saturn can outperform many other models including GEAM which uses supervised pre-training is a valuable contribution as the results suggest that these other models are not close to such a prospect.
> >
> >  \
> > \
> > [1] HOMO-LUMO STAT3 Inhibitors: https://www.researchsquare.com/article/rs-5213622/v1
> >
> > [2] DockStream: https://jcheminf.biomedcentral.com/articles/10.1186/s13321-021-00563-7
> >
> > [3] CACHE gnina: https://cache-challenge.org/challenges/app/63160acad0a52

---

> ### Author Response · Authors · 2024-11-26
> **Response to Reviewer XhFy (final continued)**
>
> ### Q3: Pre-training GEAM with less data (1k oracle calls) and large dataset pre-training
>
> GEAM uses a "Fragment-wise Graph Information Bottleneck" (FGIB) that predicts the oracle value given a noise-subgraph (this is how they extract *informative* fragments). We do not have results with less training data as we reproduced GEAM exactly as in their provided code. However, this FGIB module is responsible for extracting the "good" fragments so with much less data, there would be expectedly less "good" fragments and performance would decrease.
>
> Yes, a large dataset of pre-computed oracle data could be used but we want to emphasize that GEAM requires a separate model for every optimization task (**there were actually 5 separately trained models, 1 for each protein target**). This is because for every different task, GEAM requires labelled dataset of that particular optimization property (in total they pre-computed 1.25 million docking oracle calls - 250k molecules * 5 protein targets). There is no "general pre-training" that can be done so a large pre-computed dataset would only be applicable for that exact property.
>
> We end this response with an artificial experiment to illustrate how pre-computing a dataset like in the case of GEAM can be seen as "pre-selecting training data" since the model knows which molecules in the pre-training dataset are better. We will illustrate how this can greatly benefit Saturn.
>
> Consider a toy experiment where we want to generate molecules > 1000 molecular weight (MW). We will take ChEMBL 33 and slice it into 2 fractions and pre-train Saturn on both fractions. Then we will use 1,000 oracle calls to try to generate > 1000 MW molecules (**we perform 100 replicates since this is a cheap experiment**). We provide some statistics on the pre-training fractions to show that the quantity and diversity are comparable, to limit confounding variables. `Yield` is how many unique molecules > 1000 MW were generated.
>
>
>
> ### ChEMBL 33: 1,942,081 total molecules
>
> ### MW < 375
> | Metric     | Value               |
> |------------|---------------------|
> | N          | 969,971            |
> | IntDiv1    | 0.890              |
> | #Circles    | 6235              |
> | Yield      | 0     (**all 100 replicates failed**)          |
> | Max MW     | 509.28 ± 20.77  (**harder to shift the distribution**)   |
>
> ### 375 < MW < 600
> | Metric     | Value               |
> |------------|---------------------|
> | N          | 972,110            |
> | IntDiv1    | 0.879              |
> | #Circles    | 5531              |
> | Yield      | 12 ± 13 (**98/100 replicates were successful**)   |
> | Max MW     | 1,114.24 ± 77.21   |
>
> By training on a "better" fraction, i.e., more suitable for the optimization task, that very same optimization task becomes much easier, as expected. This is because we already "pre-curated" the dataset. In some sense, this is what GEAM's supervised pre-training is doing. It is a **huge** advantage to already know what the best molecules in the pre-training dataset are. In practice, this cannot be done for any arbitrary property because it is expensive to pre-compute. This is a toy example because MW is extremely cheap to compute and enables us to convey this message.
>
> \
> \
> Thank you to the reviewer for replying to us and we are very eager to continue the discussion. Constructive criticism is useful for us to improve.

---

> > ### Comment · Reviewer_XhFy · 2024-12-02
> >
> > Thank you for the reply! I agree that going from docking/DFT to MD is a big step, and the validation you're conducting to show that MD is more accurate than docking definitely makes sense.
> >
> > In response to Q3, thank you for clarifying that GEAM's pre-training must be done with oracle calls on the same protein target that one is generating molecules for. I think I misunderstood initially. If I'm now understanding correctly, GEAM is not at all sample-efficient because it requires a large dataaset of precomputed oracles. Thus, Saturn seems to be clearly better in the case of expensive oracles. Because of this, I raise my score from 3 -> 5, as Saturn seems to be clearly superior than existing work on the important case of optimizing expensive oracles. I did not raise my score any higher, though, because my concerns about the technical novelty remain even after reading the author's response to Q1.

---

### Official Review · Reviewer_feFf · 2024-11-04

**Soundness:** 3
**Presentation:** 3
**Contribution:** 2
**Rating:** 6
**Confidence:** 2

**Summary:**

- This paper tackles the problem of sample efficiency in chemistry generative models.
- It proposes to use experience replay and data augmentation to train a Mamba model and show its sample efficiency compared to other state-of-the-art molecular generative models.

**Strengths:**

- The manuscript is well written and was easy to follow. The appendix is extensive and provides enough details on each component of the method.
- The experiments are well thought and extensively demonstrate most of the laid out claims.

**Weaknesses:**

- The authors have claimed that promised candidates from docking are subjected to MD simulation which is a higher fidelity oracle and direct optimization for higher fidelity oracle may be possible with sample efficient generative models (Saturn). However, this paper remains short of demostrating the utility of Saturn on MD simulations and leave it as future work. It is not clear if the sample efficiency gains observed in case of a comparatively lower-fidelity oracle (docking) would translate to high-fidelity oracle (MD). For the sake of completeness and justification of laid out claim, I would suggest that the authors provide some preliminary results on application of Saturn for MD.

**Questions:**

- On the effect of diversity filter in augmented memory: Does setting the reward to 0 and removing all instance of over-represented scaffolds in the buffer, limits the generator from learning from potentially useful scaffold, if the removed scaffold had high reward before being penalized for redundancy?

---

> ### Author Response · Authors · 2024-11-15
> **Author Response (1/1)**
>
> We thank the reviewer for their feedback and their interest in the prospect of optimizing MD oracles. This is also exciting for us and achieving this is a goal for us! We will answer each question individually. For questions that we address in the General Response, we will explicitly note this.
>
> ## **Q1: Optimizing MD simulations**
>
> Please see **General Response 1** which also shares preliminary results on direct Density Functional Theory (DFT) optimization, which can sometimes be even more expensive than MD (depending on the type). We would just like to take this opportunity to emphasize that moving from docking to MD is an enormous jump, and proper experimentation takes time (***and also access to sufficient GPU compute resources***). We have made a genuine effort in thoroughly benchmarking Saturn (***we ran every single experiment across 10 seeds***) and compared to > 20 models, including extremely strong baselines such as GEAM [1].
>
> ## **Q2: Does setting the reward to 0 for over-represented scaffolds limit the generator’s learning?**
>
> Given the local sampling behavior of Saturn, where generated molecules across similar Generator states can differ only by a few atoms, removing instances of over-represented scaffolds in the buffer is necessary. ***The purpose of this purging is to prevent detrimental mode collapse.*** If this is not done, the Generator can get stuck and only generate the same molecules repeatedly. In the original Augmented Memory [2] work, this was shown to a lesser extent. In that work, when the authors did not perform this purging, mode collapse was observed and due to no oracle caching mechanism (Saturn uses caching), this made Augmented Memory intolerable at higher augmentation rounds (how many times to learn from the augmented SMILES). In the original work, it was problematic to move beyond 2 augmentation rounds. The addition of oracle caching (store and re-use oracle evaluations), though straightforward, enables Saturn to drastically increase the number of augmentation rounds and still not lead to detrimental mode collapse. In Appendix C, we performed hyperparameter studies with augmentation rounds up to 20.
>
> Next, we want to comment on “learning from useful scaffolds”. In Saturn, Bemis-Murcko scaffolds are used which consider heavy atoms. Given the same scaffold, swapping any heavy atom with another heavy atom results in a different Bemis-Murcko scaffold. Therefore, purging over-represented scaffolds in the buffer does not prevent Saturn from learning from very similar scaffolds. The choice to use Bemis-Murcko scaffolds is to have this leniency.
>
> We are grateful for the reviewer’s time and hope we have answered their questions. We would also like to take this opportunity to highlight the DFT experiment we showed in **General Response 1.** In our experiment the direct optimization of full geometry in generative design. We hope to have conveyed that Saturn is a valuable step towards high-fidelity oracle optimization and kindly ask the reviewer to consider raising their evaluation of our work.
>
> Please let us know if you have any questions!
>
> \
> \
> [1] GEAM: https://arxiv.org/abs/2310.00841
>
> [2] Augmented Memory: https://pubs.acs.org/doi/10.1021/jacsau.4c00066

---

> > ### Author Response · Authors · 2024-11-28
> > **Eager to continue discussion**
> >
> > We want to thank the reviewer for taking the time to read our work.
> >
> > As the rebuttal period is nearing an end, we hope that the additional results on DFT optimization in the attached **Supplementary file** and **General Response 1** are convincing to the reviewer that Saturn is a step towards direct optimization of high-fidelity oracles. We are always eager to continue discussion.

---

### Official Review · Reviewer_SmbD · 2024-11-06

**Soundness:** 3
**Presentation:** 3
**Contribution:** 3
**Rating:** 6
**Confidence:** 3

**Summary:**

The authors present Saturn, an RL-approach using a Mamba backbone for generative molecular design. Design choices surrounding implementation are thoroughly explored and motivated towards increasing sample efficiency to enable the use of higher fidelity (and more computationally expensive) objective functions.

**Strengths:**

The authors focus on a weakness of the current trend in generative chemistry to couple an optimization algorithm with an insilico oracle function, suggesting two avenues for improvement: (i) increase the fidelity of the fast oracle, or (ii) increase the sample efficiency of the generative process to enable more costly oracles at reasonable throughput. In this paper they focus on (ii) which is a valuable contribution to the field.

The authors previously described the RL paradigm and here focus on substituting the language model backbone within the constraints that it is sufficiently simple to fine-tune on the fly. Given the recent success of Mamba as a linear time sequence modelling architecture this choice seems well motivated and suitable.

The reinforcement learning paradigm including SMILES augmentation, and experience replay buffer update with a genetic algorithm is intricately described along with its effect on the diversity of generated outputs.

**Weaknesses:**

Major:
- The main weakness of the method seems to be the diversity reduction exhibited upon moving to Mamba from an RNN backbone. Therefore the majority of my main questions are surrounding this observation.
  - The authors describe a logical diversity filter on the augmented memory buffer that uses bemis murcko as a filter. I am surprised that this addition does not seem to mitigate the reduction in diversity. Could the authors comment on this, and possibly show the murcko scaffold diversity along with their results in Table 2.
  - Following the argument at Line 292 that Mamba generates repeat SMILES because its loss is lower during pre-training, this argument would suggest overfitting. If the authors use the Mamba checkpoint with a loss matched to the RNN, is the performance restored to the RNN levels of diversity? At what sample efficiency cost?

Minor:
- The introduction should mention the use of QSAR predictive oracles since these are very common in addition to docking.
- The paper is motivated by an intent to use molecular dynamics as an oracle but does not achieve this goal, could the authors add a comment on what it would take to achieve this in the discussion section.
- The authors discuss the trade off of sample efficiency and diversity, this is similar to the common fidelity vs diversity tradeoff among image generative models. The authors might consider a 2D summary plot showing the tradeoff of these while varying batch size and n augmentation rounds since its a lot of information to digest in a tabular form.
- I am not clear on the logic presented in the section “squeezing the likelihood of Augmented SMILES” the authors might consider clarifying.
- Figure 2d compares the trajectory of generation throughout augmentation rounds. It is unclear that the Mamba path is more linear from these plots in my opinion. (i) The UMAP should be fit to the same set in order to compare and (ii) the RNN should also have 10 rounds.

**Questions:**

- Is there a possibility that the molecules generated by GraphGA impair the quality of the replay buffer molecules as this is an imperfect generative approach itself, would it be better to use a higher fidelity method to breed the members of the buffer.

---

> ### Author Response · Authors · 2024-11-15
> **Author Response (1/3)**
>
> We thank the reviewer for their feedback and questions on interrogating diversity. We will answer each question individually. For questions that we answer in the General Response, we will explicitly note this.
>
> ## **Q1: Diversity reduction is a weakness**
>
> Please see **General Response 3** for a discussion around diversity in generative molecular design.
>
> ## **Q2: Bemis-Murcko filter does not mitigate diversity reduction**
>
> Based on **General Response 3**, the diversity reduction is by design, but if more diversity is desired, one can increase batch size and decrease augmentation rounds. We comment more specifically on what the Bemis-Murcko filter is doing. The Bemis-Murcko is based on the “Diversity Filter”, first introduced in 2020 [1] as part of the REINVENT [2, 3] generative model. At the time, what this filter did was keep track of the scaffold generated in the run and if a scaffold is generated too many times (by some threshold), all future generated molecules possessing such scaffold have its reward truncated to 0. This heuristic helped REINVENT generate more diverse molecules. Augmented Memory [4] builds on REINVENT 3.2 [5] and in the original work, the authors found that this Diversity Filter is insufficient to prevent mode collapse. In **General Response 4 Sub-Response 1**, we describe in detail what Augmented Memory is doing. By learning from the same molecule too many times, mode collapse can occur which causes the generative model to generate the same molecules repeatedly. When this happens, the generative model is no longer generative. Therefore, in the Augmented Memory work, the authors introduced “Selective Memory Purge” which also purges the Replay Buffer of all molecules possessing penalized scaffolds. This was enough to rescue mode collapse. In Saturn, we adopt this mechanism and ***the purpose of the Bemis-Murcko filter is to prevent detrimental mode collapse.*** In Table 2 of the main text, we report the IntDiv1 on the molecules, which shows, again by design, that Saturn generates more high reward molecules possessing unique Bemis-Murcko scaffolds (Scaffolds metrics) than Augmented Memory, at a trade-off of diversity (IntDiv1 metric).
>
> ## **Q3: Mamba is overfitting. Can diversity be recovered to RNN level?**
>
> Our argument and results at around Line 292 is that Mamba overfits the distribution of augmented SMILES. This is done by design because it improves sample efficiency. We refer to **General Response 4 Sub-Response 1** again. Similar to our response to the previous question, increasing batch size and lowering augmentation rounds will recover diversity at the expense of sample efficiency (in all the case studies in the paper). For quantitative metrics, we cross-reference Appendix C for the tables on pages 29-30, where we report the trade-off between **Yield** and **Oracle Burden** sample efficiency metrics with **IntDiv1** and **Scaffolds** metrics for diversity.
>
> ## **Q4: Introduction should include QSAR models**
>
> We have added an additional sentence to introduce QSAR models:
>
> ***“We note that QSAR models are often used, which can have great predictive accuracy, but may suffer from a narrow domain of applicability (within their training data)”***
>
> ## **Q5: MD oracle**
> Please see **General Response 1.**
>
> ## **Q6: 2D summary plot of sample efficiency-diversity trade-off**
>
> Thank you for this suggestion. We have added a new 2D plot for Mamba batch size 16 (since these are the parameters of Saturn) in the Appendix (Fig. C3).
>
> ## **Q7: What does squeezing the likelihood of augmented SMILES mean?**
>
> The likelihood of a SMILES string is the negative log-likelihood (NLL) of its token sequence. Please see **General Response 4 Sub-response 1** for more details on Saturn’s optimization mechanism. Here, we will focus on what “squeezing” means. Given a SMILES, augmentation means generating a set of augmented SMILES. Every single augmented SMILES equates to a different token sequence and hence, different likelihood. In Fig. 2c, we show that executing Augmented Memory [4] makes the augmented SMILES more likely. This is shown by the delta NLL. In the main text, we expressed that the softmax function saturates, such that already ***likely*** SMILES, even if they exhibit a large loss, do not become ***much more likely***. Squeezing the likelihood means all the ***unlikely*** augmented SMILES converge towards the most likely one.

---

> ### Author Response · Authors · 2024-11-15
> **Author Response (2/3)**
>
> ## **Q8: UMAP trajectories**
>
> We performed the UMAP embeddings on the Mamba and RNN generated molecules sets separately because they start from (slightly) different pre-trained distributions (based on stochasticity of pre-training - the same dataset was used). The plot is meant to qualitatively depict how similar generated molecules are, as the generative experiment progresses. In Fig. 2e, we quantitatively show that the Mamba generated molecules are indeed more similar. We also show the UMAP for RNN with 10 augmentation rounds in Fig. D5 in the Appendix. With increased augmentation rounds, the RNN sampling trajectory looks a lot more like Mamba. However, we additionally quantitatively assess the similarity in the same Fig. D5 in a heatmap and show that Mamba is still more similar. This is in agreement with the ability of “overfit” distribution more, being advantageous for sample efficiency in the settings in the paper.
>
> ## **Q9: is GraphGA impairing the quality of the replay buffer molecules?**
>
> Thank you for this question - it was very interesting for us to study GraphGA’s effect during Saturn development! We would like to first discuss why we even used GraphGA. Saturn’s mechanism of optimization involves SMILES augmentation on the Replay Buffer molecules. In our view, we want to populate the Replay Buffer as fast as possible, because they store the highest reward molecules. As Saturn performs local exploration, we want to find high reward molecules as fast as possible to facilitate this local exploration. We provided our perspective on diversity in **General Response 3** that we think too much onus is placed on it sometimes, simply because it is a metric reported in literature. However, we do believe diversity is important in many cases and we used GraphGA to jointly interrogate diversity recovering (which in the end, is the proposed use in Saturn) and its effect on Replay Buffer populating. GraphGA [6], first proposed in 2019 is actually used as is, as a component in many generative models. ***In fact, GEAM [7] directly uses GraphGA.*** Interestingly, in 2021, a generative model using RL explicitly mentions using an evolutionary algorithm for the purpose of promoting diversity [8]. It is expected that a genetic algorithm can promote diversity, given the crossover and mutation operations, but we were also interested in its effect on sample efficiency, given an already sample-efficient model. ***We now answer how GraphGA affects the Replay Buffer.*** In Tables 24 and 25 in the Appendix, we report sample efficiency and diversity metrics when activating GraphGA in Saturn. We investigated how to select the offspring (random or Tanimoto dissimilarity to the Replay Buffer). Selecting by dissimilarity (to the Replay molecules) may promote more diversity, which was our hypothesis. The summary of the results is that choosing randomly outperforms choosing by dissimilarity. This was somewhat expected given that similar molecules, on average, have similar properties. Molecules more dissimilar to the Replay Buffer molecules would be less likely to possess good properties. In the tables, we report **“Buffer Replace”** (how many GraphGA molecules replaced molecules in the buffer? Which means how many were better than at least the top-100 molecules by reward.) and **“Buffer Best”** (how many times were GraphGA molecules the best in the buffer throughout the run?). ***The summary of the findings is that GraphGA generated molecules do indeed replace the buffer molecules but rarely are they the best.*** In the end, activating GraphGA leverages its crossover and mutation operations to recover diversity but does not improve sample efficiency. We now answer the reviewer’s question regarding higher fidelity methods to breed Replay Buffer molecules. Recent work [9, 10] showed that fragment retrieval can be beneficial for sample efficiency. This would entail choosing similar molecules or fragments to those present in the Replay Buffer and using these as a higher-fidelity breeding method/replacement method. This would require a priori known fragments though. This being said, this is an area we are also interested in as future work and we thank the reviewer again for this comment.

---

> ### Author Response · Authors · 2024-11-15
> **Author Response (3/3)**
>
> We are really grateful for the reviewer’s time and their feedback. In particular, their suggestion for a 2D plot to visualize the diversity/sample efficiency trade-off was particularly helpful for us to better convey our results. We would also like to take this opportunity to highlight the DFT experiment we showed in **General Response 1**. In our experiment, we showed the direct optimization of full DFT geometry in generative design. We hope to have conveyed that Saturn is a valuable contribution towards high-fidelity oracle optimization and understanding optimization dynamics.
>
> Please let us know if you have any questions and we would be grateful if the reviewer would consider raising their evaluation of our work!
>
> \
> \
> [1] Diversity Filter: https://jcheminf.biomedcentral.com/articles/10.1186/s13321-020-00473-0
>
> [2] REINVENT 1.0: https://jcheminf.biomedcentral.com/articles/10.1186/s13321-017-0235-x
>
> [3] REINVENT 2.0: https://pubs.acs.org/doi/10.1021/acs.jcim.0c00915
>
> [4] Augmented Memory: https://pubs.acs.org/doi/10.1021/jacsau.4c00066
>
> [5] REINVENT 3.2: https://github.com/MolecularAI/Reinvent
>
> [6] GraphGA: https://pubs.rsc.org/en/content/articlelanding/2019/sc/c8sc05372c
>
> [7] GEAM: https://arxiv.org/abs/2310.00841
>
> [8] DrugEx2: https://jcheminf.biomedcentral.com/articles/10.1186/s13321-021-00561-9
>
> [9] RetMol: https://arxiv.org/abs/2208.11126
>
> [10] Fragment Retrieval at NeurIPS 2024: https://f-rag.github.io/

---

> > ### Author Response · Authors · 2024-11-28
> > **Eager to continue discussion**
> >
> > We want to thank the reviewer again for their constructive feedback, and especially the question around GraphGA as this was something we found very interesting during development.
> >
> > As the rebuttal period is nearing an end, we want to ensure we have at least properly conveyed our findings and are always open to continue discussion.

---

### Author Response · Authors · 2024-11-15
**Manuscript Changes**

### **Manuscript Changes**

All changes are highlighted in the updated manuscript version.

* Additional sentence introducing QSAR models in the Introduction **following Reviewer SmbD’s feedback**

* Fig. C3 in the Appendix showing a 2D heatmap of sample efficiency and diversity trade-off for Mamba with batch size 16 (Saturn uses this). **This follows Reviewer SmbD’s feedback**

* Contributions list previously mentioned performing > 5,000 experiments. **Following Reviewer f1gd’s feedback**, we changed this to “> 500 experiments, all across 10 seeds)”

* Appendix C.2 description of how SMILES augmentation is performed **following Reviewer XhFy’s feedback**

* Appendix F.2 wall times for reproducing GEAM **following Reviewer u4yG’s feedback**. Saturn’s wall times were previously already reported in Appendix F.1. Saturn’s generation time is faster.

* Page 4, clarity on what “augmentation round” means **following Reviewer f1gd’s feedback**.

* Page 4, added sentence about fixing the docking oracle seed which to support our statement about deterministic oracles **following Reviewer f1gd’s feedback**

* Page 5, clarity on what “local sampling” means **following Reviewer f1gd’s feedback**

### **New Results (attached zip file)**

* Showing direct high-fidelity oracle optimization. Density Functional Theory (DFT) oracle optimization - plots and discussion attached as a supplementary file

* Further analysis between Saturn and GEAM results showing that GEAM generates molecules with lower ligand efficiency and are larger, which can inflate docking scores. This is due to GEAM not optimizing QED to the same extent as Saturn.

---

> ### Author Response · Authors · 2024-11-15
> **General Response (1/6)**
>
> We would like to express our gratitude to the reviewers for their constructive feedback and questions related to all aspects of the paper. There are some shared concerns/questions amongst reviewers which we wanted to first address. ***We wanted to provide a comprehensive answer and we would be grateful if the reviewers could consider the full text.***
>
> ## **1. Showing explicit optimization of high-fidelity oracles**
> * **Reviewers SmbD, feFf, XhFy:** show MD results
>
> We are excited that reviewers are interested in the potential to directly optimize MD. **Reviewer feFf** suggests that good performance on comparatively low-fidelity (docking) oracles does not necessarily translate to higher-fidelity oracles. We agree with this statement, but want to emphasize that there exists no work that directly optimize for MD. The closest are MFBind [1] which uses multi-fidelity active learning to sometimes queries an MD oracle to update the surrogate predictor model and REINVENT + MD [2] which uses active learning with supercomputer initial dataset generation (via MD). All generative molecular design works use docking as the oracle to predict binding affinity and docking can be very useful. We want to reference this recent review paper [3] which compiles all experimentally validated examples of generative models for drug candidate design. **Every single structure-based design case study used docking.** Therefore, showing Saturn is sample efficient on docking oracles is a valuable contribution.
>
> Going from docking to MD imposes some engineering challenges (cluster related ) and **access to sufficient compute resources. MD requires GPUs and each simulation imposes GPU hours.** Consider a real life case study where a model generates a batch of molecules. *Sequentially* computing an MD oracle would take # molecules * MD time per generation epoch. This is prohibitively time expensive. Instead, we want to compute the entire batch in parallel, which would require 1 GPU per molecule or multiple molecules parallelized on a single GPU (the compute/speed trade-off will be case dependent). This is also why we use a batch size of 16 because while <= 16 GPUs **may be possible in academic labs**, access to more GPUs is quite prohibitive. Therefore, we respectively suggest that it is not fair to suggest we show MD results as there are no works that can do this or attempted this. It is an **enormous** jump from docking to MD. This is why there is so much work on making MD faster [4]. Our progress on optimizing docking is a valuable contribution.
>
> However, to show some evidence that our results can transfer to higher-fidelity oracle settings, ***we will show preliminary results of direct optimization of Density Functional Theory (DFT) oracles.*** DFT also requires parallelization but is possible on lots of CPUs. We will show direct optimization of the HOMO-LUMO gap which can be useful for molecular stability [5]. There is to the best of our knowledge, only 1 other work [**GA Suzuki: more information at the bottom**] that directly optimizes DFT. Other existing works use both xTB (lower-fidelity) and DFT on the most promising molecules (or only single-point DFT), but not DFT alone. Here, we will show, full geometry optimization (**this is the expensive part**) with DFT and the direct minimization of the HOMO-LUMO gap using the B3LYP functional with D3 dispersion correction and def2-TZVP basis set [6-10]. We used Saturn (out-of-the-box, batch size 16, 10 augmentation rounds) with an **oracle budget of 300**. Every molecule in the batch is parallelized on 72 CPUs. 16 molecules = 16 nodes, each node with 72 CPUs. CPUs are more feasible for us to access and **we want to convey that had we used a larger batch size, this would have been prohibitive.** It is difficult to report an exact wall time as we had to use a shared compute cluster and jobs could be stalled in queue. In fact, if we had >> 16 molecules, this would’ve been almost sequential due to cluster traffic, which would be prohibitively time expensive. Each molecule generally took 30 minutes to 2 hours depending on the size, as DFT scales $O(N^3)$. This is why DFT is expensive. ***Please see the attached supplementary file for the results.***
>
> ***We hope by showing this explicitly, we convey to the reviewers that Saturn is a valuable step towards directly optimizing high-fidelity oracles and that they would consider raising their evaluation of our work.***
>
>
> **November 29 edit:** We were made aware of 1 recent work that also performed full DFT geometry optimization but for transition metal catalysts. Importantly, the genetic algorithm here swaps ligands *based on a pre-defined list of 91 ligands* which heavily restricts the generative freedom (though this can be useful depending on the use case). Therefore, we believe showing direct DFT optimization on an *unconstrained* model is a meaningful result.
>
> https://chemrxiv.org/engage/chemrxiv/article-details/6641eea7418a5379b02375bd

---

> ### Author Response · Authors · 2024-11-15
> **General Response (2/6)**
>
> ## 2. **Saturn and GEAM: Performance and Tanimoto dissimilarity**
>
> * **Reviewer XhFy:** Adding Tanimoto dissimilarity to GEAM and make the comparison
>
> * **Reviewer f1gd:** Saturn performs worse than GEAM so Tanimoto dissimilarity was introduced. Why was this not done for GEAM?
>
> We want to first clarify the results between Saturn and GEAM around performance. **Reviewer f1gd** states that Saturn performs really bad compared to GEAM. We first acknowledge that GEAM is a strong baseline and was the main reason we compared to them (although their final publication date is almost concurrent work, ICML 2024).
>
> We want to begin this response by highlighting that GEAM proceeds by first constructing an initial fragment vocabulary, which is trained on ZINC 250k. ***Every single molecule in ZINC 250k was actually computed by the oracle to enable this step.*** If the oracle were particularly expensive, this makes “pre-training” GEAM also very expensive. This is very different to the unsupervised pre-training scheme of Saturn. We are only maximizing the likelihood of re-constructing ZINC 250k molecules during pre-training. ***This does not require any oracle calls.*** So while we evaluated Saturn and GEAM both on 3,000 oracle calls, GEAM’s pre-training step actually technically incurs 250k oracle calls. ***We hope by this fact alone, that reviewers can view Saturn’s sample efficiency more favourably.*** We continue with our response.
>
> To compare exactly with GEAM, we followed their case study and used the provided pre-training data and oracle code in their codebase. GEAM reports results for 2 metrics:
>
> * **Hit Ratio:** better docking score than reference ligand, QED > 0.5, SA < 5
>
> * **Novel Hit Ratio:** Hit Ratio with the added constraint that the molecule < 0.4 Tanimoto similarity to the pre-training data (ZINC 250k)
>
> In our work, we further defined:
>
> * **Strict Hit Ratio:** better docking score than reference ligand, QED > 0.7, SA < 3
>
> * **Strict Novel Hit Ratio:** Strict Hit Ratio with the Tanimoto similarity constraint
>
> ***Why did we enforce stricter thresholds?*** Because jointly optimizing docking score, QED, and SA score is the objective.
>
> We now discuss the **Hit Ratio** results. In Table 3, we show Saturn outperforms or matches GEAM. However, investigating further with the **Strict Hit Ratio** in Table 5 shows that GEAM does not generate optimal molecules according to the optimization objective. The majority of the generated molecules do not possess QED > 0.7 and SA < 3. ***But are we just defining an arbitrary metric to make Saturn look better?*** No, and we cross-reference GEAM’s reward function, which we also write out in the Saturn paper as equation 5. The objective is simply to jointly optimize all these metrics. Molecules with 0.5 QED and 5 SA cannot give maximum reward.  ***Therefore, this set of results is showing that Saturn is optimizing the objective to a much greater extent. In generative design, we want optimal molecules.***
>
> Next, we discuss the **Novel Hit Ratio** results. **Reviewer f1gd** has expressed that Saturn performs much worse than GEAM. In Table 4, Saturn indeed has much lower **Novel Hit Ratios** than GEAM. ***We would first like to pose the question, why do we want generated molecules to possess less than 0.4 Tanimoto similarity to the pre-training data?*** The argument is that we want dissimilarity to generate “new” molecules. We want to first motivate that this threshold is somewhat arbitrary. In 2014, a study found that > 90% FDA approved drugs have > 0.5 Tanimoto similarity to human metabolites [12]. An example of one of these metabolites is “carnosine” which is present in ZINC (ZINC2040854 and ZINC8583964). We note that it is not present in ZINC 250k, but had the pre-training data been another fraction of ZINC, it could be present. This paper suggests that similarity to these molecules might actually be desired for bioactive safety. ***In this case, we may not even necessarily want particular dissimilarity.*** Conversely, though, another study in 2014 suggested that Tanimoto similarity < 0.5 could be used to distinguish “similar to non-similar” pairs of molecules for orphan drug registration [13]. ***In this case, dissimilarity may be desired.*** We wanted to highlight this because similarity to some reference set of molecules is not necessarily bad, and that the metric is somewhat arbitrary. Now answering the question explicitly, we wanted to show that if this was a desired constraint, that it is straightforward to adapt Saturn to satisfy it by simply running a ***few minutes*** of optimizing dissimilarity to the pre-training data.
>
> The reviewer suggested running “GEAM Tanimoto” but GEAM constructs their initial fragment vocabulary to be aligned with the objective ***based on all of ZINC 250k which needs to be computed by the oracle.*** Training it to be dissimilar to the pre-training data would prevent GEAM from constructing such a vocabulary that is aligned with the objective.

---

> > ### Comment · Reviewer_f1gd · 2024-11-20
> >
> > "The reviewer suggested running “GEAM Tanimoto” but GEAM constructs their initial fragment vocabulary to be aligned with the objective based on all of ZINC 250k which needs to be computed by the oracle. Training it to be dissimilar to the pre-training data would prevent GEAM from constructing such a vocabulary that is aligned with the objective."
> >
> > Is there any experiment to support your argument?

---

> > > ### Author Response · Authors · 2024-11-20
> > > **Response to Reviewer f1gd**
> > >
> > > GEAM uses a "Fragment-wise Graph Information Bottleneck" (FGIB) module that extracts informative fragments. This process uses the labelled ZINC 250k dataset which has pre-computed all the oracle values for the docking targets. The training scheme is presented in their code repository: https://github.com/SeulLee05/GEAM?tab=readme-ov-file. From this extracted fragment vocabulary, GEAM begins generation where the reward is the same as the pre-computed reward already. This is what we mean by extracting a vocabulary already aligned with the objective. If the reward is now dissimilarity to ZINC 250k, then the fragment vocabulary would change and no longer be the optimal fragments extracted from the supervised pre-training. This is because the fragment vocabulary would have to be aligned with dissimilarity instead of the docking oracles which was the point of GEAM's pre-training in the first place.

---

> ### Author Response · Authors · 2024-11-15
> **General Response (3/6)**
>
> ## **3. Clarifications around diversity**
>
> * **Reviewer SmbD:** Main weakness is diversity reduction
>
> * **Reviewer feFf:** Does removing over-represented scaffolds limit the generative ability?
>
> * **Reviewer XhFy:** Would tweaking the hyperparameters of GEAM be able to trade-off diversity for improved sample efficiency? If so, would it outperform Saturn?
>
> * **Reviewer u4yG:** Trading off diversity may affect the capability for exploratory molecular design
>
> For the specific reviewer questions regarding diversity and its effect on generative ability/GEAM, we will respond in the individual responses. Here, we discuss diversity as a general metric.
>
> In molecular design literature, it is common to report a notion of diversity, via IntDiv1 (internal pair-wise Tanimoto similarity) and/or #Circles (packing number), which we report both in our work. However, aside from reporting a metric, ***why exactly*** do we want diversity? Typically, in real life projects, generated sets of molecules are clustered and then representative molecules (with optimal properties) selected to be subjected for MD simulations and finally prioritized for experimental validation. Diversity is often desired because proxy oracles (like docking) can lead to many false positives. Having other “good” molecules can essentially act as back-up plans. If we now consider the prospect of directly optimizing higher-fidelity oracles, they should in principle lead to less false positives. There is literature precedent for this [4 is an example]. In this scenario, “high” diversity may not actually be required because we have much higher confidence in our oracle. Perhaps in such an event, we are interested in the number of oracle calls required to generate molecules with optimized oracle values. ***This is why we report Oracle Burden (# oracle calls required to generate N unique molecules above a reward threshold).*** Our sub-response so far is to offer some discourse on diversity and ***treating it as an actionable objective so it is aligned with real life use***, rather than just as a metric that is reported.
>
> Next, we discuss the diversity reduction and that it is ***by design.*** In Appendix C, we performed extensive ablation experiments to ultimately demonstrate that low batch size and high augmentation rounds leads to higher sample efficiency at the expense of diversity. ***What is the model doing?*** It is performing local exploration, generating molecules that may differ only by a few atoms. Please see Fig. D6 in the Appendix where we explicitly illustrate this with molecular structures. We find that for all docking case studies, purposely trading off diversity for sample efficiency is beneficial. ***If desired, one can increase batch size and decrease augmentation rounds to recover diversity.*** We show this explicitly in Appendix C, and reference the IntDiv1 values reported in the ablation experiments.
>
> At a very high level, what do we want from a generative model? We want molecules that possess optimal properties. We then pose a question: ***Would it be more advantageous to generate less diverse molecules that optimize the multi-parameter optimization (MPO) objective to a greater extent, than generating more diverse molecules that only moderately optimize the MPO objective?*** We argue that optimizing the MPO objective is most important, because we want optimal molecules. This is exactly what our Part 3 experimental results are commenting on. In our comparison with GEAM [11], we followed their MPO objective (jointly optimize docking, SA score, QED). Following their Hit Ratio metric, they filter molecules with SA score < 5 (**optimal is 1**) and QED > 0.5 (**optimal is 1**). This means optimal molecules according to the MPO objective should have much lower SA score than 5 and much higher QED than 0.5. This is why we introduced the Strict Hit Ratio which filters SA score < 3 and QED > 0.7. In Table 5, we show that most of the molecules generated by GEAM actually do not pass these thresholds. ***Whether or not low SA and high QED translates to real life success is not the point of this strict filter. It is to highlight that a generative model should be able to generate optimal molecules according to the MPO objective function.***

---

> ### Author Response · Authors · 2024-11-15
> **General Response (4/6)**
>
> ## **4. Novelty: What is the contribution of our work?**
>
> * **Reviewer XhFy:** Saturn just applies Augmented Memory but with Mamba
>
> * **Reviewer f1gd:** The novelty of this paper seems to be limited to the application of Mamba
>
> While to the best of our knowledge, we are the first to apply the Mamba architecture for molecular generation, we do not believe this is the main contribution of our work. In the Introduction, we stated the following three additional contributions:
>
> * Elucidate the mechanism of Augmented Memory [14]
>
> * Comprehensively evaluate language model backbones to show model-intrinsic and scaling properties that lead to improved sample efficiency.
>
> * Demonstrate local sampling in chemical space can be a key component for sample efficiency. Our results provide discourse on the nature of optimization landscapes commonly encountered in drug discovery.
>
> We omit discussion here on the empirical results as the reviewers above cite this as a strength. **Instead, we focus our discussion here on the deeper architectural and optimization insights in the context of molecular design.**
>
> ### **1. Elucidating the mechanism of Augmented Memory and showing its effect at both the RL and chemistry levels**
>
> In the original Augmented Memory work, the authors only showed its empirical benefits. In this sub-response, we will first discuss the steps of Augmented Memory, and then discuss its effect during RL, and finally show that by understanding this mechanism, we can have granular control over the “type” of chemistry that is generated.
>
> **Augmented Memory.** Given a molecular graph, a SMILES representation is generated by picking a node (atom) and performing a depth-first search (as is done in RDKit). The traversal order yields **a specific SMILES form**. Starting from another node results in an alternative SMILES form. **This is known as SMILES augmentation** and the key fact is that **all augmented SMILES map to the same molecular graph.** This non-injective property means that if many SMILES forms of the same molecular graph are generated, **the same reward can be assigned to them.** This is what allows Augmented Memory to “learn from the same molecule many times”. In the original Augmented Memory work, the authors showed that the act of SMILES augmentation is vital and that if only one single SMILES form is used and learned repeatedly from, detrimental mode collapse occurs. This results in the model generating only 1 molecule, thus losing generative ability. ***What does this have to do with Saturn?*** In Part 1, we studied the exact mechanism of what Augmented Memory is actually doing: by learning (via RL) from different SMILES forms, it becomes more likely to generate any SMILES representation of the same molecular graph. In Part 1, we did a sub-experiment where we took all the SMILES in the Replay Buffer and performed 10x SMILES augmentation on them, resulting in 1,000 SMILES. We ran Augmented Memory on these SMILES and then computed the difference in their likelihoods of being generated before and after Augmented Memory. We cross-reference Fig. 2c which shows that after 1 round of Augmented Memory, the NLL shift can be quite large, indicating increased likelihood of generating the augmented SMILES forms. ***What does this mean at the chemistry level?*** If it becomes likely to generate some SMILES representation of the same molecular graph, then small changes in the SMILES sequences (via token sampling stochasticity) results in only minor changes to the molecular graph. We cross-reference Fig. D6 in the Appendix which shows that unique molecules generated at similar Agent states share significant similarity, often differing only by a few atoms. **This means that Saturn performs local sampling around “good” molecules and is the reason why there is a trade-off in diversity.** This local sampling behavior can even interrogate the neighborhood of “good” molecules, which is important in drug discovery for structure-activity relationships (SAR). Recent work even trains models to do this task for use in commercial drug discovery [15]. Finally, this set of results shows the SMILES augmentation is actually quite an effective regularizer - despite learning from the same molecular graph so many times, the model does not suffer from detrimental mode collapse. We cross-reference Fig. 2a which takes Agent states across 20 generation epochs (at every epoch, Augmented Memory is run), and computes the average max token sampling probability: during generation, track the highest probability token at each step and then take the average. The Figure shows that Saturn ***approaches*** mode collapse (if the average generation probability = 1), but does not do so which is extremely important, as the generative ability of the model is retained. **The final message we convey in this sub-response is that Saturn intentionally overfits on augmented SMILES which directly leads to improved sample efficiency by trading off diversity.**

---

> ### Author Response · Authors · 2024-11-15
> **General Reponse (5/6)**
>
> ### **2. Model-intrinsic and scaling properties that lead to improved sample efficiency**
>
> Building on the last sentence in the previous section, “intentional overfitting can benefit sample efficiency”, we perform deeper investigations to answer 2 questions:
>
> ***(1) Is Mamba intrinsically better than RNN and decoder Transformer?***
>
> ***(2) How does scaling improve architectures?***
>
> We begin our response by cross-referencing Table 1 in the main text which contrasts RNN (5.8M), Decoder transformer (6.4M), and Mamba (5.2M). The key metric we want to highlight is **“Repeats”**. This tracks how many repeat molecules were generated throughout the run. Note that Repeat molecules do not need to be scored by the Oracle. **Why is Mamba generating so many more repeat molecules compared to RNN and decoder Transformer?** This is because it “overfits” the distribution when Augmented Memory is applied. This is also what is being shown in Fig. 2a which we also referenced in the previous section: Mamba approaches mode collapse. This sub-response so far only conveys the observation that under the current model sizes, Mamba can capitalize on “overfitting” for sample efficiency. ***But is this Mamba intrinsic?*** To answer this question, we scaled up the RNN (from 5.8M to 24.7M) and decoder Transformer (from 6.4M to 25.3M) for use in Part 3 experiments. ***Does scaling up the other architectures close the gap to Mamba?*** Yes, and we show this in two ways. First, all models need to be pre-trained (on ZINC 250k). The loss with larger model sizes converges to lower NLL (maybe not exactly surprising, but we report the numbers for completeness):
>
> * **RNN (5.8M):** NLL = 30.88
>
> * **Mamba (5.2M):** NLL = 28.10
>
> * **RNN (24.7M):** NLL = 29.32
>
> * **Decoder Transformer (25.3M):** NLL = 26.96
>
> ***How does this translate to sample efficiency?*** Using these larger models, we ran Part 3 experiments again across 10 seeds (0-9 inclusive). ***All the results are in Appendix F.6 Saturn: Architecture Scaling.*** The results show that the larger models have notably better sample efficiency with RNN (24.7M) still worse than Mamba (5.2M) but Decoder Transformer (25.3M) about the same - these larger models once again trade off diversity. This is the expected behavior. As we have shown that local sampling can improve sample efficiency, a model that can fit distributions better directly translates to such sampling behavior. Does this guarantee that this local sampling behavior is beneficial for all optimization tasks? No, but in all generative literature to date, docking tasks with physicochemical property modulators are used and we have shown through 8 docking targets that this indeed is beneficial. We believe these insights are interesting and directly lead to our next sub-response. We end this sub-response by asking: ***if larger RNN and Decoder Transformer can close the gap between Mamba, is Mamba still advantageous?*** We believe so because Mamba can use just 5.2M parameters, thus requiring less GPU memory. This is practically useful, especially if other oracles require GPU memory (GPU docking being a prime example). ***Mamba (5.2M) requires only ~2GB GPU memory to run so it fits on basically any GPU.*** Light-weight models that can be tuned quickly are practically advantageous. Finally, we note that one could scale up the model to even larger but we want to highlight that the pre-training data only contains ~250k molecules (ZINC 250k), unlike the classic LLM paradigm. Larger models require more compute resources and we chose to allocate more compute resources to run the Part 3 experiments across 10 seeds for thoroughness.

---

> ### Author Response · Authors · 2024-11-15
> **General Response (6/6)**
>
> ### **3. Local sampling and optimization landscapes encountered in drug discovery**
>
> Building on the last point in the previous sub-response: Saturn performs well via local sampling - is beneficial for all optimization tasks? We have shown how strategic overfitting and model scaling leads to local sampling behavior that can improve sample efficiency for all docking case studies in the paper. We approach this question in 2 scenarios:
>
> ***(1) Provided that local sampling is desired, what do our results suggest for interesting future work?***
>
> ***(2) Provided that local sampling is not desired, what do our results suggest for interesting future work?***
>
> If local sampling leads to high sample efficiency as is the case in this work, then paralleling LLMs, scaling up pre-training can yield benefits because we can start from a better initial distribution. A better initial distribution can allow the model to find a small set of “good” molecules as fast as possible, and then locally explore. Indeed, recent work has shown that moving beyond the common ZINC 250k (about 250k molecules) and ChEMBL (about 2 million molecules) to PubChem (about 100 million molecules) for pre-training can directly lead to higher sample efficiency, **all else fixed** [15, 16]. While this may seem obvious in light of LLMs, chemistry data can be quite different and often you should be prudent in combining data from different sources [17]. We pre-train because we believe that the pre-training data is useful for our task. For a counter-example, consider a recent ICLR 2024 paper [18] which shows that pre-training on more suitable, but less data, is better than just more data. This is expected and PubChem, containing many bioactive molecules, is often suitable, as shown by the cited works [15, 16]. In this case, future work can investigate pre-training for generative design, for instance work at ICLR 2024 [19]
>
> What if local sampling is not desired? Through all the ablation experiments performed in our work, we have shown ***how*** batch size and augmentation directly control the exploration-exploitation trade-off and we have also shown how these hyperparameters affect the type of chemistry generated (as discussed in sub-reponse 1, we cross-reference Fig. D6 in the Appendix). We can ***predictably control*** the sampling behavior of Saturn to control for the types of chemistry (local atom changes or not) we want to generate.
>
> To conclude, we believe that the development of Saturn has led to valuable insights into what components can lead to a sample-efficient model. We have studied the effect of architecture scaling and its implications on sample efficiency and the diversity of results. Beyond this, we have proposed a model that achieves state-of-the-art sample efficiency when compared to **GEAM (ICML 2024)** which is the strongest baseline and on their optimization task **and without a pre-training scheme that requires 250k oracle calls.** Throughout the entire study, we have made an effort to be rigorous, performing every experiment for 10 replicates (10 seeds, 0-9 inclusive so no cherry-picking seeds). We hope our general response is useful in clarifying Saturn’s contributions.
>
> Please let us know if you have any further questions!
>
> Sincerely,
>
> The Authors
>
> \
> \
> [1] MFBind: https://arxiv.org/abs/2402.10387
>
> [2] REINVENT + MD surrogate: https://pubs.acs.org/doi/10.1021/acs.jctc.4c00576
>
> [3] Review Paper: https://www.nature.com/articles/s42256-024-00843-5
>
> [4] AQFEP: https://pubs.acs.org/doi/10.1021/acs.jctc.4c00399
>
> [5] HOMO-LUMO: https://pmc.ncbi.nlm.nih.gov/articles/PMC10569544/#pone.0283271.ref056
>
> [6] D3 dispersion: https://pubs.aip.org/aip/jcp/article/132/15/154104/926936/A-consistent-and-accurate-ab-initio
>
> [7] B3LYP: https://journals.aps.org/prb/abstract/10.1103/PhysRevB.37.785
>
> [8] B3LYP: https://pubs.aip.org/aip/jcp/article/98/7/5648/842114/Density-functional-thermochemistry-III-The-role-of
>
> [9] def2-TZVP: https://pubs.rsc.org/en/content/articlelanding/2005/cp/b508541a
>
> [10] def2-TZVP: https://pubs.rsc.org/en/content/articlelanding/2006/cp/b515623h
>
> [11] GEAM: https://arxiv.org/abs/2310.00841
>
> [12] Metabolite similarity: https://link.springer.com/article/10.1007/s11306-014-0733-z
>
> [13] Orphan drug similarity: https://jcheminf.biomedcentral.com/articles/10.1186/1758-2946-6-5
>
> [14] Augmented Memory: https://pubs.acs.org/doi/10.1021/jacsau.4c00066
>
> [15] Mol2Mol: https://www.nature.com/articles/s41467-024-51672-4
>
> [16] REINVENT Transformer: https://jcheminf.biomedcentral.com/articles/10.1186/s13321-024-00887-0
>
> [17] Combining chemistry data: https://pubs.acs.org/doi/10.1021/acs.jcim.4c00049
>
> [18] Pre-training data in RL: https://openreview.net/forum?id=nqlymMx42E
>
> [19] JMP pre-training: https://openreview.net/forum?id=PfPnugdxup

---

### Author Response · Authors · 2024-11-20
**We are eager to discuss with reviewers**

As half the discussion period has passed, we wanted to acknowledge that our responses are long but we wanted to be thorough and would be grateful if reviewers can consider the points we raised. **We only want to reiterate the following points:**

### **1. GEAM's pre-training requires a labelled dataset**
All ~250k molecules in ZINC 250k were docked (across all 5 protein targets). ***Therefore, GEAM pre-training requires an up-front cost of 1.25 million docking oracle calls (250k molecules, 5 protein targets).*** GEAM then trains a model separately for each protein target. This is in stark contrast to Saturn's unsupervised pre-training, which requires no up-front oracle calls and the pre-trained model is applied as is, to all protein targets. The shared sentiment amongst reviewers is that GEAM's performance is similar to Saturn. ***We just want to note that we are evaluating Saturn at an extreme disadvantage***. Despite this, Saturn still optimizes the objective to a ***much greater*** degree (**see Strict Hit Ratio results**).

### **2. GEAM’s generation scheme**
***Always*** starts from benzene and adds fragment attachments to construct the full molecule. This means it cannot generate molecules without benzene. However, due to their use of GraphGA directly as their genetic algorithm, final generated molecules *may* not have benzene due to crossover and mutation operations in the GA. Nonetheless, the constraint of always starting from benzene is a practical limitation.

### **3. DFT optimization**
Given the above points, we hope to have conveyed to the reviewers that Saturn considerably outperforms GEAM. We go further to show preliminary results on direct DFT optimization (**attached supplementary file**). We believe most models are not close to this prospect of direct high-fidelity oracle optimization.

\
\
We are eager to engage in discussion with the reviewers to improve our work and incorporate feedback.

---

### Meta-Review · Area_Chair_Pmur · 2024-12-17

**Metareview:**

This submission introduces Saturn for generative molecular design based on the language model backbone Mamba. Saturn integrates augmented memory in RL for more efficient generation with strong empirical performance in drug discovery.

The reviewers agreed that the performed experiments, including the ones during rebuttal, demonstrated the sample efficiency and generation performance by Saturn. However, most of the reviewers consider that the technical contributions are limited given both Mamba and augmented memory mechanisms are not new. The authors in the rebuttal explained that the novelty lies in elucidating "the deeper architectural and optimization insights in the context of molecular design".

The authors may consider the review comments for possible future submission, better positioning the research work, presenting the technical contributions in more self-explaining and rigorous ways, and providing stronger evidence of how the design of Saturn and obtained insights may help further improve molecular design.

**Additional Comments On Reviewer Discussion:**

Most of the reviewers have participated during the author-reviewer and/or reviewer-AC discussion phases. During these discussions, the authors have provided clarification of the research scope as well as additional experimental results, demonstrating the effectiveness of Saturn in generative molecular design. However, most of the reviewers still have concerns regarding the technical novelty of the presented work. The reviewers believe that, while the submission presents "a useful application work with extensive experimentation and well documented codebase", the technical novelty is still limited. "This paper will be more suitable for other high impact avenue where the readership is more inclined towards drug discovery applications."

---

### Decision · Program_Chairs · 2025-01-22

Reject